# A revised map of volcanic units in the Oman ophiolite: insights into the architecture of an oceanic proto-arc volcanic sequence

Thomas M. Belgrano[a]*, Larryn W. Diamond[a], Yves Vogt[a], Andrea R. Biedermann[a,b], Samuel A. Gilgen[a], Khalid Al-Tobi[c]

*Corresponding Author (thomas.belgrano@geo.unibe.ch)

[a]Institute of Geological Sciences, University of Bern, Baltzerstrasse 3, 3012 Bern, Switzerland
[b]Institute for Rock Magnetism, University of Minnesota, 100 Union St SE, 55455 Minneapolis, USA
[c]National Earth Secrets Co., P.O. Box 1242, PC 130 Athaibah, Muscat, Sultanate of Oman

**Abstract**

Numerous studies have revealed genetic similarities between Tethyan ophiolites and oceanic 'proto-arc' sequences formed above nascent subduction zones. The Semail ophiolite (Oman–U.A.E.) in particular can be viewed as an analogue for this proto-arc crust. Though proto-arc magmatism and the mechanisms of subduction-initiation are of great interest, insight is difficult to gain from drilling and limited surface outcrops in marine settings. In contrast, the 3–5 km thick upper-crustal succession of the Semail ophiolite, which is exposed in an oblique cross-section, presents an opportunity to assess the architecture and volumes of different volcanic rocks that form during the proto-arc stage. To determine the distribution of the volcanic rocks and to aid exploration for the volcanogenic massive sulphide (VMS) deposits that they host, we have re-mapped the volcanic units of the Semail ophiolite by integrating new field observations, geochemical analyses and geophysical interpretations with pre-existing geological maps. By linking the major element compositions of the volcanic units to rock magnetic properties, we were able to use aeromagnetic data to infer the extension of each outcropping unit below sedimentary cover, resulting in in a new map showing 2100 km$^2$ of upper-crustal bedrock.

Whereas earlier maps distinguished two main volcanostratigraphic units, we have distinguished four, recording the progression from early spreading-axis basalts (Geotimes), through axial to off-axial depleted basalts (Lasail), to post-axial tholeiites (Tholeiitic Alley), and finally boninites (Boninitic Alley). Geotimes ('Phase 1') axial dykes and lavas make up ~55 vol% of the Semail upper crust, whereas post-axial ('Phase 2') lavas constitute the remaining ~45 vol% and ubiquitously cover the underlying axial crust. Highly-depleted, boninitic members of the Lasail unit locally occur within and directly atop the axial sequence, marking an earlier onset of boninitic magmatism than previously known for the ophiolite. The vast majority of the Semail boninites, however, belong to the Boninitic Alley unit and occur as discontinuous accumulations up to 2 km thick at the top of the ophiolite sequence and constitute ~15 vol% of the upper crust. The new map provides a basis for targeted exploration of the gold-bearing VMS deposits hosted by these boninites. The thickest boninite accumulations occur in the Fizh block, where magma ascent occurred along crustal-scale faults that are connected to shear zones in the underlying mantle

rocks, which in turn are associated with economic chromitite deposits. Locating major boninite feeder zones may thus be an indirect means to explore for chromitites in the underlying mantle.

## 1. Introduction

A growing body of geochemical and geochronological evidence indicates that the crustal sequences of many Tethyan ophiolites—including the Oman–U.A.E. (Semail) example—formed in response to upper-plate extension during the initiation of lower-plate subduction and roll-back (Belgrano and Diamond, 2019; Dilek et al., 2007; Dilek and Furnes, 2009; Guilmette et al., 2018; Ishikawa et al., 2002; MacLeod et al., 2013; Metcalf and Shervais, 2008; Rollinson, 2009; Rollinson and Adetunji, 2015; Shafaii Moghadam et al., 2013; Stern and Bloomer, 1992). These findings have led to reinterpretation of many ophiolites as counterparts to difficult-to-access magmatic records of subduction initiation preserved in submarine forearcs and arc basement ('proto-arcs'; Stern et al., 2012; Stern and Bloomer, 1992; Whattam and Stern, 2011). The significance of the Semail ophiolite as a proto-arc analogue has recently been highlighted by tectonic modelling and metamorphic geochronological studies, indicating that the Semail outcrops may constitute a rare example of proto-arc crust produced by the tectonically-induced mode of subduction initiation, rather than by spontaneous gravitational collapse (Duretz et al., 2016; Guilmette et al., 2018; Stern and Gerya, 2017).

Though apparently dissimilar in their modes of onset, both the mid-Cretaceous Tethyan and Eocene West Pacific subduction initiation events appear to have formed vast swathes of proto-arc crust thousands of km along strike and hundreds of km wide, all within a few million years (Fig. 1; Arculus et al., 2015; Hickey-Vargas et al., 2018; Ishizuka et al., 2011; Meffre et al., 2012; Moghadam et al., 2010; Reagan et al., 2019; Shafaii Moghadam et al., 2013). As a potential remnant of this crust, the Semail ophiolite, exposed over ~20,000 km$^2$, presents an excellent opportunity to assess the bulk composition and lateral continuity of these swathes and their underlying mantle.

Differential uplift and erosion of the Semail ophiolite has exposed a 300 km long strip of volcanic rocks that dip 20–50˚ east along the north-eastern margin of the Oman mountains (Fig. 1). This simple structure, the excellent exposure, and the wealth of previous petrogenetic studies means that the distribution and approximate volume of rocks produced by each volcanic episode should be evident in map view.

Four regionally-extensive, mappable volcanic units have been recognised in the Semail ophiolite (Geotimes, Lasail, Alley and Boninitic Alley), with an increasingly-pronounced subduction influence recorded by each successively erupted unit (Alabaster et al., 1982; Belgrano and Diamond, 2019; Gilgen et al., 2014; Ishikawa et al., 2002; Kusano et al., 2012, 2014, 2017). However, as previous regional mapping was carried out prior to the recognition of boninites, and the Lasail unit was either lumped in with Geotimes (e.g., BME, 1987a; Umino et al., 1990), or Alley (e.g., BRGM, 1993a), these existing regional maps distinguish only two regionally-distributed units: the early (V1) and late (V2) extrusives. Updating the geological maps of the northern ophiolite to the four-unit volcanostratigraphy is the aim of the present study.

This update is necessary to incorporate the findings of numerous detailed investigations of isolated volcanic sections (e.g., Einaudi et al., 2003; Godard et al., 2006; Kusano et al., 2014, 2012). Variations in volcanostratigraphy can then be used to test previously proposed along-strike variations in subduction-zone influence (e.g., Goodenough et al., 2010; Python et al., 2008) and hydrothermal processes (Alabaster et al., 1982; Gilgen et al., 2016; MacLeod and Rothery, 1992), as well as the overall

significance of the 'Phase 2' magmatic overprint, which is debated (de Graaff et al., 2019). Furthermore, a map differentiating the ophiolite's boninites facilitates exploration for the gold-bearing volcanogenic massive sulphide (VMS) ore deposits hosted by these lavas (Gilgen et al., 2014). Finally, a map perspective of the Semail upper crust provides an areally extensive and well-studied comparison to drilled sections from the Izu-Bonin-Mariana (IBM) proto-arc basement (Belgrano and Diamond, 2019; Ishikawa et al., 2002; MacLeod et al., 2013; Pearce et al., 1992; Reagan et al., 2013, 2017; Shervais et al., 2019; Stern

et al., 2012).

In the following we review the previous work on the Semail volcanic units and document the combined field, geochemical and aeromagnetic approach used to distinguish them during mapping. We then present a new map of the northern Semail volcanics, displaying both outcrop and sediment-covered bedrock occurrences, split for the first time into the major basal, arc-like, and boninitic volcanic groups. This will serve as a renewed basis for volcanological studies and for ore exploration in the

Semail ophiolite. The volcanostratigraphic implications of the new map are then discussed and briefly compared to the IBM proto-arc record.

## 1.1. Approach

Improvement of the excellent pre-existing regional maps required a multidisciplinary approach, which is summarised below:

1) Previously published geological maps (listed in Sect. 6) and locations of geochemically analysed lava samples were

georeferenced, digitised, and a correspondence was established between their volcanostratigraphies and that employed here (Table 1).

2) Field mapping was conducted between 2014 and 2019 by teams of 2–4 geologists during 6 field campaigns each of one month duration. We collected 190 new lava samples at strategic locations and analysed their elemental compositions. This

allowed us to assign each sample to a volcanostratigraphic unit by comparison with compositional fields defined by previously-published, stratigraphically-confirmed analyses. These assignments in turn allowed the field discrimination criteria for the lava units to be iteratively refined during each campaign.

3) The bulk magnetic and magnetic–mineralogical properties of a subset of samples from each unit were analysed to aid

interpretation of the regional aeromagnetic survey of the volcanic sequence (Isles and Witham, 1993). This interpretation allowed us to infer which volcanostratigraphic units are buried beneath the supra-ophiolitic sedimentary cover.

4) We integrated these datasets into a geological map covering ~950 km$^2$ of exposed upper-ophiolitic crust and showing a further ~1200 km$^2$ of buried volcanic bedrock.

## 1.2. Nomenclature of volcanostratigraphic units

Different nomenclatures for the volcanostratigraphy have been employed by different research groups (Table 1). We use the names assigned by the seminal studies of the volcanic sequence (Alabaster et al., 1980, 1982), i.e., *Geotimes*, *Lasail* and *Alley*, with the addition of *Boninitic Alley* from Gilgen et al. (2014). As well as being easy to distinguish, these names carry no connotations of chronological order, and hence they remain practical when mapping intercalated sequences. For clarity in the present article, when referring to previous work we append to our names the term used in the cited study (e.g. Geotimes/LV1). Notwithstanding these differences in nomenclature and some disagreement regarding petrogenetic groupings, these unit divisions are clearly defined by field observations and are widely accepted (Alabaster et al., 1982; Gilgen et al., 2014; Godard et al., 2006; Kusano et al., 2017; MacLeod et al., 2013; Umino et al., 1990).

## 1.3. Geology and petrogenesis of the Semail upper crustal units

The upper crust of the Semail ophiolite consists of a continuous sheeted dyke complex (SDC) 1–1.5 km thick which is conformably overlain by a 1–3 km thick sequence of extrusives (Lippard et al., 1986). Though the Semail magmatic history can be broadly split into two magmatic phases (Phase 1 and 2; Goodenough et al. 2014), geochemical indicators of subduction-zone input gradually increase upwards through the volcanic stratigraphy (A'Shaikh et al., 2005; Alabaster et al., 1982; Belgrano and Diamond, 2019; Ernewein et al., 1988; Godard et al., 2003; Kusano et al., 2012, 2014, 2017; Lippard et al., 1986; Umino et al., 1990). This progression is manifested by the more-or-less sequential eruption of four regionally-distributed and mappable volcanic units (Fig. 2):

1) *Geotimes*: This unit comprises basal basalts and basaltic andesites produced during the axial spreading stage of ophiolite formation (Phase 1). The Geotimes lavas were fed and contemporaneously emplaced with the comagmatic SDC predominantly between 96.5 Ma and 95.5 Ma (Alabaster et al., 1982; Rioux et al., 2013; Umino et al., 1990). The petrogenetic affinity of Geotimes is disputed between mid-ocean ridge basalts (MORB), back-arc basin basalts and forearc basalts (Alabaster et al., 1982; Ernewein et al., 1988; Godard et al., 2006; Kusano et al., 2012; MacLeod et al., 2013). In comparison to MORB, the geochemical composition of Geotimes is marked by negative Nb–Ta anomalies and by major-element fractionation trends indicative of elevated water contents (Alabaster et al., 1982; MacLeod et al., 2013).

2) *Lasail*: This is dominantly a low-Ti, primitive basaltic unit intercalated with, but mostly overlying, the Geotimes unit. It occurs as discontinuous off-axis accumulations (Alabaster et al., 1982; Belgrano and Diamond, 2019; Kusano et al., 2012). The basal intercalations with Geotimes demonstrate a clear stratigraphic association between Lasail and Geotimes/V1 (Belgrano and Diamond, 2019; Kusano et al., 2012; Umino et al., 1990). However, Lasail is closer to Alley/V2 in terms of

incompatible trace element depletion (Alabaster et al., 1982; Belgrano and Diamond, 2019; Ernewein et al., 1988; Gilgen et al., 2014). This mismatch has led to varying interpretations as to whether the Lasail unit belongs to Phase 1 or 2. Lasail's rare-earth element depletion can be modelled by elevated (~30 wt.%) partial melting of the same depleted-MORB mantle source that produced Geotimes (Godard et al., 2006). Such high-degree melting fits with the distinctly primitive composition of the Lasail lavas. Lasail's elevated Th/Nb ratios indicate that this melting was assisted by additions of a high-temperature subduction component (Alabaster et al., 1982; Belgrano and Diamond, 2019; Gilgen et al., 2014; Godard et al., 2006; Pearce, 2008; Shervais, 1982). Thus, high-degree melting of a supra-subduction MORB-mantle source, triggered by localised infiltration of a slab-derived fluid, succinctly explains Lasail's geochemical features as well as its differences with respect to the overlying Alley lavas (Belgrano and Diamond, 2019).

3) *Tholeiitic Alley*: This is part of the original 'Alley' volcanic group formalised by Smewing et al. (1977) and Alabaster et al. (1982). The group includes tholeiite-series lavas that we differentiate here from their boninitic counterparts. The Tholeiitic Alley unit spans a fractionation series from basalt through to high-magnesium andesite to rhyolite, and it is generally accepted as belonging to Phase 2 of the Semail magmatic history (Alabaster et al., 1982; Gilgen et al., 2014; Kusano et al., 2014, 2017; Umino et al., 1990). The moderate incompatible-element depletion and fluid-mobile element enrichment of Tholeiitic Alley/LV2 lavas and glasses are interpreted as the result of flux melting of the axial mantle source residue by a slab-derived hydrous fluid (Alabaster et al., 1982; Kusano et al., 2014, 2017).

4) *Boninitic Alley*: High-Ca boninitic lavas were first discovered at the top of the stratigraphy in the Wadi Jizi area (Ishikawa et al., 2002). Kusano et al. (2014) then showed that boninites occur as a ~140 m thick package (termed UV2) overlying and intercalated with Tholeiitic Alley/LV2 lavas in Wadi Bidi (Hilti block). Regional sampling by Gilgen et al. (2014) and Kusano et al. (2017) further showed that boninites occur at numerous locations throughout the northern ophiolite. However, the volume and continuity of the boninites was unknown prior to this study. While Boninitic Alley and Tholeiitic Alley have previously been grouped on the basis of their appearance and stratigraphic intercalations, their source composition and geochemical fractionation trends are clearly distinct from one another (Gilgen et al., 2014; Kusano et al., 2014), thereby justifying their definition as distinct volcanostratigraphic units.

Other locally-distributed volcanic units were also described by Alabaster et al. (1982). The *Clinopyroxene-phyric* unit occurs locally around Wadi Jizi, and is compositionally intermediate between Tholeiitic and Boninitic Alley (Gilgen et al., 2014). Owing to its small extent and intermediate nature, we have not included it in our map. The mildly alkaline, obduction-related *Salahi* volcanics (V3; Alabaster et al., 1982) have so far been found only in the Wadi Salahi area (Hilti block), where their extent is well-defined and where they are separated from the underlying Alley group volcanics by either an eroded base or by several metres of pelagic sediments (BME, 1987b; Kusano et al., 2014; Umino, 2012). Our map displays the previously known extent of the Salahi lavas without modification.

### 1.4. Previous works and scope of new map

#### 1.4.1. Previous geological maps

The first regional-scale geological map of the Semail ophiolite with a differentiated Lower and Upper extrusive sequence was compiled by Lippard et al. (1986) at a scale of 1:250,000. This work was expanded upon by the 1:50,000 and 1:100,000 maps produced by Japanese and French teams together with the Ministry of Petroleum and Minerals (BME, 1987c, 1987b, 1987a; BRGM, 1986b, 1986a, 1993a, 1993b). This collection covers our entire mapping area and is referred to hereafter as the 'regional map set'. The regional map set displays the distribution of an essentially two-unit, lower and an upper extrusive stratigraphy (Semail Extrusive 1 and 2, respectively), with the addition of the Salahi unit (Semail Extrusive 3) as well as dyke-cut and felsic subunits. The different regional maps generally agree well along their margins, with the exception of a mismatch at the gabbro–SDC contact between the Fizh (BRGM, 1993b) and Wadi Bani Umar (BME, 1987c) map sheets. Here, we used the more recent field map of Adachi and Miyashita (2003) to correct the contact trace.

The map presented in this study primarily builds on this regional map set. In addition, numerous local maps have since been published together with articles and reports on the extrusives. In particular, the detailed field maps around VMS prospects published by JICA (2002, 2000) follow a similar stratigraphic scheme to the one used in the present study, and those maps provided useful constraints for our map in the Sarami, Haylayn and Yanqul areas.

The outcrop outlines on the existing regional map set are commonly offset from their true geodetic positions by 50–100 m. To remedy this, we retraced the outcrop outlines of the sheeted dyke complex and extrusives using 1 m resolution OrbView-3 satellite images made available by the United States Geological Survey through the Earth Explorer platform. We verified that the registration accuracy of these images is within 20 m by comparison with GPS positions recorded at outcrops and road intersections.

#### 1.4.2. Aeromagnetic map

In addition to the above geological maps, we used the Batinah aeromagnetic survey performed in 1992 by World Geoscience Cooperation (Isles and Witham, 1993) and made available by the Oman Public Authority for Mining, Muscat. The survey was flown with a line spacing of 200 m at a mean terrain clearance of 80 m. This corresponds to an approximate resolution of 80 m, where magnetic bodies that are less than 80 m wide or that are separated by less than 80 m will be detected, but not necessarily resolved (Flint et al., 1999). The acquired data were processed by reduction-to-pole (RTP) transformation prior to distribution, but details of this transformation are unfortunately unavailable. Such RTP transformations are performed to compute the anomalies that would be generated by the same source bodies in a vertically-orientated magnetic field, thus better situating anomalies over their sources and facilitating geological interpretation (Blakely, 1995; Clark, 2014).

### 1.4.3. Scope of new map

The present study focuses exclusively on the northern ophiolite, where the vast majority of the ophiolite's volcanic rocks occur (Fig. 1). Volcanic rocks also occur outside of this mapped area: as a continuation of the Aswad block into the U.A.E., as discontinuous outcrops along the western flank of the northern ophiolite, and in the southeastern Semail and Tayin blocks. These occurrences fell beyond the scope of this study but they would be amenable to mapping using the methodology outlined here.

As our mapping was conducted on GPS-guided digital tablets and the coverage of pre-existing map information was variable, the detail achieved in our map varies somewhat over the mapping area and cannot be defined by a traditional scale. In the northern blocks, the detail of mapping is approximately equivalent to the 1:50,000 scale regional base maps (e.g., BME, 1987). However, in the Sarami and Haylayn blocks the coarser scale of the base maps (1:100,000; BRGM, 1986b, 1986a) is inherited to some extent, but improved to approximately 1:50,000 equivalence in areas where we focussed our field mapping, and it was generally improved by re-digitisation with modern satellite imagery. Use of the digital map file provided in the supplement to this article allows a wide range of scales to be displayed.

To provide geological context below the volcanic sequence, the sheeted dyke complex (SDC) and upper crustal intrusions (above the base of the SDC) were traced from the regional map set and where necessary their outlines were adapted to match new observations and satellite imagery. Below the SDC, the lower crustal units were grouped and simplified based on the regional map set, as was the mantle section. Selected dyke swarms and umbers at the unit contacts are newly drawn or reproduced on our map, but we have omitted many other small-scale umbers, dykes, veins and geographical features that are shown in the regional map set. Accordingly, we recommend using the new map in tandem with the existing regional map set.

The top of the volcanic sequence in our new map corresponds to either the true stratigraphic top, which is conformably overlain by pelagic sediments of the Suhaylah Formation (Robertson and Woodcock, 1983b), or to an eroded surface that is overlain by post-Suhaylah sediments, or to a faulted contact with overthrust sheets of the Batinah complex (Woodcock and Robertson, 1982). To mark the top of the volcanic sequence where it crops out, we adapted the outlines of the Suhaylah Formation and the volcanic conglomerates of the Zabyat Formation (Robertson and Woodcock, 1983b) from the regional map set. All other supra-ophiolitic sediments (e.g. olistostrome, gravels) that appear on previous maps are undifferentiated in our map.

Where permitted by the aeromagnetic survey, we inferred the occurrence of the upper crustal ophiolitic units, together with the volcaniclastic Zabyat Formation, under all other post-volcanic sedimentary cover (Suhaylah sediments, olistostrome, gravels). The areas of the map showing such inferences are to be viewed as a best-estimate of the identity of the bedrock units at the upper surface of the Semail igneous sequence. No attempt was made to infer the presence of volcanic units beneath surficial fault traces.

## 2.  Analytical Methods

Major-element compositions of whole-rock samples were determined by X-ray fluorescence (XRF) at ETH Zurich using the same method as Gilgen et al. (2016, 2014). Electron microprobe (EMP) analyses were conducted on a Jeol™ JXA-8200 EMP at the University of Bern using the same standardisation described in Belgrano & Diamond (2019). The major element compositions of volcanic glass samples were determined by EMP with a beam voltage of 15 kV, a beam current of 10 nA, and a beam diameter of 10 µm. This low current density (0.13 nA/µm$^2$) was chosen to minimize Na loss during measurement (Morgan and London, 2005), and current across the specimen was monitored as stable during measurement, indicating that minimal diffusion of Na occurred. The compositions of igneous clinopyroxenes were determined by EMP with a beam voltage of 20 kV, a beam current of 15 nA, and a beam diameter of 3 µm.

Trace-element analyses were conducted on a subset of samples from each unit using the pressed-powder-pellet laser-ablation inductively-coupled plasma spectrometry (PPP-LA-ICP-MS) method described by Peters and Pettke (2016). This was performed using a GeoLas-Pro 193 nm ArF Excimer™ laser system in combination with an ELAN DRC-e™ quadrupole mass spectrometer at the University of Bern, with USGS basalt glass GSD-1G as the primary calibration standard (reference values from Jochum et al., 2005). All major elements were also analysed, allowing the anhydrous trace-element composition to be calculated by closure to 100 wt.% oxides and trace elements. Accuracy was monitored through measurements of basalt standard BRP-1 (Cotta and Enzweiler, 2008) and highly-depleted komatiite standard OKUM (Kane et al., 2007) prepared using an identical method to the samples.

Bulk magnetic susceptibility was determined by two methods: a handheld Exploranium™ KT-5 kappameter and a desktop Magnon™ kappameter at the Institute for Rock Magnetism (IRM), University of Minnesota. The KT-5 measurements were made on flat-sawn hand sample surfaces. The Magnon™ measurements were made on 25 mm diameter rock cores 20–30 mm in length. The comparability of the two methods is indicated by a good correlation (1:0.97, $R^2 = 0.97$) between measurements of the same samples (Fig. S1). Natural remanent magnetization (NRM) was determined on the same rock cores on a 2G Enterprises 760 RF™ SQUID superconducting rock magnetometer at the IRM.

High-temperature magnetic susceptibility was measured on rock powders (0.2–0.5 g) in air with an AGICO KappaBridge™ Magnetic KLY-2 susceptometer operating at a frequency of 920 Hz from room temperature to 700 ˚C and back to room temperature in steps of ~3 ˚C. Initial attempts in an argon atmosphere resulted in significant artificial magnetite production during the experiment. Low-temperature measurements were conducted on 0.2–0.5 g rock chips using a Quantum Designs Magnetic Properties Measurement System (MPMS) including saturation magnetization as a function of temperature (field 2.5 T), field-cooled (FC) and zero-field-cooled (ZFC) remanences, and temperature cycling of a room temperature isothermal remanent magnetization (RT-IRM) down to 20 K and back to room temperature.

## 3.  Field descriptions

The field characteristics of the Semail volcanic units were recently reviewed by Gilgen et al., (2014). Here we summarise and expand on the key field features which aid in unit discrimination, adding our own observations of potentially misleading exceptions and complicating features.

### 3.1. Geotimes

The Geotimes unit is typically made up of monotonous sequences of basaltic to basaltic-andesitic pillow lavas (Fig. 3a) with occasional columnar-jointed massive flows up to 20 m thick and pillow breccias. Pillowed outcrops weather to hematitic-red to dark brownish-grey colours and tend to be more topographically prominent than the other volcanic units. Magnetism, as tested with a small field magnet, is generally strong for Geotimes lavas. Geotimes pillows are usually weakly vesicular and aphyric. Where present, vesicles are mostly filled with the greenschist-facies minerals chlorite, epidote or quartz. Exceptions
to this typical appearance include occasional pore-filling sub-greenschist alteration mineralogy (e.g. celadonite and zeolite) in the area a few km south of Wadi Fizh (sample TB3-11F), and in the up-faulted block northeast of the Lasail mine (e.g. TB3-07A2). Geotimes lavas in the Aswad block also commonly feature millimetric, subequant phenocrysts of relatively fresh plagioclase. Evolved, andesitic Geotimes lavas tend to form large, rusty brown, flat pillows and lobate flows, and are particularly common around Wadi Jizi and the Yanqul area.

### 3.2. Lasail

Lasail lavas typically consist of small, bun-shaped pillows and pahoehoe flows, often with more irregular shapes than is typical for Geotimes or Tholeiitic Alley (Fig 3B). These pillows may be interspersed with occasional massive flows up to 10 m thick. Pillowed outcrops are typically pale pastel-green to light grey in colour. Lasail lavas are either weakly- or non-magnetic when tested with a small field magnet. White variolites a few mm across tend to concentrate in the pillow rims. Lasail pillows are
usually only weakly vesicular, with chlorite, epidote and quartz fillings. Slightly more evolved Lasail lavas and those transitional with Geotimes may take on a darker appearance similar to Geotimes (e.g. Wadis Fizh and Ashar; Belgrano & Diamond, 2019; Kusano et al., 2012). In this case, field discrimination is challenging. Andesites and dacites purportedly belonging to the Lasail unit appear to be limited to the Wadi Jizi area (Alabaster et al., 1982). However, our observations in the area indicate that these evolved lavas always overlie the Lasail basalts, and the associated dyke sheets and plutonic complex
cut the overlying Tholeiitic Alley lavas. The evolved 'Lasail'-affinity lavas and sheets around Wadi Jizi are therefore probably related to Alley-stage volcanism, consistent with other large areas of evolved lavas elsewhere in the ophiolite.

### 3.3. Tholeiitic Alley

The Tholeiitic Alley unit is composed of mixed sequences of basaltic to high-magnesium andesitic pillow lavas (Fig. 3c,d) interspersed with occasional high-magnesium andesite–dacite columnar-jointed massive flows. These sequences are locally
intercalated with lenses of hyaloclastite breccias up to tens of meters thick, which occasionally preserve fragments of volcanic glass in a palagonitised matrix (Kusano et al., 2017). Basaltic to andesitic Tholeiitic Alley lavas are typically strongly attracted

to small field magnets. Tholeiitic Alley pillows often have ~1–10 mm diameter, dark-grey spots or spherules spaced evenly throughout their cross-sections (Fig. 3c, d). The alteration mineralogy is dominantly of zeolite and pumpellyite-prehnite facies, however, minor chlorite can commonly be observed in thin-section (Alabaster and Pearce, 1985; Pflumio, 1991). Celadonite alteration pervading the rock matrix is common, especially in massive flows, and contrasts with the rare, mainly pore-filling celadonite found in the Geotimes unit. Irregularly shaped fingers, pods and tabular zones of greenish-yellow pumpellyite alteration (Pflumio, 1991) are common in both Tholeiitic and Boninitic Alley, but rare in Geotimes (where epidotisation dominates; Gilgen et al., 2016). Pillow flows are typically pervasively vesicular, ranging from millimetric vesicles in pillows (Fig. 3c, d) to fist-sized cavities in andesite flows. These vesicles are commonly filled by quartz, chalcedony, celadonite, zeolite and late calcite, giving outcrops a characteristic white-spotted appearance. Primitive Tholeiitic Alley lavas take on a pale greyish-green appearance (Fig. 3d), which may have been confused with Lasail lavas around the Lasail mine in the past (as shown by Gilgen et al., 2014). In this case, Tholeiitic Alley can be distinguished by its spotted, dark grey spherules (and by its clinopyroxene compositions; Gilgen et al., 2014). Thick accumulations of andesitic to dacitic Alley lavas are common in the Haylayn block as well as in the Wadi Jizi area west of the Lasail mine. Where continuous enough to form mappable units, we differentiate these as *Felsic Alley*: a sub-unit of Tholeiitic Alley. These lavas are generally composed of strongly celadonitic or hematitic, columnar-jointed andesite to dacite massive flows and local rhyolite flows with relict pods of unaltered obsidian (Alabaster et al., 1982).

### 3.4. Boninitic Alley

Boninite lavas in the Semail ophiolite are composed of mixed sequences of pillow, pahoehoe, and massive flows with local accumulations of hyaloclastite breccias up to tens of meters thick. The pillows are of various sizes and shapes even in the same outcrop, and they fit snugly together, indicating that they formed from low viscosity lavas (Fig. 3e). Thick (~150 m) sequences of blocky breccias interspersed with boninite lava flows and calcareous sediments locally top the sequence in the Wadi Jizi area sampled by Ishikawa et al., (2002) and along the section between Wadi Hayl and Wadi Bani Umar (Fizh block). The Semail boninites commonly have abundant macroscopic olivine phenocrysts or their pseudomorphs preserved in their flow rims. They also tend to be highly vesicular (Fig. 3f), even with micro- or macroscopically foamy textures, though weakly vesicular examples also exist (Fig. 3e). Boninitic Alley lavas are typically only weakly attracted to small field magnets where they are darker and fresher. Depending on their thickness and thus burial depth, which ranges from 0–2 km, the metamorphic grade and secondary alteration mineralogy of Boninitic Alley varies between greenschist-, prehnite–pumpellyite and brownstone-facies (i.e., clay-rich; Alabaster and Pearce, 1985). The outcrop appearance of the boninites strongly depends on this alteration grade, with brownstone-altered boninites generally being indistinguishable from Tholeiitic Alley equivalents, having weathered to a mixture of dull greys, browns and greens reflecting various clays (e.g. Wadi Jizi, Wadi Bidi). At prehnite-pumpellyite facies, the boninites take on a light-brown colour (Fig. 3e). In areas of thicker boninite accumulations (e.g. Rajmi), greenschist-facies alteration generally transforms the boninites to pale green, reflecting abundant chlorite and albite, similarly to Lasail (Fig. 3f). A key difference to Lasail is that these pale boninites generally have highly spheroidal

volcanic textures (Fig. 3g) with abundant white or grey spherules which increase in size and coalesce towards the pillow cores, presumably related to fluid saturation during solidification (Ballhaus et al., 2015). Excellent exposures of all of these alteration grades can be found in newly opened roadcuts along the Batinah Expressway between Liwa and Hatta.

### 3.5. Late dyke swarms and sills

Swarms of 'V2' dykes cutting the axial volcanic strata are described in several places throughout the ophiolite in the regional map set. These swarms grade from spaced sets of dykes to late, fully-sheeted complexes. In the Wadi Rajmi–Safwa area (Fizh block), boninite dyke swarms (samples TB5-22A and TB5-22B; also dyke samples from Adachi and Miyashita, 2003) feed thick accumulations of boninite lavas (Ishikawa et al., 2002). In the Yanqul area, late sheeted dyke swarms confirmed as Tholeiitic Alley by geochemistry (samples TB3-25H, TB4-20L) are several hundred meters wide and run parallel to thick accumulations of blocky Tholeiitic Alley breccias and their associated normal faults.

Late sills locally make up significant proportions of the upper crust. In the East Fizh block, numerous columnar-jointed Tholeiitic Alley sills (confirmed by samples TB2-43C, TB2-44D) up to 20 m thick intrude the Lasail lavas. Their character as sills rather than massive flows is indicated by their planar basal contacts, their symmetric upper and lower chilled margins, and their apophyses locally injecting the surrounding Lasail pillows.

### 3.6. Non-ophiolite rocks

Triassic volcanics of the Haybi complex crop out in the same area as Semail lavas in the Wadi Hawasinah and Harimah areas (Sarami, Haylayn blocks), where they are exposed in tectonic windows or exist on top of the ophiolite as olistostrome mega-clasts or as part of the overthrust Batinah complex (BRGM, 1986a, 1986b; Searle et al., 1980). Many of the Haybi tholeiites are pillowed and have comparable geochemistry to Geotimes lavas, whereas celadonite-altered alkali Haybi volcanics can resemble altered Alley lavas. Nevertheless, occurrences of the Haybi volcanics can generally be recognised because they are seldom continuous for more than a few hundred meters outside of the Hawasina window, they are always tectonically juxtaposed against the Semail volcanics, and they are often intercalated with limestones and cherts (Searle et al., 1980).

Quartz–hematite veins (labelled Q′ in the regional map set), often with magnetised hematite and distinctively chloritised alteration haloes up to tens of meters wide, frequently cut the extrusives throughout the mapping area. BRGM (1993a) suggested these veins are associated with the Phase 2 intrusives. In fact, these veins cross-cut the entire crustal sequence from the layered gabbros to the Boninitic Alley lavas (e.g. Aswad, Fizh blocks) with the same vein texture, mineralogy and chloritic alteration haloes. Any syn-magmatic veins emplaced throughout the vertical extent of the *in-situ* oceanic crust should show changes in texture and mineralogy as a function of their emplacement depth. Given the uniformity of these veins with depth, we conclude that they were emplaced after significant cooling of the ophiolite and probably following the warping of the ophiolite into its current anticlinal structure, i.e. during or after obduction and hence after the Phase 2 magmatic stage. Other, possibly-related, quartz–carbonate veins of similar dimensions but with indurated carbonation haloes are common as

topographically prominent lineaments in the easternmost outcrops of the Aswad block. Similar carbonate veins form listvenites where they intersect ultramafic intrusions near Wadi Shaffan (Haylayn block; BRGM, 1986b).

## 4. Geochemical assignment of samples

### 4.1. Samples

Samples of 179 spilitic lavas, 9 sills and dykes and 11 fresh volcanic glasses were collected throughout the mapping area for geochemical analysis to provide reference points for mapping. The lava samples were mostly collected from spilitic, pillowed lavas which were visually representative of the area, as well as occasional massive flows. Possible sills and dykes were avoided, except where deliberately sampled to determine the unit affiliation of such features (these samples are differentiated in our Figures).

Previously published lava analyses from useful locations (Gilgen et al., 2014, 2016; Kusano et al., 2017) were also considered and are differentiated as squares in our Figures. Though our interpretations mostly agree with these studies, we suggest a revised assignment for a minority of samples, which are asterisked in our Figures and listed in supplementary Table S2. MacLeod et al., (2013) also documented the major-element compositions of numerous V1 samples. With only major elements available for unit confirmation, we restricted this dataset to samples with >1 wt% $TiO_2$ and >0.1 wt% $P_2O_5$ (i.e., the

approximate lower limits of other reliably assigned Geotimes basalts).

### 4.2. Approach to identify units

Classical trace-element diagrams for the determination of tectonic setting serve as useful unit discriminants in the Semail ophiolite (Alabaster et al., 1982; Ernewein et al., 1988; Gilgen et al., 2014; Kusano et al., 2014, 2017). However, these diagrams are intended for characterising entire suites of lavas (Pearce, 2014), for which interpretation is relatively resistant to the

presence of outliers. Multiple diagrams based on independent principles are therefore required to best characterise individual samples.

  Alabaster et al., (1982), Umino et al., (1990) and later Gilgen et al. (2014, 2016) developed a geochemical workflow for identifying spilitised Semail lavas and dykes of unknown affiliation by comparing their immobile element and relict clinopyroxene compositions to those of lavas reliably assigned to a unit on the basis of field relationships. We firstly updated

these geochemical discrimination fields to include new stratigraphically-assigned datasets from sections along Wadi Shaffan (Geotimes/V1; Einaudi et al., 2003), Wadi Fizh (Geotimes/less-depleted LV1 and MV1 and Lasail/UV1; Kusano et al., 2012) and Wadi Bidi (Tholeiitic Alley/LV2 and Boninitic Alley/UV2; Kusano et al., 2014). We then compared our samples to these fields and the differentiation trends that define them. If a sample fell outside of the pre-defined fields or within an overlapping area in one plot, its position along trends in other plots was taken into consideration along with any field-based constraints to

best assign it to a unit. The pre-defined fields were then expanded to include the independantly-assigned sample, producing a

new set of slightly larger discriminatory fields. The possible assignments of samples indicated by each diagram are summarised in the supplementary Table S1.

## 4.3. Geochemical results

The Semail volcanostratigraphy records a progressive depletion in incompatible elements during the ophiolite's formation
(Fig. 4a, b; Alabaster et al., 1982; Ernewein et al., 1988; Kusano et al., 2017, 2014). Incompatible-element concentrations are also sensitive to magmatic fractionation and consequently they tend to overlap with their under- and overlying units (Fig. 4). Interestingly, Fig. 4 shows that Hf/Zr ratios tend to increase upwards through the volcanostratigraphy, which may prove instructive in narrowing down the nature of the slab component contaminating these lavas.

Comparisons of compatible elements (Cr), or somewhat-compatible elements (e.g. Ti, V) with incompatible elements (Zr,
Y) allow individual samples to be classified as potentially evolved or primitive members of each unit (Fig. 5). Firstly, the Zr–Ti diagram (Fig. 5a; Alabaster et al., 1982; Pearce et al., 1981) allows for the assessment of magmatic Ti-bearing magnetite fractionation, a process which must be ruled out before applying the Ti–V discriminant diagram (Fig. 5b; Shervais, 1982). Compositions falling beneath the main positive correlation trend in Fig. 5a testify to magnetite fractionation, and therefore they are assigned to felsic sub-units and excluded from Fig. 5b. The Zr–Ti diagram also shows whether such evolved lavas
belong to the Tholeiitic Alley or Geotimes suites, as Ti-bearing magnetite fractionation occurs at lower Zr concentration in Tholeiitic Alley relative to Geotimes (Alabaster et al., 1982; Pearce et al., 1981). Figure 5a further provides a simple discrimination of many of the Geotimes and Boninitic Alley samples on the basis of the absolute concentrations of their incompatible elements. The compositions of Phase 1 and 2 plagiogranites from the length of the ophiolite are also plotted for comparison in Fig. 5a (de Graaff et al., 2019; Haase et al., 2016). The Zr–Ti evolution of the Phase 1 intrusives closely matches
that of Felsic Geotimes, whereas most of the Phase 2 compositions overlap with Felsic Alley. A subset of these Phase 2 compositions depart the linear Zr–Ti trend at the upper limit of the Boninitic Alley array, at lower Zr concentration than for Tholeiitic Alley.

The Ti–V diagram (Fig. 5b) is particularly effective at discriminating units within the Semail extrusive suite and within ophiolites in general (Gilgen et al., 2014, 2016; Pearce, 2014; Shervais, 1982). This discrimination is partly based on the redox-
sensitive compatibility of V relative to Ti in the mantle source: source oxidation decreases V compatibility and thus increases V/Ti ratios in its partial melts (Mallmann and O'Neill, 2009; Shervais, 1982). This combines with increasing V/Ti at higher partial melt degrees and Ti-depletion upwards through the Semail extrusive sequence to trace an anti-clockwise progression in Fig. 5b from Geotimes through to Boninitic Alley, with each unit falling along radiating V/Ti trends. Apart from the aforementioned issue with magnetite fraction (affected samples have been screened out with Fig. 5a), V appears to be slightly
mobile during epidosite and pumpellyite alteration (e.g. Gilgen et al., 2016). Consequently, the occasional samples affected by incipient alteration of these types have been excluded from Fig. 5b.

The Cr–Y diagram (Fig. 5c; Pearce, 1980) allows incompatible-element depletion (monitored by Y concentration) to be considered as a function of magmatic differentiation (monitored by Cr depletion). Accordingly, it is particularly useful in

distinguishing primitive Geotimes from Lasail samples which fall in the overlapping area of Fig. 5b. For these samples, the incompatible-element depletion that is diagnostic of Lasail (Alabaster et al., 1982; Godard et al., 2006; Kusano et al., 2012) allows for mostly unambiguous sample assignment to either a Lasail or Geotimes fractionation path at respectively lower or higher Y content for a given Cr content (Fig. 5c). As chromium is rapidly depleted during differentiation of wet melts (Pearce,

1980), its concentration lies below the XRF-detection limit (~4 µg/g) for much of our sample set. Concentrations below the detection limit are not plotted in Fig. 5c and therefore the Alley and Geotimes fields in fact extend to lower Cr and higher Y contents than those indicated by the coloured fields.

Following our addition of many recent analyses, the unit fields of Alabaster et al. (1982) and Gilgen et al. (2014) now mostly overlap in the Zr/Y–Zr diagram. The exceptions are for particularly-depleted Boninitic Alley lavas and particularly-

evolved Geotimes lavas (Fig. 5d). Boninitic Alley and Depleted Lasail also have characteristically steeper trends than the other units in this diagram.

## 4.4.   Geochemical discrimination of Lasail from Tholeiitic Alley

The expanded dataset of analyses now available for the Semail volcanics show that the Lasail and Tholeiitic Alley units cannot be straightforwardly discriminated at the regional scale based on whole-rock geochemistry alone. In Fig 5a–d, for example,

the fields for Lasail and Tholeiitic Alley overlap to a large extent. Godard et al. (2003) speculated that their units V2-I and V2-II may be equivalent to Lasail and Alley, respectively, and they distinguished them based on their contrasting incompatible-element patterns and related ratios. However, the differences between Lasail and Tholeiitic Alley are not apparent when the data of Kusano et al. (2014, 2012) are plotted in the Godard et al. (2003) diagrams (Belgrano and Diamond, 2019). In fact, V2-I rather appears to correspond to the Tholeiitic Alley unit, as suggested by the along-strike continuity of V2-I described by

Godard et al. (2003). Lavas directly comparable to V2-II were neither recovered during this study nor by Kusano et al. (2014). However, as noted by Kusano et al. (2014), the spoon-shaped incompatible-element patterns of V2-II fit those of an evolved Boninitic or Transitional Alley lava (Fig. 4c).

The only reliable way to geochemically discriminate Lasail and Tholeiitic Alley identified thus far is through their divergent clinopyroxene compositions (Fig. 5e, f; Alabaster et al., 1982; Belgrano and Diamond, 2019; Gilgen et al., 2016, 2014).

Fortunately, the outcrop appearance and stratigraphic associations of these two units are typically distinct from one another, so they could generally be discriminated during this study by combining field observations with their whole-rock XRF composition. In other cases, clinopyroxene analyses were necessary for assignment. In Fig. 5e, the median Mg# and Ti concentrations in each sample are plotted relative to the previously-published fields (Alabaster et al., 1982; Belgrano and Diamond, 2019; Gilgen et al., 2014, 2016). Compositional zonation and intra-sample variation among the clinopyroxenes are

also useful for discriminating between units whose medians fall in overlapping zones. Steep trends of dispersion at high Mg# are diagnostic of Lasail lavas (Gilgen et al., 2016). To show this in a reproducible and representative way, we plot the median of the upper and lower quintiles of Mg# with their corresponding Ti values for the totality of EMP measurements on each

sample and join them with a line. Core-to-rim zonation is indicated by an arrow where these trends were petrographically clear and consistent.

### 4.5. Interpretation of transitional compositions

The rather large fields of the four main volcanic units in Fig. 5 reflect the true range of compositions within each unit, as these fields encompass samples taken from clearly stratigraphically-defined positions. However, where samples with intermediate geochemistry coincide with an intermediate stratigraphic position, i.e., where they outcrop between two clearly identifiable units, these samples are probably transitional. Such lavas are important as they demonstrate temporal overlaps between eruptive episodes. Transitional lavas between Geotimes and Lasail have been widely reported (A'Shaikh et al., 2005; Alabaster et al., 1982; Belgrano and Diamond, 2019; Kusano et al., 2012). A subset of our samples is also geochemically and stratigraphically intermediate between Geotimes and Lasail and thus assigned to a Transitional Geotimes–Lasail group. Similarly, samples with geochemistry intermediate between Tholeiitic and Boninitic Alley occasionally occur at the top of Tholeiitic Alley and in areas where both Tholeiitic Alley and Boninitic Alley are present, as also reported by Kusano et al. (2014). We assign these intermediate samples to a 'Transitional Alley' group. Transitional lavas between Geotimes and Alley have so far not been described. However, the lowermost Tholeiitic Alley lavas overlying Lasail and Geotimes in the northern Fizh block consistently have slightly lower V/Ti than typical Alley as well as less-depleted incompatible element patterns in Fig. 4 (e.g. samples YV15-21, TB2-41A, TB2-46A). Though marginally assigned to the Tholeiitic Alley unit on the basis of the literature fields, these lavas appear to represent an intermediate stage between Geotimes and Alley and can be traced in the RTP geomagnetic map into the footwall of the Mandoos VMS deposit. These transitional units also tend to have intermediate field characteristics; however, these differences are subtle and generally not mappable without prior knowledge and geochemical reference points.

### 4.6. Interpretation of Depleted Lasail compositions

Certain rare lavas exhibit Lasail-like clinopyroxene compositions but have highly-depleted, low-silica boninite whole-rock major element and immobile element compositions (Figs. 4a, 5, 6). No fresh glasses were recovered from these lavas, so the spilite compositions in Fig. 6 are potentially altered, however the reasonable correlation of Si and Mg in Fig. 6a suggests mostly limited mobilisation of these elements. Depleted Lasail's immobile element compositions are comparable to Boninitic Alley (Fig. 4a, 5), but with somewhat deeper negative Nb–Ta anomalies and an absence of spoon-shaped light rare-earth element enrichment (Fig. 4a). Gilgen et al. (2016) sampled similar lavas and their dyke equivalents (pillow lava sample RAM010 was remeasured with ICP-MS for this study), assigning them to Lasail on the basis of their clinopyroxene compositions. Belgrano and Diamond (2019) also reported similar lavas within a lens of 'Axial Lasail' lavas intercalated within Geotimes near Wadi Hawarim. In our Figures, these samples are distinguished from normal Lasail as 'Depleted Lasail'. The presence of Depleted Lasail lavas has been confirmed by sampling in the Hilti, Fizh and Aswad blocks. Stratigraphically, Depleted Lasail lavas occur within the axial Geotimes sequence near Wadi Hawarim (sample TB3-04B; Belgrano and

Diamond, 2019), directly overlying Geotimes and underlying Tholeiitic Alley near Wadi Bidi (sample TB5-27A), and atop a thick accumulation normal Lasail lavas just south of the Mandoos VMS deposit (samples RAM005, RAM010, TB2-40D). The observation of these lavas beneath the less-depleted Tholeiitic Alley sequence marks a departure from the otherwise consistent trend of depletion upwards through the volcanic stratigraphy. Further work is required to understand the significance of these early, depleted lavas. In the interim, the newly collected Depleted Lasail lavas are differentiated in our Figures but are not incorporated into the expanded geochemical fields for the normal Lasail unit.

## 4.7. How boninitic is the 'Boninitic Alley' unit?

Boninites are defined by IUGS (Le Bas, 2000) as volcanic rocks with whole-rock $SiO_2 > 52$ wt%, $MgO > 8$ wt% and $TiO_2 <$ 0.5 wt% after normalization to 100 wt%. The definition of boninite-series lavas has been incrementally expanded to include fractionation series which pass through the IUGS fields, extending from low-Si compositions to high-magnesium andesites (HMA) at < 8 wt% (Fig. 6; Pearce and Reagan, 2019; Pearce and Robinson, 2010; Reagan et al., 2017). Boninites can be further divided into low-Si boninite (LSB) and high-Si boninite (HSB) series, with a dividing composition along an olivine fractionation line which passes through $SiO_2 = 57$ wt% at $MgO = 8$ wt% (Pearce and Reagan, 2019).

To test whether the Semail Boninitic Alley unit is really 'boninitic', we consider the major element compositions of unaltered Boninitic Alley/UV2 volcanic glasses reported by Kusano et al. (2017), which fall along the same immobile-element trends as the Boninitic Alley spilites in Fig. 5. In addition, the major-element compositions of several volcanic glasses collected during this study are plotted for assignment into the boninite series or tholeiitic basalt–andesite–dacite–rhyolite (BADR) fields. Boninites from the West Pacific are also plotted for comparison with the Semail examples.

The Boninitic Alley glasses all fall within the LSB and HMA fields (Fig. 6). Tholeiitic Alley follows a fractionation path from basalt through the high-magnesium andesite field into dacite. The compositional differences between the two Alley units is slight in Fig. 6a, and are more clearly-defined by greater incompatible-element depletion (including Ti), higher V/Ti ratios and higher Cr in the boninites (Figs. 4b, 5b,c, 6b). Spilitised samples assigned to Boninitic Alley via Fig. 5 are also plotted for comparison in Fig. 6. Though these altered compositions should be treated with caution, the majority plot along similar trends to the fresh glasses, suggesting that the higher MgO content is not simply due to spilitization. The Boninitic Alley series may therefore extend to more magnesian compositions than those recorded by the glasses (e.g. to ~17 wt% MgO).

The trace element patterns of Boninitic Alley samples (as classified by Fig. 5) have highly-depleted incompatible element trends in Fig. 4b. Of these, seven tend towards monotonic, compatibility-controlled depletion with negative Nb–Ta and positive Th anomalies. The remaining sample (TB2-42A, Wadi Zab'in, Fizh block) has a classic boninite 'spoon'-shaped MORB-normalised pattern enriched in Th, Nb, Ta and light rare earth elements. A similar range in compositions was noted by Kusano et al. (2017), who subdivided Boninitic Alley/UV2 into 'low-Si' and 'high-Si' groups with monotonically-depleted and spoon-shaped MORB-normalised patterns, respectively (Fig. 4b). These differences were explained by the addition of sediment melts to the 'high-Si' group, supported by attendant shifts in Nd and Hf isotopic compositions (Kusano et al., 2017).

Taken together, the compositional similarities between the Boninitic Alley glasses, Boninitic Alley spilites and the West Pacific boninites in Figs. 4–6 indicate a low-Si boninite-series protolith for the Boninitic Alley spilites and demonstrate the effectiveness of the immobile-element fields in Fig. 5 for discriminating altered boninites.

## 5. Interpretation of the aeromagnetic survey

Aeromagnetic interpretation is greatly clarified by understanding the magnetic petrology of the surveyed units (Clark, 1997). To aid interpretation of the Batinah aeromagnetic map (Isles and Witham, 1993), we measured the magnetic susceptibility ($K$) and the natural remanent magnetization (NRM) intensity of samples that had been assigned to a specific volcanic unit by either a clear stratigraphic situation or by geochemical criteria. To identify the origin and establish the potential reliability of these magnetic properties, we further determined the magnetic mineralogy of a subset of these samples.

### 5.1. Bulk magnetic property results

Aeromagnetic anomalies (represented here by the Batinah RTP map) are principally caused by differences in magnetization between adjacent rock bodies. These differences are controlled by the different magnetic susceptibilities, i.e. induced magnetization, of the rocks, as well as by differences in NRM (Blakely, 1995; Clark, 1997, 2014). A basic assumption of the RTP transformation carried out on the Batinah survey is that source magnetization is parallel to the local geomagnetic field, which may not be the case for rock bodies with strong remanent magnetism (Blakely, 1995; Clark, 2014). The Koenigsberger ratio ($Q$) of remanent to induced magnetization is therefore a useful parameter for assessing the potential for artefacts or 'remanence effects' in the RTP map data used in this study (Clark, 1997). For rock bodies with $Q \ll 1$, induced magnetization dominates. Thus, in the absence of anisotropy of susceptibility, RTP processing accurately centres anomalies over their geographic sources. For bodies with $Q \gg 1$, magnetism is dominated by remanence. In this case, and if the NRM direction is different from the field direction, remanence effects may be introduced into the RTP map, leading to inaccurate estimates of the magnetization and geographic location of anomalies (Clark, 1997, 2014).

Figure 7 shows that Geotimes and Tholeiitic Alley lavas have generally higher susceptibilities and remanences than Lasail and Boninitic Alley lavas. Further, $Q$ is on average greater than one for all units, with Boninitic Alley having the largest spread in values. Although averaging NRM and $Q$ on a unit basis is properly achieved by incorporating the vector directions of these properties for each sample (Clark, 2014), the medians of our unoriented sample set give a sound idea of the *potential* for remanence effects in each unit (Fig. 7).

Comparison of these results with the chemical and mineralogical compositions of the lavas allows the origin of the magnetic properties to be identified (primary vs. secondary). This in turn permits assessment of the reliability of the aeromagnetic data as an aid in geological mapping. The magnetic properties of volcanic rocks are principally controlled by the partitioning of Fe between strongly magnetic (ferromagnetic) oxide phases and weakly magnetic (paramagnetic) silicate phases (Clark, 1997). Therefore, it is instructive to compare measured magnetic susceptibilities with the whole-rock proportion of Mg relative to Fe

(expressed as Mg# = molar Mg/(Mg + Fe$_{total}$)). Figure 8 shows this comparison, including the measurements of Geotimes lavas from Wadi Shaffan by Einaudi et al. (2003), which we have recalculated to mass-normalised susceptibility using the reported densities.

Between Mg# values of 45 and 80 (Fig. 8), which are typical of Geotimes and Tholeiitic Alley, susceptibility increases along a scattered array with decreasing Mg#. Above Mg# ~ 80, the primitive lavas of the Lasail and Boninitic Alley units mostly have susceptibilities close to zero. One particularly fresh, high Mg# boninite (TB3-01A) has low but significant susceptibility, continuing along the extension of the Geotimes–Alley trend. From this array we can infer that, for the Semail spilites, Fe is primarily incorporated into paramagnetic silicates in rocks with Mg# > 80, whereas in rocks with Mg# between 45 and 80 or in fresh lavas, Fe is significantly incorporated into ferromagnetic oxides.

At Mg# < 45 magnetic susceptibility is scattered towards lower values. This could be due to magmatic fractionation of titanomagnetite from evolved magmas (e.g. high-Si outliers in Fig. 8). However, apart from these two extreme samples, there is no clear link between low magnetic susceptibility and the felsic units shown to have fractionated magnetite in Fig. 5a. An alternative explanation is that for many of these low Mg# samples (e.g. from Einaudi et al., 2003), relatively intense hydrothermal alteration (e.g. incipient epidotisation) has either leached Mg or sequestered the available Fe into paramagnetic silicates or weakly magnetic oxides (e.g. epidote, hematite). Intense calcite alteration around late carbonate veins also appears to result in destruction of the magnetic minerals (TB3-15C; Fig. 8).

## 5.2. Magnetic mineralogical results

As shown above, the Semail lavas mostly display increasing magnetic susceptibility with decreasing Mg#, as would be expected for fresh lava suites (e.g., Vogt and Johnson, 1973), even though the rocks are pervasively hydrothermally altered. The ferromagnetic mineralogy of our samples elucidates how these apparently primary magnetic characteristics could have persisted through spilite alteration. This mineralogy was determined by high-temperature susceptibility (*KT)* and low-temperature magnetic experiments, for which four representative examples are given in Fig. 9. Plots for each sample, grouped by unit and mineralogical interpretation, are shown in supplementary Figs. S2–9.

Figure 9a shows a simple *KT* curve with magnetic unblocking upon heating corresponding with a d$K$/d$T$ minimum at 493 ˚C. This unblocking temperature ($T_{ub}$), is typical of titanomagnetite, with Ti-substitution depressing $T_{ub}$ from stoichiometric magnetite's Curie temperature of 580 ˚C (Dunlop and Özdemir, 1997). The weak $K$ upon cooling indicates that this titanomagnetite was oxidised to weakly magnetic (Ti-) hematite above 600 ˚C during the experiment. The curve in Fig. 9b is similar, except unblocking occurs rapidly at a $T_{ub}$ of ~574 ˚C, almost equivalent to the Curie temperature of stoichiometric magnetite.

The *KT* curves in Figures 9c and d are more complex, with an increase in $K$ upon heating from room temperature to ~350 ˚C punctuated with a step at 130–40 ˚C. In Fig. 9c, this is followed by a drop in $K$ at around 409 ˚C, but the majority of magnetization persists until unblocking at ~576 ˚C. The initial increase in $K$ from room temperature in Figs. 9c and d is a common feature of maghemite and has been explained as the thermally-prompted relaxation of lattice stresses at the contact

between maghemite rims and magnetite cores ('maghemite bump'; Kontny and Grothaus, 2017; Liu et al., 2004; Velzen and Zijderveld, 1992). Maghemite ($\gamma$-Fe$_2$O$_3$) is a typical oxidation product of magnetite which retains the cubic spinel structure and much of the magnetism of its precursor magnetite (Clark, 1997). We interpret the marked drops in $K$ at 300–450 ˚C (Figs. 9c, d) as the structural inversion of maghemite to hematite during heating (Dunlop and Özdemir, 1997).

The low-temperature magnetic behaviour of these samples supports our high-$T$ interpretations (Fig. 9e–h). The absence of a Verwey transition at ~120 K for TB3-01A (Fig 9e) is consistent with Ti-substitution in magnetite (Moskowitz et al., 1998). Contrastingly, a marked Verwey transition occurs at ~110 K in TB2-33C (Fig. 9f), supporting the interpretation of stoichiometric magnetite from Fig. 9b. For TB3-25E, a weak Verwey transition is detectable (Fig. 9g). This is consistent with the mixture of magnetite and maghemite deduced from the high-$T$ data, as the Verwey transition is supressed by partial

oxidation of magnetite and is absent in maghemite (Dunlop and Özdemir, 1997). For $KT$ curves featuring drops in $K$ at ~330 ˚C, similar to that in Fig. 9D, which we attribute to maghemite inversion, the presence of monoclinic pyrrhotite (with $T_{Curie}$ = 320 ˚C), is difficult to rule out. However, upon cooling through ~30–35 K, monoclinic pyrrhotite should undergo a characteristic 'Besnus' transition leading to a loss of remanence (Dunlop and Özdemir, 1997; Rochette et al., 1990). Such a transition was not detected for any of the Semail lavas, which strongly supports the interpretation that maghemite is present in

samples like TB3-07C (Figs. 9d, h).

A summary of ferromagnetic mineral occurrences deduced from the high and low-$T$ data (excluding weakly-magnetic hematite) is given for the entire sample set ($n = 38$) in Fig. 9i. Multiple ferromagnetic phases often occur within a single sample (e.g. Fig. 9c). Titanomagnetite (e.g. Figs. 9a, e) occurs in 37% of our samples but is particularly prevalent in Tholeiitic Alley (55%), in agreement with Perrin et al. (1994). The good positive correlation with Zr in Fig. 5a (prior to magnetite fractionation)

shows that Ti is rather immobile during spilitization. These titanomagnetites therefore appear to be primary magmatic phases. Almost stoichiometric magnetite, as evidenced by $T_{ub}$ = 570–580 ˚C (e.g., Figs. 9b, f), is present in 30–60% of samples from each unit. This magnetite could either be interpreted as hydrothermal, or as the Ti-poor phase of magmatic titanomagnetites which have unmixed during cooling (Dunlop and Özdemir, 1997). Maghemite 'bumps' and inversions comparable to those in Fig. 9D occur in 47% of the total sample set and in 86% of Geotimes samples. Given the often significant proportion of

magnetism lost during these inversions, we conclude, in agreement with Perrin et al., (1994), that maghemite is a prevalent carrier of magnetism in the Semail lavas, and in particular in the Geotimes unit.

## 5.3.  Implications for aeromagnetic interpretation

Figure 7 demonstrates that, typically, highly magnetised areas in the aeromagnetic map should correlate with the occurrence of Geotimes and Tholeiitic Alley lavas, whereas weakly magnetic zones should correlate with Lasail and Boninitic Alley.

Figure 8 demonstrates that this correlation is connected to the characteristically high Mg# of Lasail and Boninitic Alley in comparison to Geotimes and Tholeiitic Alley. As these differences are also clearly visible on a unit basis (Fig. 8), the high Mg# of Lasail and Boninitic Alley must be inherited from their protoliths, as supported by these units' high Cr, abundant olivine (phenocrysts or pseudomorphs; Kusano et al., 2014, 2012), and magnesian clinopyroxene compositions. The mixed,

variably-oxidised ferromagnetic mineralogy of the Semail lavas may explain much of the scatter in Fig. 8. However, though this scatter exists for each unit (Fig. 7), at the spatial resolution of the aeromagnetic survey (~80 m), the average character of each unit is more likely to be represented. The common occurrence of relict titanomagnetite in Alley, as well as oxidised, but nevertheless magnetic maghemite in Geotimes (Fig. 9), provides a mineralogical explanation for how these primary properties

could have been partly preserved through spilite alteration. These conclusions agree with the qualitative differences observed with field magnets (described in Sect. 3), and they attest to the usefulness of the relationships shown in Fig. 7 for aeromagnetic mapping.

However, Königsberger ratios greater than unity for the Semail extrusives (Fig. 7) indicate the potential for significant remanence effects on the RTP data (Clark, 1997; Flint et al., 1999). As the ophiolite was formed during a period of normal

geomagnetic polarity (Perrin et al., 1994), the extrusive sequence should not be complicated by naturally opposing primary remanence directions. This is supported by measurements from the northern extrusives, whose characteristic remanence directions can be reconciled with each other by tectonic rotations of < 90° (Perrin et al., 1994, 2000). Nevertheless, these minor block rotations, as well as any syn-volcanic tilting or later remanence resetting during obduction and rotation (Feinberg et al., 1999; Morris et al., 2016) could all lead to inconsistent RTP anomalies. We accordingly acknowledge the limitations of the

RTP map for remote mapping. To mitigate the influence of remanence effects on our interpretation, each structural block was considered separately in terms of the RTP character of each unit, and precedence was given to field-based constraints. In this fashion, aeromagnetic inferences could be made between reference points and under cover on a case-by-case basis.

## 5.4.  Observed reduced-to-pole anomalies

The alternation between strongly and weakly magnetic units upwards through the Semail volcanostratigraphy suggests that

uniformly dipping extrusive sections should be straightforward to interpret in aeromagnetic data. Good examples of this exist along the east-dipping Fizh and Hilti blocks. Nevertheless, over the mapping area, a variety of RTP anomaly characters are observed for each volcanic unit (Fig. 10)

The Geotimes unit typically corresponds to relatively positive but patchy RTP anomalies between 0.5 and 2 µT. This inconsistency makes Geotimes challenging to infer in many areas. Fortunately, Geotimes is usually well-exposed at the surface

and accurately delineated by the existing regional map set. A number of factors may explain this patchiness. Firstly, Geotimes has generally undergone higher grade spilitic alteration than the overlying units (Alabaster and Pearce, 1985), and it is locally altered to weakly-magnetic epidosite (Gilgen et al., 2016). Intercalated lenses of the weakly-magnetic Lasail unit within Geotimes are also occasionally present, and can explain some of the patchiness around wadis Hatta, Ashar, Fizh (Fizh block) and Ghuzayn (Alabaster et al., 1982; Belgrano and Diamond, 2019; Kusano et al., 2012; Umino et al., 1990). Commonly,

however, patches of weak RTP magnetism apparently wholly confined to the Geotimes unit do not correspond to weakly magnetic or anomalously altered lavas as confirmed in the field (e.g. along the Hilti block). One possible explanation for this is that shallow, weakly-magnetic intrusions exist beneath the volcanics. Another explanation, which better explains the

widespread distribution of this patchiness, is that magnetic remanence effects (as permitted by Königsberger ratios above unity; Fig. 7c), locally have a significant influence on the aeromagnetic anomalies related to the Geotimes unit.

Exposures of Lasail lavas greater than ~100 m wide (comparable to the spatial resolution of the aeromagnetic survey) consistently coincide with relatively weak RTP anomalies between 0 and 0.5 µT. The consistency of the Lasail unit's weak RTP anomalies is presumably due to the unit's consistently weaker magnetism compared to other units (Fig. 7a, b).

Tholeiitic Alley typically corresponds to consistent, stratiform positive anomalies between 1 and 2.5 µT, which are straightforward to interpret and to trace between reference points. There are some areas of Tholeiitic Alley lavas which correspond to weak RTP magnetism, but this is typically where Tholeiitic Alley is anomalously thin (e.g. Aswad block) or cut by major faults (e.g. Haylayn block).

Although the measured magnetism of the Boninitic Alley unit is rather weak (Fig. 7a, B), it has an inconsistent RTP character ranging from 0 to 2.5 µT. This can possibly be attributed to its highly variable grade of hydrothermal alteration. In the prominent accumulations of boninites in the Aswad, Rajmi and Daris areas, the sampled boninites are mostly altered to greenschist-facies and are weakly magnetic (Fig. 3d, 7). Contrastingly, a relatively fresh sample (TB3-01A; Fig. 9a) from the Eastern Fizh block retains significant magnetic susceptibility as well as its primary titanomagnetite (Fig. 9a). This sample, and presumably other fresh boninites are sited within an NNW–SSE oriented positive RTP anomaly which extends for ~30 km along-strike. This inconsistency makes the inference of boninites from aeromagnetic data alone challenging, and accordingly, we have confirmed all the mapped boninite accumulations by field observations or sampling.

### 5.4.1. The Batinah Complex

Inferences under cover at the top of the section in the Sarami and Haylayn blocks are complicated by the discontinuous presence of magnetic Triassic volcanics tectonically emplaced over this part of the ophiolite as part of the Batinah complex (BRGM, 1986b; Woodcock and Robertson, 1982). Given the shallow dip of the top of ophiolite in this area (Shelton, 1990), the magnetic anomalies situated over the Batinah complex volcanics could also be influenced by the underlying Semail volcanics. Where exposed, the faulted contact between the ophiolite and the Batinah complex is readily picked in the field and is well-defined by the regional maps (BRGM, 1986a, 1986b). However, northeast of this contact, or where it disappears under cover, there is some uncertainty in interpretation. To avoid confusion we have delineated our bedrock interpretation of the overthrust Batinah complex in Fig. 10, and we have not attempted to interpret the ophiolite beneath the faulted contact.

### 5.4.2. Repeated ophiolite blocks visible in the aeromagnetic map

A positive gravity anomaly suggests that a repeated ophiolite block exists underneath the Batinah plane to the northeast of the Sarami block (Shelton, 1990). Isolated outcrops of serpentinite and listvenite around the anomaly may be the surface expressions of this block, or may simply be large olistoliths (BRGM, 1986b). This repeated block is well-resolved as an arcuate, 2–2.5 µT positive RTP anomaly 30–40 km long and 5–8 km wide in Fig. 10 (Anomaly #3). The intensity and dimensions of the magnetic anomaly resemble those of the east-dipping Alley sequences in the Hilti and East Fizh blocks.

However, with limited outcrop available for confirmation, it is also feasible that the geophysical anomaly corresponds to a sheet of serpentinised, magnetite-bearing peridotite. Less-clearly resolved, but nevertheless positive magnetic anomalies also extend as swathes to the north (Anomaly #1) and south (Anomaly #2) of the exposed East Fizh block lavas. These anomalies indicate that the East Fizh repetition probably continues for some 80 km along-strike and is open to extension along the eastern coast of the UAE.

### 5.4.3. Block boundaries and fault zones in the aeromagnetic map

The aeromagnetic map reveals weak and even negative magnetic anomalies at the boundaries between all the mapped tectonic blocks (Fig. 10). The boundary zones between the Fizh, Hilti and Sarami blocks are complex and occur across multiple structures which isolate smaller blocks between the main blocks (e.g. between Wadi Bargah and Jizi, also south of Wadi Ahin). These boundary zones along Wadi Jizi, Wadi Ahin, and Wadi Bani Suq are occupied by sediment-dominated melange (Robertson and Woodcock, 1983a), which explains the weak RTP magnetism of these zones.

The aeromagnetic map also reveals strips of weak and slightly negative RTP anomalies up to kilometres wide coinciding with major fault zones within the blocks (Fig. 10). Where both sides of the fault consist of igneous rocks (e.g. Wadi Bargah and north and south of Wadi Hawqayn; Fig. 10) these anomalies must be explained by de- or re-magnetization around the fault planes. The approximate spatial extents of these tectonically-controlled weak aeromagnetic anomalies are marked in Fig. 10. The slightly negative RTP values within many of these anomalies cannot be achieved purely by magnetic mineral destruction. Rather, they imply that a secondary magnetic remanence that is not parallel to the geomagnetic field has been imparted to the surrounding volcanics. Fluid-related remagnetization of the lower crustal and mantle sections of the ophiolite during obduction and rotation has previously been documented throughout the ophiolite (Feinberg et al., 1999; Morris et al., 2016; Usui and Yamazaki, 2010). The structurally-controlled remagnetization outlined in Fig. 10 indicates that the fluids responsible for resetting magnetism deeper in the section may have locally transgressed into the upper crust. Field evidence for this hydrothermal resetting is compelling, as the idiomorphic hematite in the obduction-related quartz–hematite veins (Q') is often strongly magnetic (described in Sect. 3.6). As pure hematite is only very weakly magnetic, this magnetism presumably results from pseudomorhpic replacement of hematite by magnetite (mushketovisation) which post-dates the emplacement of the quartz–hematite veins.

### 6. Map construction and presentation

Previously published volcanostratigraphic sections and geological maps from 21 publications were used as a basis for mapping the Semail volcanic units. The suite of 190 lava samples that we assigned to volcanostratigraphic units by geochemistry were combined with 89 previously published sample locations to provide reference points for our mapping. To illustrate the confidence in which the various areas are mapped, Fig. 11 shows the locations of these previous maps, the 279 lava samples, 9 dyke or sill samples, and also the sites where we mapped the units using GPS-equipped digital tablets.

Our sampling and observation density is highest in the north of the mapping area. This is partly due to the wider, more tectonically complex sequence in the north, but also due to the availability of samples and observations collected during parallel projects in this area. Between these sampling points, the certainty of our unit identifications depends on the field and aeromagnetic characteristics of the units, on the local structural complexity, and on the coverage by previous maps. For example, our fewer reference samples in the southern blocks are somewhat offset by the excellent coverage of the JICA (2002, 2000) maps in those areas.

As Geotimes is mostly well defined in the regional map set and typically readily recognizable in the field, this unit is delineated with high confidence in our map. Similarly, the generally consistent field appearance, stratigraphic position and magnetic character of the Lasail unit allowed for relatively unambiguous mapping on the surface and inference of its presence below cover.

Collectively, the Alley group lavas were relatively straightforward to delineate on the basis of their distinctive field characteristics and generally positive RTP anomalies. Where boninites are thickly accumulated and altered to pale, greenschist-facies spilites at their basal contact with Tholeiitic Alley (e.g. Aswad, Fizh, Daris), this contact is easy to follow in the field and in satellite imagery. However, where Boninitic Alley occurs as a thin cap on the sequence (e.g. Hilti, Sarami blocks), it often proved difficult to distinguish from Tholeiitic Alley without geochemical analysis and careful field observations. It is therefore possible that further occurrences of Boninitic Alley may be present as thin cappings in addition to the mapped layers. Consequently, areas where boninites are suspected, but are untested by sampling, are marked on the map as 'Undifferentiated Boninitic & Tholeiitic Alley'.

Tabular intrusions intruded along faults, as well as epidote-cemented fault-breccias were taken as evidence for 'syn-magmatic' deformation, similarly to Reuber (1988). Unlike chlorite, epidote is not part of the overprinting Q' quartz–hematite vein assemblage, and pervasive epidote alteration in the upper crust occurred mostly during Tholeiitic Alley-phase magmatism (Gilgen et al., 2016).

The complete map of the volcanic units is presented in Fig. 12. For detailed use, the reader is referred to the Geospatial PDF provided in the supplement to this article. The Geospatial PDF format allows the georeferenced input and final map layers to be viewed in Adobe Acrobat™ and directly imported into common geospatial mapping software (e.g. Avenza MapPublisher™).

## 7. Discussion of newly mapped features

### 7.1. Tectonic omissions of Geotimes in the Haylayn block

The Geotimes unit is present above the SDC throughout most of the ophiolite (Fig. 12). However, in the Haylayn block (in the region of Wadi Wadiyah), Geotimes occurs only discontinuously and the volcanic section is almost invariably bounded above and below by faulting. These faults correspond to strip-like negative anomalies in the RTP map (Fig. 13). Along these faults, Tholeiitic Alley lavas are commonly juxtaposed against the SDC, and Geotimes is either absent or is only a few hundred

meters thick, as previously noted by Lippard et al. (1986). Rather than viewing the contact between Alley and the SDC as conformable, as previously mapped by BRGM (1986a) and Juteau et al. (1988), we attribute the missing Geotimes lavas in the Haylayn block to faulting (Fig. 13). This conclusion is supported by the negative RTP anomalies along this contact as well as our observation, ~500 m to the north of Wadi Wadiyah, of a fault zone several meters wide hosting a Q' quartz–hematite vein

5    marked by sub-horizontal slickensides. This zone occurs within the weak RTP anomaly and between outcrops of the SDC and Tholeiitic Alley (confirmed by sampling; Fig. 13). The tectonically-disturbed character of the contact between the SDC and the volcanics in the adjacent Sarami block is also demonstrated by a major fault zone several meters wide at the base of Geotimes in Wadi Shaffan (Einaudi et al., 2003). The presence of such faults precludes simple comparisons between the southern and northern blocks based on volcanostratigraphy.

## 7.2. Proportions of the upper-crustal units

Early descriptions of the Semail volcanic stratigraphy showed Geotimes with a relatively consistent thickness of 1–1.75 km, overlain by Lasail or Alley sequences ~0.75 km in thickness (Alabaster et al., 1982). In the Fizh block, Reuber (1988) contrastingly indicated that the 'V2' lavas could attain thicknesses of up to 1.5 km, whereas Geotimes can be as thin as 0.5

15    km. Our mapping indicates that complete, tectonically undisturbed volcanic sections between the SDC and the post-Alley Suhaylah formation are rare. The stratigraphic columns in Figure 12 are constructed from dip measurements along some of these complete sections (sections B, D, E) and along other instructive sections. These new sections are more consistent with Reuber (1988), indicating that undisturbed Geotimes can be as thin as 0.3 km, whereas Tholeiitic Alley is up to 1 km thick, and Boninitic Alley locally up to 2 km thick.

In addition to these stratigraphies, the inferred areas of buried bedrock in our map permit realistic estimates to made of the areal proportions of the different upper crustal units. Uniformly dipping but otherwise tectonically undisturbed sections yield the closest approximations of volumetric proportions from areal proportions. For this reason, the tectonically imbricated volcanics on the western side of the ophiolite are excluded from the area calculations in Table 2. The north-eastern side of the northern ophiolite (i.e., the Batinah Coast), dips relatively consistently to the east. From this strip we calculated the bedrock

areas of each upper crustal unit within the mapping area (Table 2). Due to tectonic complications in the Aswad, East Fizh, Sarami, and Haylayn blocks, however, we additionally calculated these areas based on the relatively intact upper-crustal exposures along the West Fizh block (Wadi Fayd to Wadi Kabiyat) and Hilti block (Wadi Bargah to Wadi Ahin). These Fizh and Hilti areal proportions are the most representative available for the pre-obduction state of the Semail crust (Table 2).

Interestingly, the Batinah Coast and Fizh–Hilti spatial subsets are relatively consistent with each other. In both cases, our

inferred bedrock maps indicate that Tholeiitic Alley is the most areally-extensive volcanic unit. If the SDC, which is largely comagmatic with Geotimes (Miyashita et al., 2003; Pearce et al., 1981), is considered with the volcanics and upper-crustal intrusions to make up an upper-crustal total, the proportion of axial upper crust (SDC + Geotimes; Phase 1) is ~54 vol%,

whereas Phase 2 off- or post-axial lavas constitute ~44 vol%. In the Fizh–Hilti subset, 24 vol% of the volcanics or 15 vol% of the upper crust is made up of boninite-series lavas.

These significant volumes of Phase 2 lavas imply that their generation must have depleted considerable volumes of mantle following the axial-spreading stage (Geotimes) melt extraction. Such extensive Phase 2 melting and magmatic overprinting has previously been described in the mantle section (Arai et al., 2006; Python and Ceuleneer, 2003). Plutonic evidence for Phase 2 magmatism has also been documented throughout the ophiolite (Adachi and Miyashita, 2003; Goodenough et al., 2010; de Graaff et al., 2019; Haase et al., 2016; Juteau et al., 1988; Rollinson, 2009; Tsuchiya et al., 2013; Yamasaki et al., 2006). The volumetric significance of this late plutonism is poorly constrained, though Phase 2 intrusives apparently constitute about half of the crustal exposures in the U.A.E. portion of the ophiolite (Goodenough et al., 2010, 2014), which is comparable to the proportions of Phase 1 to Phase 2 lavas documented here. It follows from these proportions that many of the major tectonic and magmatic features in the mantle and lower crustal sections are feasibly related to Phase 2 magmatism. For example, ~14 mantle diapirs have been mapped by Nicolas et al. (2000). Although troctolitic cumulates at the head of the Maqsad mantle diapir suggest it formed during the MORB-like axial phase, the same is not clear for many of the other diapirs (Python and Ceuleneer, 2003).

The unit proportions documented in Table 2 are the first estimates of the composition of proto-arc upper crust based on a large mapped area (2059 km$^2$). The proportion of boninites in the Semail ophiolite is comparable to the exposures of boninitic upper crust estimated for the southern Mariana forearc by submarine surveying (20–30 area%; Reagan et al., 2013).

Although the unit proportions documented in the present study are representative for a significant length along the paleo-spreading axis (over ~150 km), any large-scale variations perpendicular to this axis are difficult to estimate across the 30–60 km wide Semail ophiolite. The comparable units and geochemistry between the NE (Batinah coast) and SW (Yanqul) flanks of the ophiolite, which are separated by ~30 km perpendicular to the spreading axis, suggests some continuity of volcanostratigraphy at this scale. However, flat-lying ophiolites (e.g. Mirdita), the IBM arc basement and the recently recognised Matthew and Hunter proto-arc show significant variations in upper-crustal composition with distance from the trench (Dilek et al., 2008; Hickey-Vargas et al., 2018; Patriat et al., 2019; Reagan et al., 2017). The upper crustal unit proportions documented herein are therefore not necessarily representative of an *entire* proto-arc crustal swath, but rather represent the best estimate currently available.

### 7.3. Along-ophiolite differences in subduction-zone influence

It has previously been proposed that a gradient of subduction-zone influence on magma generation existed from SE to NW along the Semail ophiolite. This proposal is based on Cr contents in mantle spinel (Python et al., 2008), abundant Phase 2 intrusives in the U.A.E. (Goodenough et al., 2010) versus abundant troctolites in the southeastern blocks (Python and Ceuleneer, 2003), and an apparent paucity of Lasail/Alley volcanics in the southeastern ophiolite blocks. The existence of this gradient has since been challenged by the apparently uniform hydrous influence on axial magmatism (MacLeod et al., 2013), the identification of numerous Phase 2 intrusions in the southern ophiolite blocks (de Graaff et al., 2019; Haase et al., 2016),

structural arguments for outcrop bias in the U.A.E. (Ambrose and Searle, 2018), and by the synchronous ages for prograde metamorphism in the southern and northern metamorphic soles (Guilmette et al., 2018).

Owing to the tectonic omissions of Geotimes in the Sarami and Haylayn blocks, our new map is somewhat equivocal on the issue of ophiolite-scale variations in subduction influence. However, it is at least clear that subduction-influenced Lasail and Alley volcanism occurred over the length of the mapping area. In fact, some of the ophiolite's thickest accumulations of Felsic- and Boninitic Alley lavas occur in the Haylayn and Wuqbah blocks (Figs. 12, 13), indicating that, at least during Phase 2 magmatism, subduction-influenced volcanism was comparably significant throughout the mapping area.

### 7.4.  Lateral continuity of the Alley unit

On the basis of the Suhaylah Village section, it has long been held that Geotimes locally tops the volcanic sequence, being directly overlain by Suhaylah Formation sediments, and therefore that the Alley strata are discontinuous (Fleet and Robertson, 1980; Lippard et al., 1986). However, the presence of Geotimes at the top of this section was apparently not confirmed by geochemical analyses. In contrast, our field observations, whole-rock analyses and clinopyroxene analyses reveal that the uppermost ~300 m of the Suhaylah section are in fact composed of Tholeiitic Alley pillow basalts and andesitic massive flows (Fig. 14).

A similar locality with Geotimes topping the section was indicated in upper Wadi Ahin (map by BME, 1987b). However, our sampling (TB4-23A) and field observations show that Boninitic Alley lavas are present there, and that they are actually faulted against, rather than overlain by, sediments on both sides of the wadi. Additional occurrences of Suhaylah sediments directly overlying Geotimes had been mapped between Wadi Hayl and Wadi Kabiyat, ~6 km north of Suhaylah in the Fizh block (BME, 1987c). Our field observations, as well as those of Robertson and Woodcock (1983), rather indicate that this sequence is disrupted by normal faulting, which has removed much of the section. Our sampling further shows that the uppermost unit along this section is Boninitic Alley (samples LD10-1461, LD10-1464). No other examples of Suhaylah sediments directly overlying Geotimes are indicated in the regional map set or were found during the present study. We thus conclude that Alley/V2-stage lavas are continuously present throughout the northern ophiolite.

### 7.5.  Lasail basaltic 'seamounts'

The discrete accumulations of Lasail lavas that overlie Geotimes have previously been likened to off-axis 'seamounts', regularly spaced and associated with VMS deposits (Alabaster et al., 1982; Lippard et al., 1986; Pearce et al., 1981). Prior to this study, these accumulations had not been mapped out in detail. The newly mapped Lasail accumulations are generally less than a few hundred metres thick, less than a kilometre wide and are irregularly distributed at intervals of 10–20 km wherever the top of the Geotimes unit is well exposed. These findings thus support the seamount interpretation, but their frequency is slightly higher, and their volumetric significance slightly lower, than originally suggested by Alabaster et al. (1982).

Thick Lasail accumulations occur in the region of proposed spreading-axis segment boundaries, e.g. Wadi Hatta to Rajmi, and Ghuzayn, as noted by Alabaster et al. (1982) and MacLeod and Rothery (1992), but the unit's distribution is not limited

to these positions. Lasail also occurs within the middle of intact structural blocks (e.g. Salahi area). Moreover, as also shown by Gilgen et al. (2014), there are no clear associations between these seamounts and VMS deposits.

The most significant of the Lasail accumulations occurs between Wadis Ashar and Rajmi (Fig. 15), and does in fact coincide with a proposed area of axial segmentation (MacLeod and Rothery, 1992; Reuber, 1988; Smewing, 1980). With a maximum thickness of ~ 1 km and a continuous base ~15 km along strike (Figs. 12, 15), the weakly magnetic Lasail lavas are well distinguished from the strongly magnetic overlying Alley lavas in the aeromagnetic survey, as confirmed by sampling and field observations (Fig. 15). Isolated occurrences of Lasail volcanism began synchronously with earliest Geotimes eruptions in this area (Alabaster et al., 1982; Belgrano and Diamond, 2019) and the Geotimes unit itself is abnormally thin, being only a few hundred meters thick (Section A in Fig. 12), suggesting short-lived axial volcanism and a rapid development of subduction influence in this area.

## 7.6. Semail boninites

Analyses of volcanic glass demonstrate that boninites (LSB) sensu stricto are present throughout the northern ophiolite (Fig. 6; Kusano et al., 2017). The well defined, high V/Ti and high Zr/Y fractionation trends followed by both 'Boninitic Alley' spilites and the boninitic glasses (Fig. 5) indicate that all of these lavas belong to the same unit, which in turn derives from a low-Si boninite parental melt (Fig. 6).

Prior to this study, little was known about the areal extent and volcanic morphology of boninites in the ophiolite. Our mapping, though unlikely to have recorded all the minor boninite occurrences within the mapping area, shows that boninites occur in each of the mapped blocks. These occurrences are discontinuous and have a highly variable stratigraphic thickness (0.1–2 km; sections in Fig. 12). In the Wuqbah, Hilti and Sarami blocks, layers of Boninitic Alley only up to ~200 m thick discontinuously cap and locally interfinger with the uppermost Tholeiitic Alley lavas (e.g., Kusano et al., 2014). Contrastingly, there are boninite accumulations 1.5–2 km thick with bases 2–5 km wide in the Aswad, Fizh and Haylayn blocks (Fig. 12).

### 7.6.1. Structurally-controlled boninitic volcanism

The best-preserved boninite accumulations can be found between Wadi Rajmi and Wadi Zab'in in the Fizh block (Fig. 16). Here, three sub-parallel dyke swarms spaced at ~5 km fed a 2 km thick accumulation of Boninitic Alley lavas. Ishikawa et al. (2002) noted the boninitic character of these dyke swarms, which apparently show mutually intrusive relationships with orthopyroxene-series (gabbronorite) intrusions (Umino et al., 1990). Together with the shallowly SSE-dipping SDC along this segment, the presence of these gabbronorites has previously been linked to axial segmentation (MacLeod and Rothery, 1992; Reuber, 1988). In light of the newly mapped Rajmi–Zab'in boninite accumulations (the only orthopyroxene-bearing volcanic unit; Ishikawa et al., 2002), the possibility that some of these gabbronorites are the intrusive equivalent of the boninite lavas warrants consideration.

The mantle and lower crustal section in the Rajmi–Zab'in area (Fig. 16) is cut by numerous high-temperature, mylonitic shear zones (BRGM, 1993b; Smewing, 1980; Takazawa et al., 2003). Though cataclastic zones beneath the Moho and Q'

quartz–hematite veins in the upper crust (BRGM, 1993b) indicate that many of the faults in this area experienced post-magmatic brittle reactivation, the high-temperature fabric and sheeted intrusions emplaced along these fault zones demonstrate the syn-magmatic timing of the original structures. Greater displacement of the axial sequence relative to the Alley sequence along these syn-magmatic faults further suggests that part of this upper-crustal deformation occurred after axial magmatism but before or during Alley-stage magmatism.

The centres of the two most prominent boninite lava outcrops at Rajmi–Zab'in are clearly fed by dykes intruding along these syn-magmatic faults (Fig. 16), demonstrating that boninite migration through the existing axial crust was structurally controlled. Moreover, the mylonitic shear zones which underlie and connect with these syn-magmatic faults indicate that boninitic melts were structurally channelled from the level of the mantle. In fact, the major mantle shear zone underlying the Wadi Zab'in boninite dyke and lava complex (Fig. 16) is well-studied, and records the passage of hydrous fluids and melts from the metamorphic sole through the entire mantle sequence (Arai et al., 2006; Kanke and Takazawa, 2014; Takazawa et al., 2003). Comparable, shear zone controlled Phase 2 magmatism is widespread in the U.A.E. portion of the ophiolite (Goodenough et al., 2010; Styles et al., 2006). The structurally-controlled boninitic volcanism documented herein completes the spectacular mantle-to-seafloor section at Rajmi, constituting a complete anatomy of a proto-arc magmatic system.

Interestingly, the boninite-focusing fault zones between Wadi Rajmi and Wadi Zab'in extend directly into the mantle (Fig. 16) and there they enclose a cluster of structurally-controlled podiform chromitite deposits around Wadi Rajmi (Boudier and Al-Rajhi, 2014; Rollinson, 2008). Chromites within these deposits record formation from a series of melts evolving from arc-like (comparable to Tholeiitic Alley) to highly oxidised, boninitic compositions (Rollinson, 2008; Rollinson and Adetunji, 2015). The short duration permitted for boninitic volcanism (~0.5 Ma; Gilgen et al., 2016 and references therein) together with the close geochemical and structural association between these chromite deposits and the thick overlying boninite accumulations supports the long-held association between hydrous, boninitic melts and podiform chromitite formation (Matveev and Ballhaus, 2002; Rollinson, 2008). The mantle sections beneath the other significant boninite accumulations (e.g., Aswad, Daris) may therefore be attractive areas for future chromite exploration.

## 7.7. Relationships between the volcanic units and the intrusive phases

The intrusive rocks of the Semail ophiolite have been subdivided into two main groups, with the axial 'High-Level' and off-axis 'Late Intrusive' groupings of Lippard et al., (1986) respectively coming under the Phase 1 and Phase 2 groupings of Goodenough et al. (2014). It is generally accepted that the Phase 1 intrusions are related to the axial SDC and Geotimes unit (Goodenough et al., 2010; Lippard et al., 1986), however, the relationship between the Phase 2 intrusives and the various Phase 2 volcanic units is less clear. This uncertainty has persisted because prior to this study the distributions and relative proportions of the Phase 2 volcanic units were unclear.

Several lines of evidence suggest that the majority of the Phase 2 intrusives are comagmatic with the Tholeiitic Alley lavas. Firstly, our map shows that the Alley suite is volumetrically far more significant than the Lasail lavas (Table 2), suggesting that Alley should have a more significant proportion of intrusive equivalents. Secondly, the Phase 2 intrusive complexes

characteristically span a range of compositions throughout the ophiolite (gabbro–diorite–tonalite–trondjemite), often within single complexes (Lippard et al., 1986). The equivalent compositional range (basalt to rhyolite) is characteristic of the Tholeiitic Alley lava suite, but not of Lasail, which is predominantly comprised of primitive basalts.

This evidence is supported by the Zr–Ti trends of the different lavas and intrusives (Fig. 5a). The Phase 1 intrusive compositions fall along and continue from the Geotimes fractionation path, supporting the hypothesis that these intrusives formed by fractional crystallisation of a Geotimes melt (Haase et al., 2016). The Phase 2 intrusive compositions rather mostly follow the Tholeiitic Alley fractionation path, extending to far lower Ti/Zr ratios than recorded for the Lasail lava unit, supporting the intrusive–extrusive equivalence of the Phase 2 intrusive and Tholeiitic Alley suites.

Interestingly, a subset of Phase 2 intrusive compositions are continuous with the upper Boninitic Alley field, suggesting that these intrusives evolved from depleted, boninite-series melts. The intrusions emplaced into Boninitic Alley lavas in the Rajmi area (Fig. 15) are also presumably boninite-series, as these boninites are the last-erupted lavas in this area.

### 7.8. Relationships between faulting, dyke swarms, boninites, and sulphide ore deposits

A genetic association between late dyke swarms, normal faulting, volcaniclastic breccias, boninitic volcanism and VMS deposits has previously been noted for the 'Bowling Alley' fault zone between Wadi Bani Umar and Wadi Kabiyat (Fizh block) and the nearby Aarja and Bayda VMS deposits (Gilgen et al., 2014; Haymon et al., 1989; Smewing et al., 1977). Early mapping viewed the fault zone as an E-facing half graben that along its southern reach swings SW into Wadi Jizi (Smewing et al., 1977). In contrast, our mapping reveals unusual repetition of the Geotimes unit to the east of Tholeiitic Alley, which is suggestive of a full graben running N–S from at least present-day Wadi Fizh to Wadi Bargah (Fig. 17). The continuation of this rift-like feature south of Wadi Jizi is marked by a strip of Boninitic Alley lavas, Felsic Alley lavas and Suhaylah sediments bounded to the E and W by faults, with Geotimes lavas repeated to the E of the Alley lavas. During ophiolite emplacement, the rift was dismembered by faulting parallel to Wadi Jizi (marked by the Zabyat conglomerates; Robertson and Woodcock, 1983) and by NNE–SSW faulting along the western edge of the East Fizh block. Several further corridors of intrusions run parallel to the Bowling Alley axis, which we interpret as related to secondary rift structures (Fig. 16). Thus, the entire axis-parallel rift is ~30 km long and it was the locus of VMS mineralization from the end of Geotimes (Lasail deposit) to Boninitic Alley volcanism (Aarja deposit; Smewing et al., 1977; Gilgen et al., 2014), indicating that extensional tectonics and vigorous hydrothermal activity occurred in this area throughout axial volcanic Phase 1 and later Phase 2.

Our mapping indicates that a similar lithotectonic association exists for the boninite-hosted Rakah and Hayl al Safil VMS deposits in the Yanqul area (Gilgen et al., 2014). Here, Alley-stage 'sheeted' dyke swarms (confirmed by analysis of dyke samples TB3-25H and TB4-20L) run parallel and adjacent to the faults and grabens bounding the deposits, which are themselves filled with blocky volcaniclastics and boninitic lavas. The boninite-hosted Safwa VMS deposit is also located between the two major W–E striking, fault-localised boninite dyke swarms between Wadi Fayd and Wadi Zab'in (Fig. 16).

The extensional tectonics associated with these late dyke swarms therefore appears to be favourable both for VMS mineralization, as pointed out by Smewing et al. (1977) and Haymon et al. (1989), as well as boninite extrusion, as similarly

noted for the 'infill' boninites of the Troodos ophiolite (Cameron, 1985; Osozawa et al., 2012). Further work is needed to resolve the specific timing and genetic associations between these features.

## 8.    Comparison with the Izu-Bonin-Mariana proto-arc sequence

The Semail ophiolite has been proposed as analogous to the Eocene Izu-Bonin-Mariana (IBM) proto-arc crust on the basis of their similar structure and their similar 1–2 Myr progression from axial basaltic to boninitic magmatism (Belgrano and Diamond, 2019; MacLeod et al., 2013; Reagan et al., 2019; Rioux et al., 2016; Rollinson and Adetunji, 2015). To facilitate comparison of the mapped Semail sequence with the IBM proto-arc record, we plot the updated Semail volcanic unit fields from Fig. 5 together with the published compositions of IBM lavas, as well as Tonga Trench boninites and MORB glasses, in Fig. 18.

The biggest geochemical discrepency between the Semail and IBM volcanic sequences is the absence of a high Ti/V, 'moist', MORB-like equvivalent to Geotimes in the IBM (Belgrano and Diamond, 2019; MacLeod et al., 2013). The highest Ti/V forearc basalts recovered from the IBM derive from the Amami-Sankaku basin rear-arc (Unit 1; Hickey-Vargas et al., 2018), and from the Bonin forearc (DeBari et al., 1999). Originally thought to be fragments of pre-subduction-initiation MORB, these Bonin forearc basalts were later reclassified as forearc (proto-arc) basalts (Reagan et al., 2010). These high Ti/V proto-arc basalts overlap with the Geotimes and MORB arrays in Fig. 18a, c, and d, but Geotimes follows a still-higher Ti/V trend in Fig. 18b. No direct equivalent for the Geotimes unit has thus been identified in the IBM proto-arc sequence. A pre-axial episode of melt extraction, inferred for the IBM source from trace elements and Hf–Nd isotopes (Li et al., 2019; Shervais et al., 2019; Yogodzinski et al., 2018), but so far not for the Semail ophiolite, may explain this discrepancy (Belgrano and Diamond, 2019). This difference in pre-axial mantle source history implies that when subduction initiation occurs above normal, depleted MORB mantle, as appears to have been the case for the Semail ophiolite (Belgrano and Diamond, 2019; Godard et al., 2006), the first proto-arc lavas to erupt may have a composition closer to MORB than forearc basalts from the IBM type locality (Reagan et al., 2010).

Recent drilling in the IBM forearc identified a unit of primitive basalts directly overlying the axial volcanics (P-FAB; Shervais et al., 2018). The low-Ti, primitive major-element composition of P-FAB is closely comparable to Lasail. The trace element composition of P-FAB mostly overlaps with Lasail but is not an exact equivalent. However, the compositional relationship between P-FAB and the axial forearc basalts which they overlie is closely comparable to the relationship between Lasail and Geotimes, with both P-FAB and Lasail being slightly more primitive, depleted and with lower Ti/V than their underlying axial lavas (Fig. 18). We therefore hypothesise from the known lateral discontinuity of the Lasail unit that P-FAB may also be laterally discontinuous atop the IBM axial crust.

Tholeiitic Alley is the closest geochemical equivalent to the normal IBM forearc basalts, especially in terms of Ti/V (Fig. 18). The key difference between these two units is that in the IBM, these low Ti/V forearc basalts make up the axial crust, whereas in the Semail ophiolite, they are erupted on top of the axial crust.

Boninitic Alley is depleted in incompatible elements and enriched in fluid-mobile elements similarly to the Bonin forearc boninites (Ishikawa et al., 2002; Kusano et al., 2014). In terms of major elements, the Semail ophiolite lacks a HSB suite, and Boninitic Alley follows a slightly shallower, lower-Si fractionation trend on the MgO–SiO$_2$ diagram than the IBM and Tonga trench boninites (Fig. 6a). In terms of maximum MgO contents, Boninitic Alley is more similar to the subduction initiation-related IBM boninites (Murton et al., 1992; Reagan et al., 2017; Shervais et al., 2019), than to the remarkably magnesian plume- and slab window-related north Tonga trench boninites (Falloon et al., 2007, 2008). In terms of the trace elements plotted in Figure 18, the IBM and Tonga Trench boninites are highly depleted and scattered, but mostly overlap with Boninitic Alley. The most notable exception to this similarity are the higher Zr contents in the IBM boninites, shifting them to higher Zr arrays in Fig. 18a and d. Positive Zr–Hf anomalies are ubiquitous in the IBM boninites and have been attributed to contamination by slab melts originating from an amphibolite-bearing slab (Pearce et al., 1992; Reagan et al., 2017). The absence of these anomalies in Boninitic Alley suggests differing boninite-contaminating slab components between the Semail and IBM settings.

Overall, the comparable structure, chronology and progressive depletion of the Semail and IBM magmatic sequences supports the proto-arc interpretation for the genesis of the Semail ophiolite. However, this short evaluation suggests that the pre-axial mantle source and contaminating slab melts were probably different between the two settings (see also Belgrano and Diamond, 2019; Kusano et al., 2017). The Semail ophiolite further lacks a suite of true HSB, which are common in the IBM. The Semail–IBM analogy is thus far from perfect. However, complex tectonic and magmatic histories appear to be common, or even necessary, preconditions for initiating subduction (Patriat et al., 2019; Stern and Gerya, 2017). Compositional differences between different proto-arc volcanic sequences and the IBM type locality are therefore perhaps to be expected, and these differences represent a promising line of research to describe the variety of situations in which subduction may initiate.

## 9.   Conclusions

We have presented the methodology and final product of a five-year project remapping the volcanic units in the Semail ophiolite. Our key conclusions can be summarised into three categories:

Mapping methodology:

- Previous regional mapping of the ophiolite relied mainly on field observations and early satellite imagery. With these methods, resolution of the various units within the 'V2' stage extrusives was impractical at a regional scale. Our study has benefitted from rapid and routine whole-rock XRF and EMP mineral analysis. These techniques have enabled large numbers of sample analyses to be used to identify units over several hundred km$^2$.
- The bulk-magnetic properties of submarine lava suites are strongly controlled by major element composition and can be well preserved despite overprinting by sub-seafloor spilitic alteration. It should therefore be feasible to distinguish volcanic units using aeromagnetic data in other ophiolite or greenstone-belt settings. However, strong remanent magnetism in submarine volcanics leads to inconsistencies in RTP data, therefore less susceptible data transformations should be used if possible.

- Combining magnetic petrology and geochemistry allows aeromagnetic data to be usefully correlated with specific units in volcanic terranes. Our confidence in these interpretations was sufficient to map specific units beneath sedimentary cover, thereby significantly expanding our bedrock map and the prospective area for VMS deposit exploration.

Upper-crustal features of the Semail ophiolite:

- Of the two lava units traditionally ascribed to Phase 2 magmatism, Alley is volumetrically far more significant than Lasail.
- Incipient boninitic volcanism ('Depleted Lasail') locally occurred both during and directly following the Geotimes axial stage as part of the Lasail unit (see also Belgrano and Diamond, 2019), but before the eruption of the Alley tholeiites and boninites.
- Alley/Phase 2 volcanism is continuously present throughout the northern ophiolite and was volumetrically comparable to axial Geotimes/Phase 1 volcanism. These findings reinforce the strongly subduction-influenced character of the ophiolite as a whole (e.g., de Graaff et al., 2019; Goodenough et al., 2010; Haase et al., 2016; MacLeod et al., 2013; Pearce et al., 1981), and caution against unconfirmed assumptions that geological features in the lower-crustal or mantle sequences of the ophiolite (e.g. mantle diapirs) must necessarily reflect processes related to an oceanic spreading axis.
- The traditional Alley unit is systematically divisible into two main units at the ophiolite scale: lavas belonging to a volumetrically dominant tholeiitic series versus lavas belonging to a subordinate low-Si boninitic series.
- Obduction-related remagnetization is common around major fault zones. Moreover, quartz–hematite and quartz–carbonate veins with significant chloritic and carbonate alteration haloes cut the entire crustal sequence and the anticlinal structure that warps the ophiolite. These veins clearly post-date the ophiolite's crust-formation stage, but nevertheless they are often sited in reactivated syn-magmatic faults. Care is therefore necessary when interpreting the significance of chloritised hydrothermal features where clear cross-cutting relationships are unavailable.
- Previously proposed associations between the ends of spreading-ridge segments, VMS deposits, and off-axis Lasail seamounts (Alabaster et al., 1982) are less convincing in light of the new map. Boninite accumulations, late dyke swarms and extensional tectonics are rather more commonly associated with economic VMS deposits. s

Accretion of proto-arc crust:

- Despite representing apparently differing modes of subduction initiation (Arculus et al., 2015; Guilmette et al., 2018), the close comparability between the Semail and IBM magmatic sequences is supported by our new mapping and sampling, as well as by recently collected datasets from both settings (Belgrano and Diamond, 2019; MacLeod et al., 2013; Reagan et al., 2017; Shervais et al., 2019; Whattam and Stern, 2011). However, subtle compositional differences between stratigraphically-equivalent lava units in each setting suggest contrasting pre-axial histories and slab-contaminations between the two settings.
- The two most substantial axial (Geotimes) and post-axial (Tholeiitic Alley) volcanic episodes were somewhat variable in thickness but almost continuous along the length of the paleo-spreading axis within the ophiolite (~150 km). Contrastingly, the off-axis Lasail and final Boninitic Alley episodes were discontinuous and variable in thickness. Overall, ~40 vol% of the Semail upper crust was accreted after the main axial stage.
- Boninitic melts in the ophiolite were locally channelled along lithosphere-scale shear zone networks which apparently extend from the metamorphic sole to the paleo-seafloor. Podiform chromitite deposits formed along these same melt channels in association with the ascending proto-arc basaltic and boninitic melts (Boudier and Al-Rajhi, 2014; Rollinson, 2008), concurrently with VMS mineralization on the seafloor (Gilgen et al., 2014). A complete cross-section through this system is accessible in the Wadi Rajmi–Zab'in area of the Fizh block.

The new map represents significant progress compared to the previous two-unit stratigraphy, and it should prove useful for further research and mineral exploration in the ophiolite. However, maps are inevitably interpretations of reality, and so the

present contribution is provided in an editable format, with the hope that it may be updated in the future as new data become available.

## 10. Data Availability

All new geochemical and rock magnetic data are available at https://doi.org/10.1594/PANGAEA.899794. The complete dataset
is also included in the supplement to this manuscript. The supplement also includes the complete set of magnetic mineralogical figures (Figs. S1–S9), tables of criteria used to assign each sample to a unit (Table S1), previously published sample locations used for mapping (Table S2), analyses of standard materials (Table S3–4), and a Geospatial PDF of the final volcanic map and input layers.

## 11. Author contributions

The project was conceived by LWD and SAG. Field mapping and geochemical analysis was led by TMB and YV. ARB and TMB measured and interpreted the rock magnetic properties. KAT provided assistance in the field and logistical support. TMB and LWD prepared the manuscript with contributions from all co-authors.

## 12. Competing interests.

The authors declare they have no conflict of interest.

## 13. Acknowledgements

We gratefully acknowledge the support of the Public Authority for Mining (PAM), Sultanate of Oman, in particular Salim Omar Al-Ibraheem, Mohammed Al Araimi, Mohammed Al-Battashi and Ali Al Hashmi. We thank Samuel Weber, Alannah Brett, Sarah Pein, Nevena Novakovic, Lisa Richter (all University of Bern) and Ali Al Hashmi (PAM) for assistance during field mapping. We thank Lydia Zehnder (ETH Zurich) for her help with numerous XRF measurements. Pierre Lanari and Elias
Kempf kindly facilitated our microprobe measurements. Daniel Peters and Thomas Pettke provided LA-ICP-MS support. Kathryn Goodenough and Yuki Kusano are thanked for their insightful reviews and helpful comments. Rock magnetism measurements were performed by TMB as a Visiting Fellow at the Institute for Rock Magnetism (IRM) at the University of Minnesota, under the welcome guidance of Dario Bilardello and Michael Jackson. The IRM is a US National Multi-user Facility supported through the Instrumentation and Facilities program of the National Science Foundation, Earth Sciences
Division, and by funding from the University of Minnesota. All other aspects of this project were funded by the University of Bern and by Swiss National Science Foundation (SNSF) Grant 200020-169653 to LWD.

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

## 15. Figure captions

**Figure 1**. Map of the Semail ophiolite structural blocks simplified after Nicolas et al. (2000), with the additional differentiation of the structurally-separate Western and Eastern Fizh blocks, showing the area mapped in this study. Inset: Map of Tethyan ophiolites adapted after Dilek et al., (2007) with U-Pb zircon ages of initial accretion indicated for the Albian–Turonian ophiolite chain of [1]Troodos, [2]Kizildag, [3]Nain, [4]Deshir, and [5]Semail after Shafaii Moghadam et al., (2013) and references therein.

**Figure 2.** Semail ophiolite volcanostratigraphy with capping pelagic sediments (Suhaylah Formation) and Sheeted Dyke Complex (SDC), with approximate stratigraphic thicknesses revised according to field observations in this study after Alabaster et al. (1982), Gilgen et al. (2014) and Belgrano and Diamond (2019).

**Figure 3.** Field photographs of typical outcrops of the mapped volcanic units, with UTM 40N location coordinates. Hammer for scale is 50 cm long. (a) Geotimes greenschist-facies spilitic pillow flows: Wadi Jizi type locality (438288 mE, 2685876 mN). (b) Lasail greenschist-facies spilitic pillow and pahoehoe flows showing abundant chlorite (Chl): Wadi Lasail (440680 mE, 2683521 mN). (c) Tholeiitic Alley zeolite/pumpellyite-facies spilitic pillow flows and hyaloclastite breccia (light streak in centre-pillow is a road-cutting artefact). Note sea-green celadonite (Cel) alteration in interpillow and breccia, yellowish-beige pumpellyite alteration around pillow rim, abundant zeolite-filled vesicles and dark grey spots (spherules) throughout pillow (sample TB4-17J): Suhaylah section, blasted roadcut around electricity pylon (434460 mE, 2683370 mN). (d) Primitive Tholeiitic Alley pillows with highly vesicular pillow rims and dark spots throughout the pillow cross-section: roadcut near Maqail South Mine (453534 mE, 2661208 m N). (e) Boninitic Alley pillow lavas with weak pumpellyite–clay altered groundmass and zeolite altered (Zeo) interpillows (sample TB2-45B; 448940 mE, 2712935 mN). (f) Boninitic Alley greenschist-facies pillow lavas with three pillow triple-junctions marked by dotted lines. Note pale chlorite–albite alteration with chlorite–epidote replacing olivine in pillow rims and large epidote-filled vesicles (Ep) through pillow centre: Wadi Zab'in (approx. 436040 mE, 2714170 mN). (g) Boninitic Alley pillow lava showing globular/spherulitic texture typical of Alley (particularly Boninitic Alley) lavas: Wadi Bidi (455370 mE, 2667380 mN).

**Figure 4.** Incompatible-element patterns normalised to N-MORB (Gale et al., 2013) for a subset of our samples. (a) Geotimes, transitional Geotimes–Lasail, Lasail, and depleted-Lasail patterns (this study). (b) Tholeiitic and Boninitic Alley patterns in comparison to the Geotimes and Lasail ranges (this study) and low- and high-silica boninite glass compositions from (Kusano et al., 2017). (c) Lasail, Tholeiitic and Boninitic Alley compositions in comparison to V2 type I and II groups of Godard et al., (2003).

**Figure 5.** Geochemical unit-discrimination diagrams (anhydrous compositions). Circles: this study. Squares: published analyses re-examined as unknowns (asterisked if assigned differently to the publications; Gilgen et al., 2016, 2014; Kusano et al., 2017). (a) Whole-rock Ti–Zr after Alabaster et al. (1982). (b) Whole-rock Ti–V and Ti/V ratios (grey lines) after Shervais (1982). (c) Whole-rock Y–Cr after Pearce (1980). (d) Zr–Zr/Y after Alabaster et al. (1982). (e) Clinopyroxene median Mg# (= molar Mg/(Mg + Fe)) vs. Ti (atoms per formula unit) for a subset of samples. (f) Clinopyroxene lower and upper Mg# quintile medians with corresponding Ti for the same dataset as in (e). Dashed lines in (a)–(d): previous whole-rock unit fields based on stratigraphically-defined data for Geotimes (Alabaster et al., 1982; Belgrano and Diamond, 2019; Einaudi et al., 2003; Godard et al., 2006; Kusano et al., 2012), Lasail (Alabaster et al., 1982; Belgrano and Diamond, 2019; Godard et al., 2006; Kusano et al., 2012), Tholeiitic Alley (Alabaster et al., 1982; Kusano et al., 2014), and Boninitic Alley (Ishikawa et al., 2002; Kusano et al., 2014). Volcanic glasses (Kusano et al., 2017) and Semail intrusives (excluding wehrlites; de Graaff et al., 2019; Haase et al., 2016). Dashed lines in (e) and (d): Previous clinopyroxene fields (Alabaster et al., 1982; Belgrano and Diamond, 2019; Gilgen et al., 2014; Kusano et al., 2012). Coloured fields: this study.

**Figure 6.** Boninite classification diagrams based on whole-rock major element oxide concentrations normalised to 100 wt%. (a) $SiO_2$ vs. MgO with rock fields after Pearce and Reagan (2019), SHMB = siliceous high magnesium basalt, HMA = high magnesium andesite, D = dacite. (b) MgO vs $TiO_2$ with boninite/low-Ti basalt field adapted to log scale after Pearce and Reagan (2019), and original BADR–boninite series divider (grey dashed line) after Pearce and Robinson (2010). Filled circles: volcanic glasses (this study). Inverted triangles: volcanic glasses (Kusano et al., 2017). Unfilled circles: Boninitic Alley and Depleted Lasail spilites (this study). Unfilled squares: Boninitic Alley and Depleted Lasail spilites (Belgrano and Diamond, 2019; Gilgen et al., 2014, 2016). West Pacific boninite analyses from the N Tonga Trench (Falloon et al., 2007; Falloon and Crawford, 1991), Izu-Bonin forearc (IODP Expedition 786; Murton et al., 1992), and Bonin forearc (IODP Expedition 352; Reagan et al., 2017).

**Figure 7.** Statistical summary of bulk magnetic measurements of hand specimens of the Semail volcanic units carried out for this study. Bold lines: medians. Coloured boxes: interquartile ranges. Whiskers: minimum–maximum. Felsic subunits are grouped with parent units. Numbers in coloured boxes show number of samples analysed. (a) Magnetic susceptibility (both Magnon and KT-5 measurements grouped). (b) Natural remanent magnetism (NRM), unoriented samples. (c) Koenigsberger ratios ($Q$) of remanent to induced magnetization, calculated for each sample with unoriented NRM, a density of 2.67 g/cm$^3$ (mean for Geotimes spillites; Einaudi et al., 2003) and the local geomagnetic field for 1992 (42,900 nT; Thébault et al., 2015).

**Figure 8.** Magnetic susceptibility vs. whole rock Mg# (= molar Mg/(Mg + $Fe_{total}$)) for the four main Semail volcanic units. Circles and diamonds: this study. Squares: Geotimes/V1 samples from Einaudi et al. (2003). Outlying analyses: fresh boninite

TB3-01A; highly carbonated Geotimes TB3-20I (LOI = 16 wt%) from vicinity of late carbonate vein; high-Si Transitional Alley TB3-15C (70 wt% $SiO_2$) and Tholeiitic Alley TB2-34 (84 wt% $SiO_2$).

**Figure 9**. Magnetic mineralogy of the Semail volcanics. (a–d): High-temperature magnetic susceptibility ($K$) of four representative samples, with the first derivative of the heating curve (temperatures marked at d$K$/d$T$ minima/maxima). (a): Titanomagnetite in relatively fresh Boninitic Alley (TB3-01A). (b): Magnetite in Tholeiitic Alley (TB2-33C). (c): A mixture of maghemite and magnetite in Felsic Alley (TB3-25E). (d): Maghemite in Geotimes (TB3-07C). (e–h): Low-temperature FC and ZFC remanences and RT-IRM during cooling and warming of the same samples as in A–D. Asterisked 'transition' in (e) possibly caused by unintentional movement of the sample. (i): Frequency of ferromagnetic mineral occurrences (excluding weakly magnetic hematite) deduced from high- and low-$T$ experiments for the entire sample set ($n = 36$) and within each unit ($n_{Geotimes} = 14$, $n_{Lasail} = 6$, $n_{Thol. Alley} = 9$, $n_{Bon. Alley} = 11$). Where multiple magnetic minerals are detected in a single sample, all the minerals contribute to the unit count.

**Figure 10.** Reduced-to-pole (RTP) magnetic anomalies for the Batinah coast and Yanqul area (Isles and Witham, 1993), marked with our interpretations of the block boundaries, the extent of volcanic bedrock, structurally-controlled re- or de-magnetised features, the potentially magnetic Batinah complex modified after BRGM (1986a, 1986b) and Woodcock and Robertson (1982), and possible repetitions or extensions of ophiolite blocks (numbered) visible in the RTP map, supported by the gravity interpretations of Shelton (1990). The Batinah RTP map and re-magnetized features are viewable as separate layers in the supplementary Geospatial PDF.

**Figure 11** Summary of datasets used to construct the final map in addition to the aeromagnetic survey in Fig. 10. Red circles: field mapping locations, this study. Reference samples include triangles: volcanic glasses (this study); filled diamonds: dykes and sills (this study); unfilled diamonds: dyke samples from Adachi and Miyashita, (2003); squares: lava and volcanic glass samples from Gilgen et al. (2016, 2014), Kusano et al. (2017) and MacLeod et al. (2013); unfilled squares: P1 and P2 intrusive samples from Haase et al. (2016) and de Graaff et al. (2019). Numbered outlines: coverage of previous publications used to support our mapping: (1) BRGM (1993b), (2) BRGM (1993a), (3) BME (1987a), (4) BME (1987c), (5) BME (1987b), (6) BRGM (1986b), (7) (BRGM (1986a) and local maps: (8) Reuber (1988), (9) Kusano et al. (2012), (10) Umino et al. (2003), (11) Haymon et al. (1989), (12) A'Shaikh et al. (2005), (13) Alabaster et al. (1980) (14) Robertson and Woodcock (1983), (15) Alabaster and Pearce (1985), (16) Ernewein et al. (1988), (17) JICA (2000), (18) Umino (2012), (19) Kusano et al. (2014), (20) JICA (2002), (21) Reuber et al. (1991), and (22) Adachi and Miyashita (2003).

**Figure 12**. New map of volcanic units in the Semail ophiolite (this study). Strong colours: upper crustal units in outcrop. Pale conjugate colours: upper crustal units inferred as bedrock underlying sedimentary cover. Lower crustal and mantle rocks (copied and simplified from the regional map set) are shown for context. See Sect. 1.4 for changes versus previous maps and

details on bedrock inferences under sediments. VMS deposit locations after Gilgen et al. (2014). The georeferenced, multi-layer version of this Figure, together with the mapping inputs in Figure 11, is provided in the supplementary Geospatial PDF.

**Figure. 13.** Mapping example: tectonic omissions of Geotimes in the Haylayn block near the boninite-hosted Daris-3A VMS deposit. (a) Distribution of volcanic units according to the pre-existing regional geological map (BRGM, 1986a). (b) RTP aeromagnetic survey (Isles and Witham, 1993) with inferred bedrock contacts and faults (this study). (c) Revised map (this study) with sample location symbols as in Fig. 11. Coordinate grid: UTM 40 N in km.

**Figure. 14.** Mapping example: The fault-bounded, northeast-dipping volcanic section at Suhaylah, modified after Robertson and Woodcock (1983) and BME (1987c), showing Tholeiitic Alley lavas (this study) overlying a thin Geotimes section and underlying the type-locality Suhaylah sediments. Our samples analysed to prove unit affiliations are numbered: (1) SP18-A1, (2) SP18-A3, (3) TB4-17J, (4) SP18-A8, (5) SP18-A9.

**Figure. 15.** Mapping example: part of a Lasail basaltic accumulation in the northern Fizh block near the Tholeiitic Alley-hosted Mandoos VMS deposit (Gilgen et al., 2014). (a) Distribution of volcanic units according to the pre-existing regional geological map (BRGM, 1993b). (b) RTP aeromagnetic survey (Isles and Witham, 1993) with inferred bedrock contacts and faults (this study). (c) Revised map (this study) with sample location symbols as in Fig. 11. Coordinate grid: UTM 40 N in km.

**Figure 16.** Boninites capping the volcanic succession in the Wadi Rajmi and Wadi Zab'in region and their structural connection to previously-mapped boninite dykes, ultramafic and orthopyroxene-series intrusives ((Adachi and Miyashita, 2003; BRGM, 1993b; Umino et al., 1990; Usui and Yamazaki, 2010), mantle shear zones (Boudier and Al-Rajhi, 2014; BRGM, 1993b; Takazawa et al., 2003), and podiform chromitites (Boudier and Al-Rajhi, 2014; BRGM, 1993b; Rollinson, 2008). The boninite-hosted Safwa VMS deposit is also shown (Gilgen et al., 2014). Coordinate grid: UTM 40 N in km. Sample symbology as in Figure 11. Strong colours denote outcrops; pale conjugate colours denote inferred bedrock beneath gravels.

**Figure 17.** Interpreted location of the 'Bowling Alley' rift, modified after Alabaster and Pearce (1985), BME (1987a), Robertson and Woodcock (1983) and Smewing et al. (1977). The rift is genetically associated with numerous VMS deposits, minor copper showings and areas of disseminated sulphide mineralization (Haymon et al., 1989; Smewing et al., 1977).

**Figure 18.** Comparison of the expanded Semail volcanic unit fields with Izu-Bonin-Mariana (IBM) proto-arc volcanic rocks, north Tonga trench boninites and MORB glasses. West Pacific boninite analyses from N. Tonga Trench/N. Lau basin (Falloon et al., 2007; Falloon and Crawford, 1991), Izu-Bonin forearc (IODP Expedition 786; Murton et al., 1992), and Bonin forearc (IODP Expedition 352; Shervais et al., 2019). Normal Forearc Basalt (N-FAB) and Primitive Forearc Basalt (P-FAB) from Shervais et al. (2019) and other IBM forearc basalts (FAB) from Reagan et al. (2010) and Ishizuka et al. (2011). High Ti/V

forearc basalts from DeBari et al. (1999) and Hickey-Vargas et al. (2018). MORB glasses from Jenner and O'Neill (2012).

**Table 1**. Comparison of different naming schemes for the Semail volcanostratigraphy and the closest-equivalent rock types identified in the Izu-Bonin-Mariana (IBM) protoarc record.

| Alabaster et al. (1980) | BME, (1987a) & BGRM, (1987a) | JICA (2000) | Godard et al. (2003) | Kusano et al. (2012, 2014) | Gilgen et al. (2014); **This study** | Goodenough et al. (2014) | Rock types | Similar IBM protoarc units* |
|---|---|---|---|---|---|---|---|---|
| Salahi | SE3 | V3 | V3 | V3 | Salahi | – | Alkali basalt | – |
| – | SE2 | V2 | V2 Type II | UV2 | Boninitic Alley | Phase 2 | Boninite, high-Mg andesite | Low Si boninite series |
| Cpx-phyric | | | V2 Type I | LV2 | Tholeiitic Alley | | Basalt, high-Mg andesite, dacite, rhyolite | Forearc basalt series |
| Alley | | | | | | | | |
| Lasail | SE1 | V1 type 2 | – | UV1 | Lasail | | Primitive, low-Ti basalt | Primitive forearc basalt |
| Geotimes | | V1 type 1 | V1 | MV1 | Geotimes | Phase 1 | Basalt, basaltic andesite, andesite | No exact equivalent: High Ti/V axial basalts |
| | | | | LV1 | | | | |

(–) = not included in that study; *Reagan et al. (2017), Shervais et al. (2018), see Section 8 for comparison.

**Table 2.** Areal extents and inferred volumetric proportions of each Semail volcanic unit based on the bedrock map in Fig. 12 for both the Batinah Coast and the structurally-intact western Fizh and Hilti blocks. Proportions are given both as fractions of the volcanics, and of the upper crust (including the Sheeted Dike Complex, SDC).

| Unit | Batinah coast | | | Fizh & Hilti blocks | | | Summary | |
| --- | --- | --- | --- | --- | --- | --- | --- | --- |
| | Bedrock area (km²) | Vol% of upper crust | Vol% of volcanics | Bedrock area (km²) | Vol% of upper crust | Vol% of volcanics | Rock types | Vol% of upper- crust (Fizh & Hilti) |
| Boninitic Alley | 120 | 6.0 | 11 | 50 | 15 | 24 | Boninite-series | 15 |
| Undifferentiated Alley | 148 | 7.4 | 13 | 2 | 0.6 | 1.0 | Post- and off-axis volcanics | 29 |
| Tholeiitic Alley | 416 | 21 | 37 | 82 | 24 | 40 | | |
| Lasail | 76 | 3.8 | 6.7 | 14 | 4.1 | 6.8 | | |
| Geotimes | 374 | 19 | 33 | 58 | 17 | 28 | Phase 1 axial lavas and dykes | 54 |
| SDC | 708 | 35 | – | 124 | 37 | – | | |
| Upper-crustal intrusions | 170 | 8.4 | – | 9.0 | 2.7 | – | | |
| Total volcanic rocks | 1134 | 56 | | 206 | 61 | | | |
| Total upper crust | 2012 | | | 339 | | | | |

Vol%: Volume fraction assumed equivalent to areal fraction

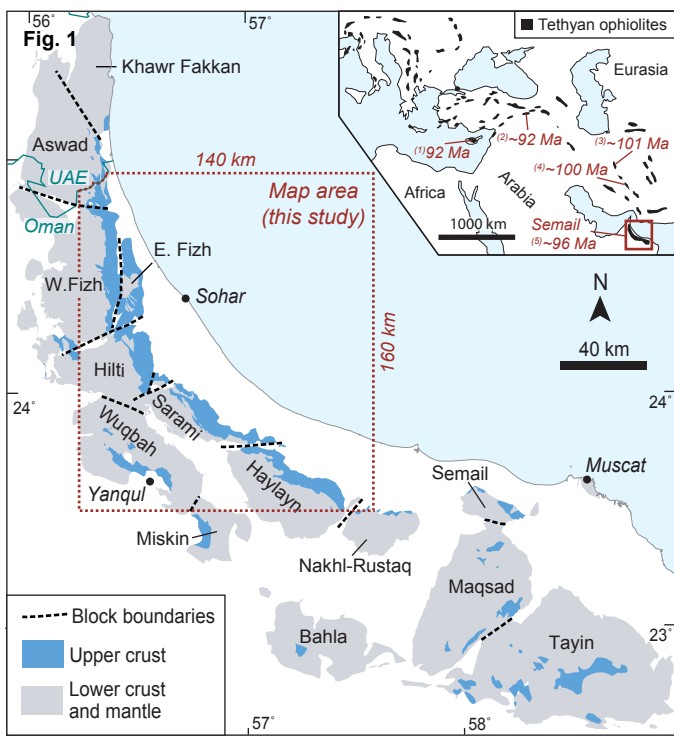

Fig. 1

**Fig. 2**

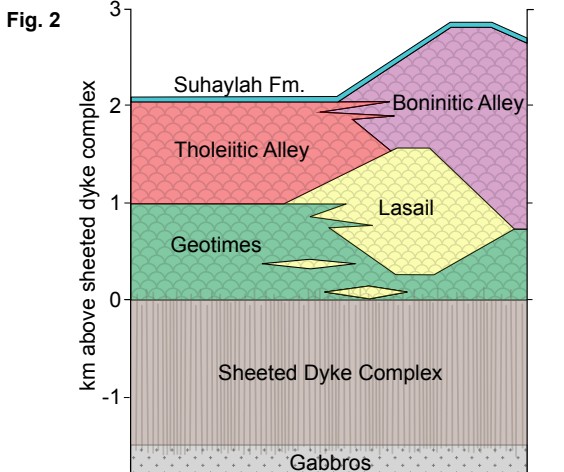

**Fig. 3**

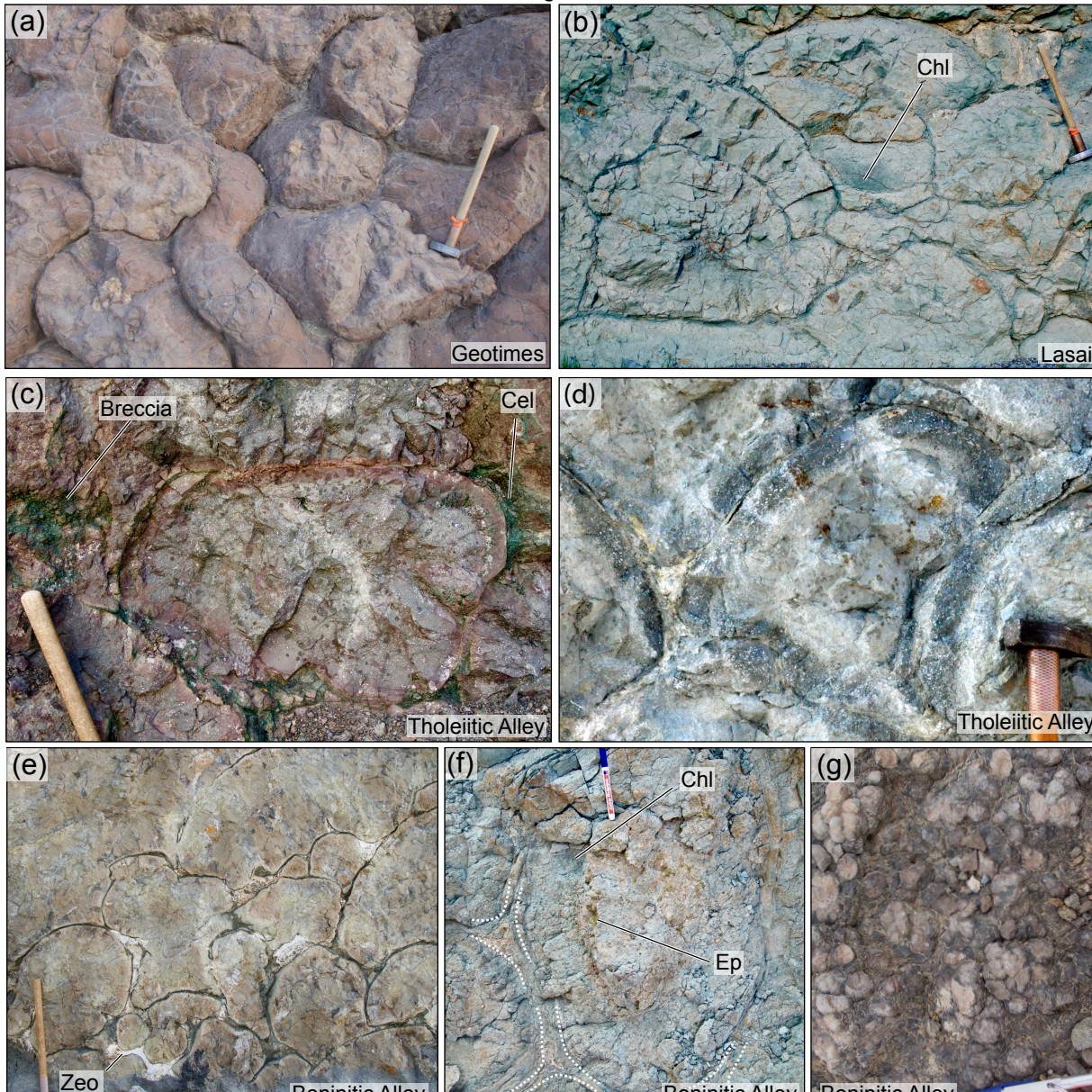

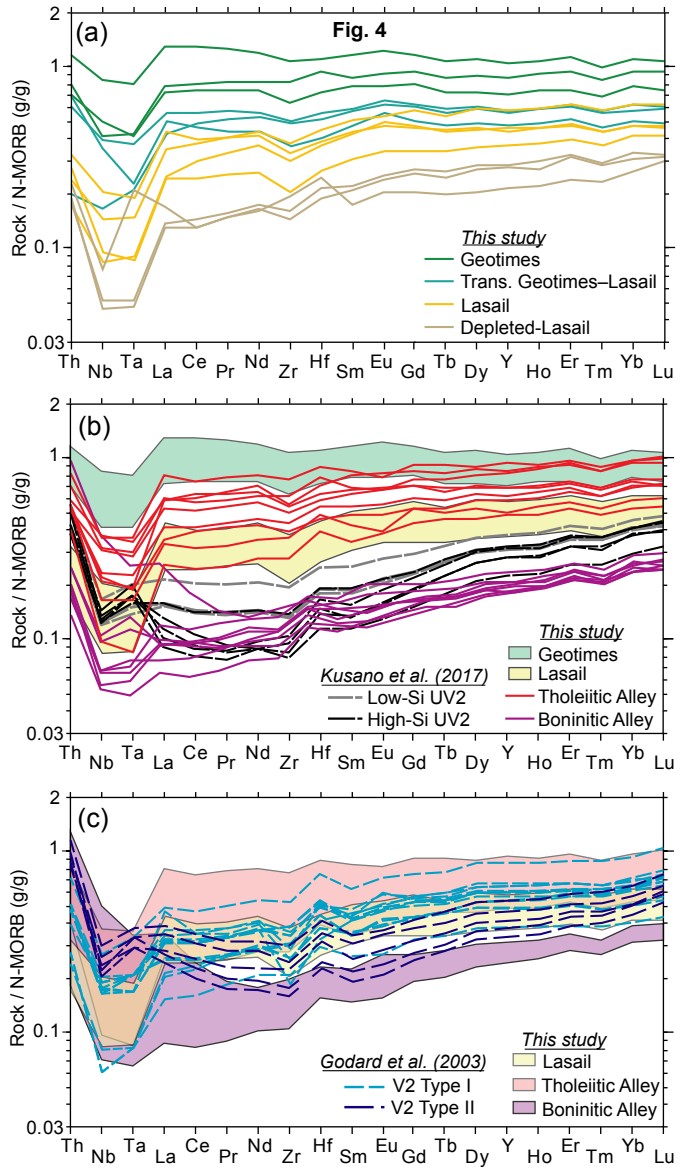

Fig. 4

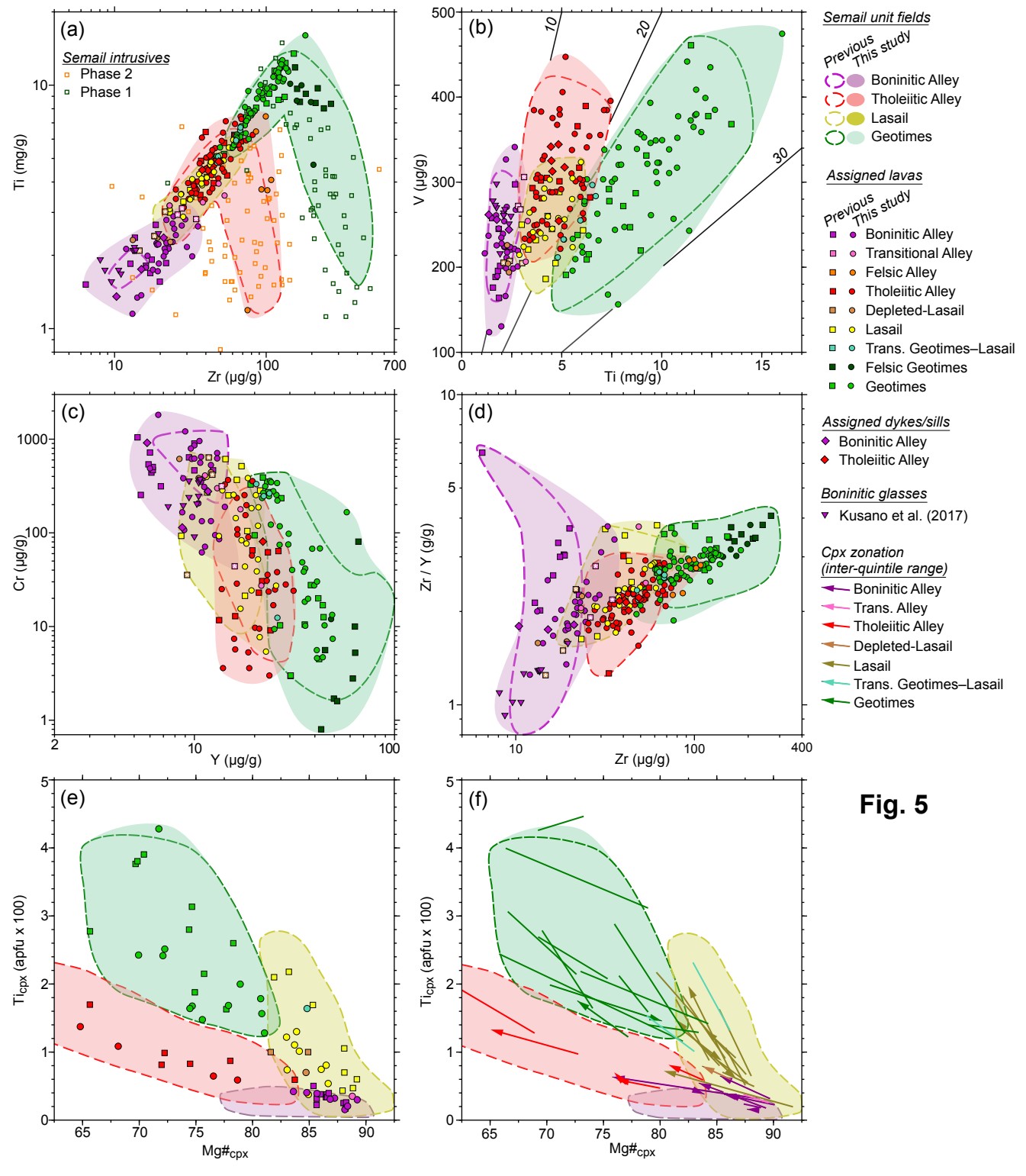

**Fig. 5**

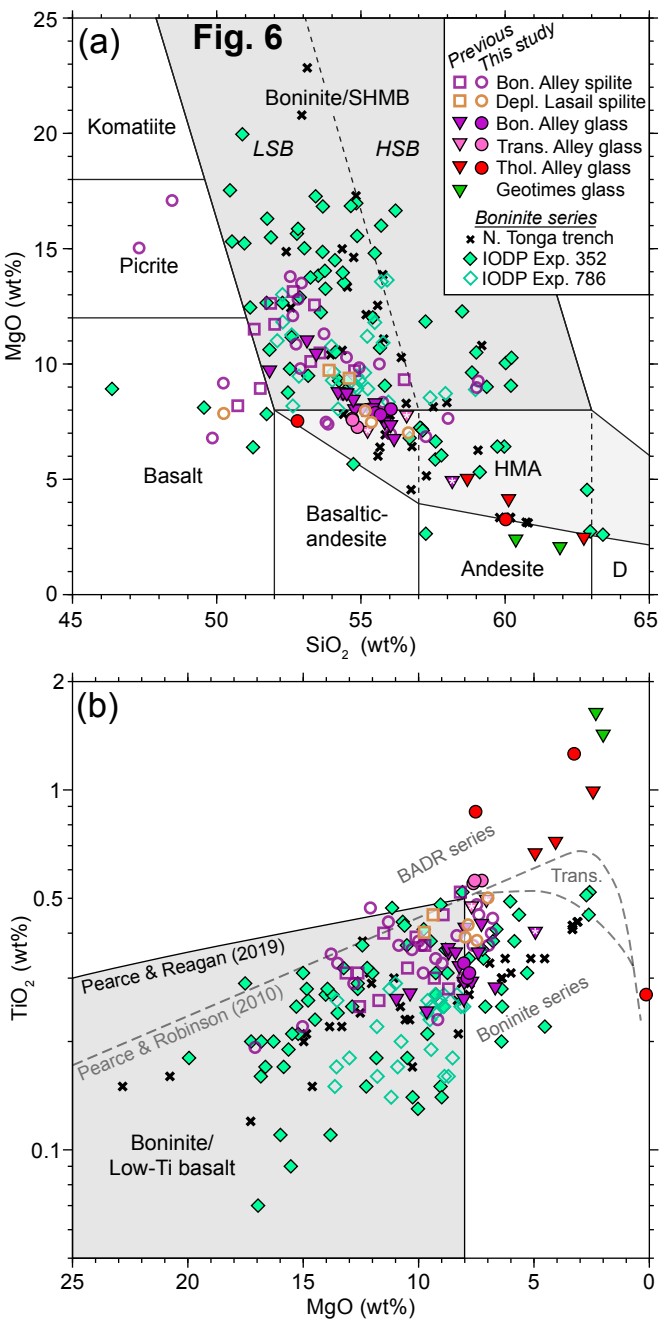

Fig. 6

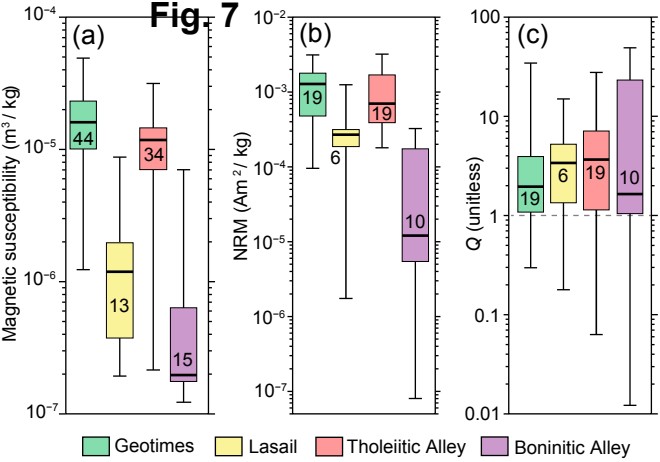

**Fig. 7**

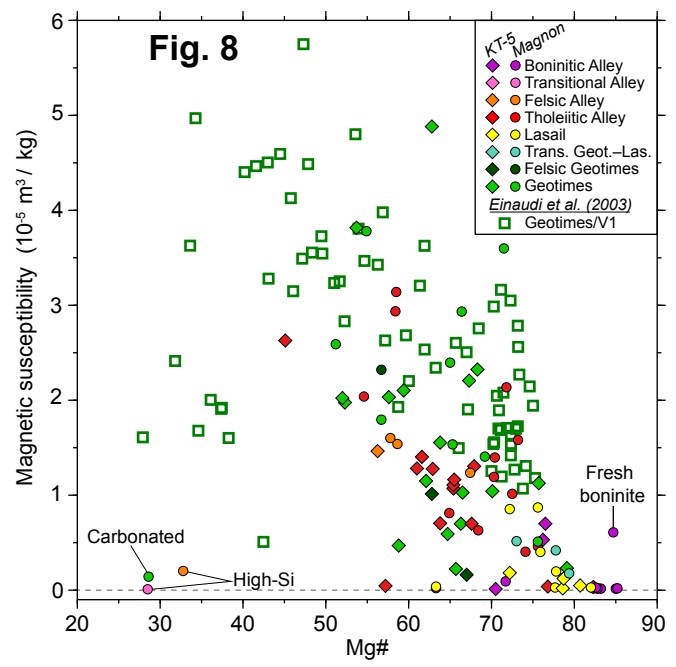

**Fig. 8**

**Fig. 9**

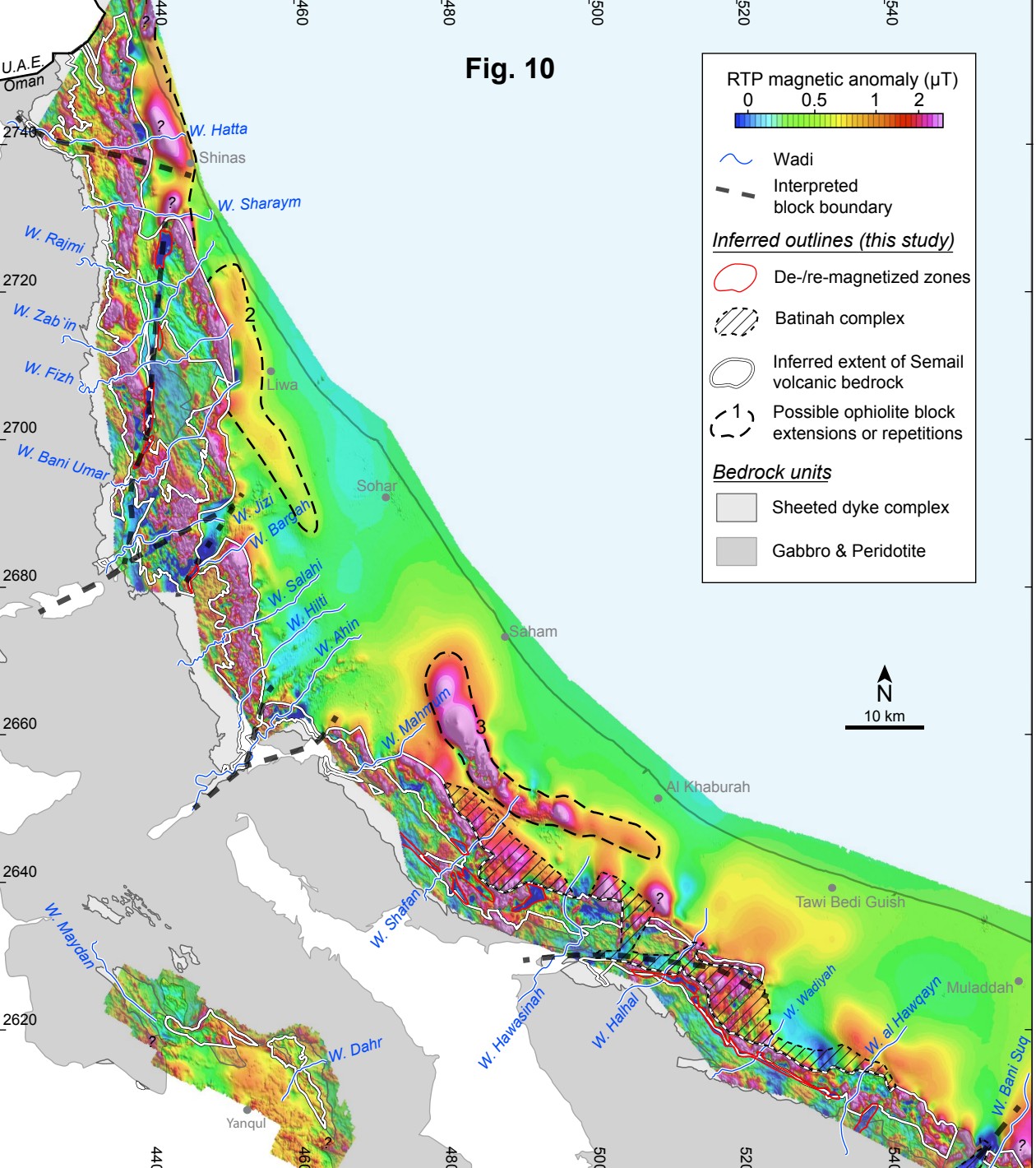

**Fig. 10**

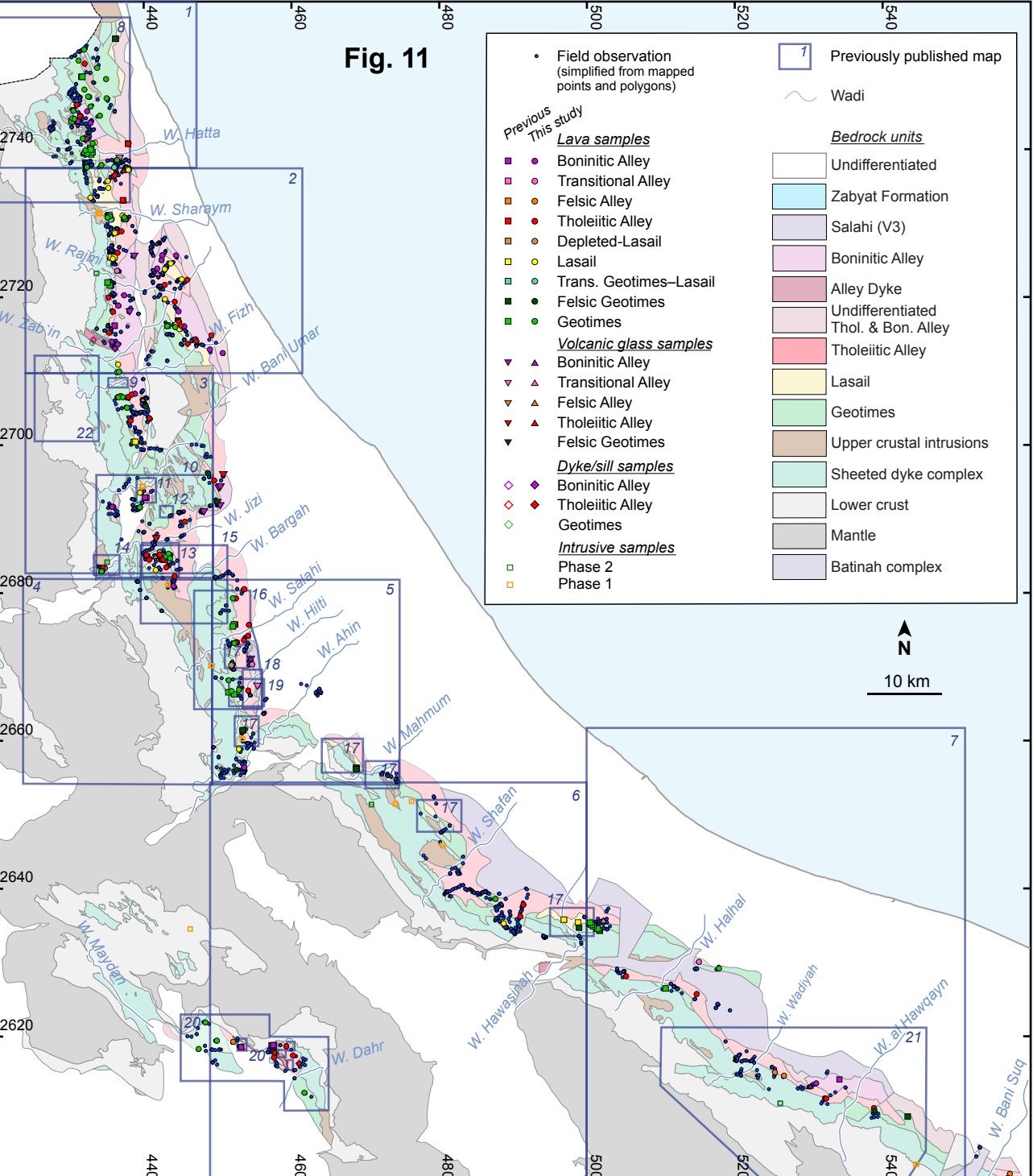

**Fig. 11**

Field observation (simplified from mapped points and polygons)

*1* Previously published map

〜 Wadi

*Previous* *This study*

**Lava samples**
- ■ ● Boninitic Alley
- ■ ● Transitional Alley
- ■ ● Felsic Alley
- ■ ● Tholeiitic Alley
- ■ ● Depleted-Lasail
- ■ ● Lasail
- ■ ● Trans. Geotimes–Lasail
- ■ ● Felsic Geotimes
- ■ ● Geotimes

**Volcanic glass samples**
- ▼ ▲ Boninitic Alley
- ▽ ▲ Transitional Alley
- ▽ ▲ Felsic Alley
- ▼ ▲ Tholeiitic Alley
- ▼ Felsic Geotimes

**Dyke/sill samples**
- ◇ ◆ Boninitic Alley
- ◇ ◆ Tholeiitic Alley
- ◇ Geotimes

**Intrusive samples**
- □ Phase 2
- □ Phase 1

**Bedrock units**
- Undifferentiated
- Zabyat Formation
- Salahi (V3)
- Boninitic Alley
- Alley Dyke
- Undifferentiated Thol. & Bon. Alley
- Tholeiitic Alley
- Lasail
- Geotimes
- Upper crustal intrusions
- Sheeted dyke complex
- Lower crust
- Mantle
- Batinah complex

N

10 km

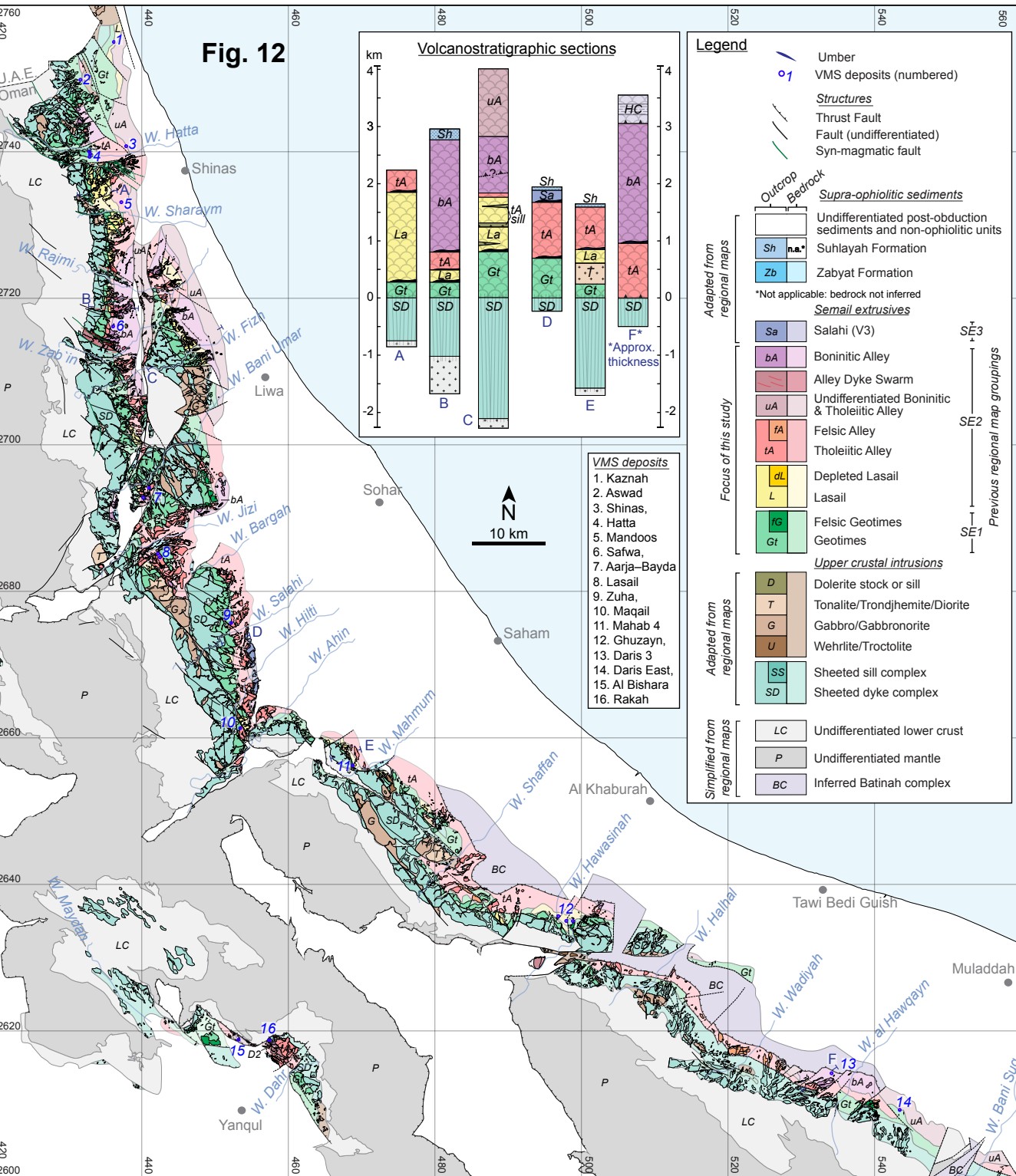

Fig. 12

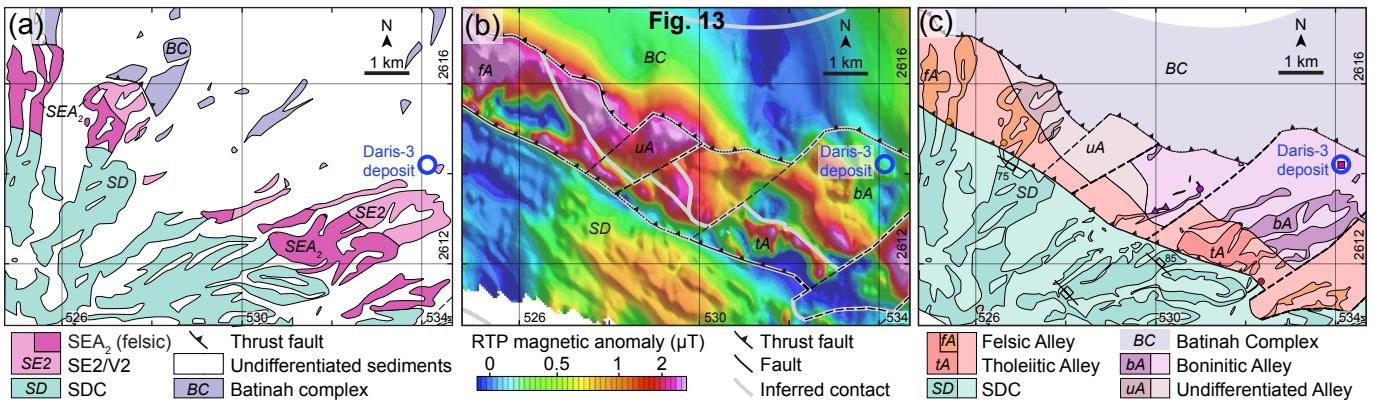

**(a)**
N
1 km
*BC*
*SEA₂*
*SD*
*SE2*
*SEA₂*
*SEA₂*
Daris-3 deposit
526  530  534  2616  2612

SEA₂ (felsic)
SE2/V2
SD  SDC
Thrust fault
Undifferentiated sediments
BC  Batinah complex

**(b)** Fig. 13
N
1 km
*fA*
*BC*
*uA*
*SD*
*bA*
*tA*
Daris-3 deposit
526  530  534  2616  2612

RTP magnetic anomaly (µT)
0  0.5  1  2
Thrust fault
Fault
Inferred contact

**(c)**
N
1 km
*fA*
*BC*
*uA*
*SD*
*bA*
*tA*
Daris-3 deposit
526  530  534  2616  2612

fA  Felsic Alley
tA  Tholeiitic Alley
SD  SDC
BC  Batinah Complex
bA  Boninitic Alley
uA  Undifferentiated Alley

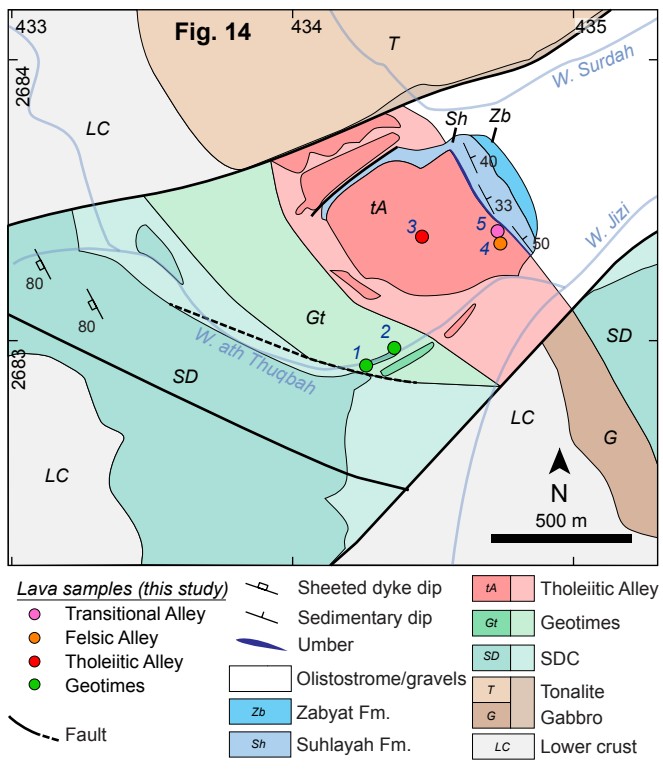

**Fig. 14**

*Lava samples (this study)*
- ⬤ (pink) Transitional Alley
- ⬤ (orange) Felsic Alley
- ⬤ (red) Tholeiitic Alley
- ⬤ (green) Geotimes

⚹ Sheeted dyke dip
⚻ Sedimentary dip
▬ Umber
▭ Olistostrome/gravels
Zb Zabyat Fm.
Sh Suhlayah Fm.

- - - Fault

tA Tholeiitic Alley
Gt Geotimes
SD SDC
T Tonalite
G Gabbro
LC Lower crust

N
500 m

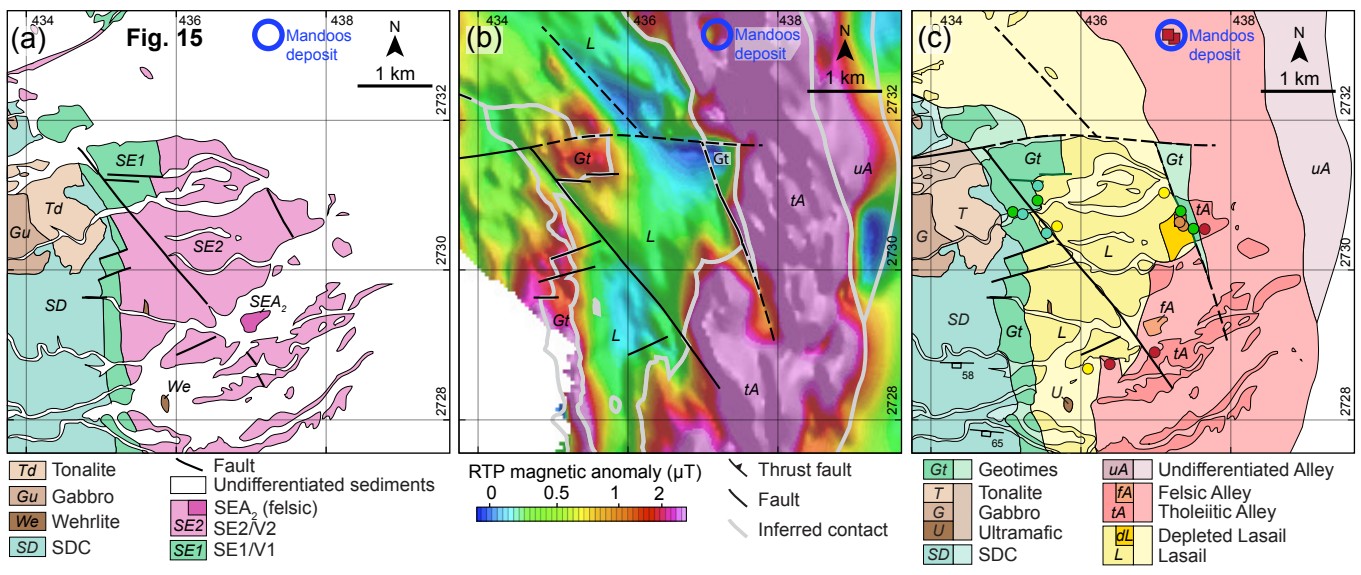

**Fig. 15**

(a)

| | | | | |
|---|---|---|---|---|
| *Td* | Tonalite | | | Fault |
| *Gu* | Gabbro | | | Undifferentiated sediments |
| *We* | Wehrlite | | *SEA₂* | SEA₂ (felsic) |
| *SD* | SDC | | *SE2* | SE2/V2 |
| | | | *SE1* | SE1/V1 |

**RTP magnetic anomaly (µT)**
0   0.5   1   2

Thrust fault
Fault
Inferred contact

| | | | | |
|---|---|---|---|---|
| *Gt* | Geotimes | | *uA* | Undifferentiated Alley |
| *T* | Tonalite | | *fA* | Felsic Alley |
| *G* | Gabbro | | *tA* | Tholeiitic Alley |
| *U* | Ultramafic | | *dL* | Depleted Lasail |
| *SD* | SDC | | *L* | Lasail |

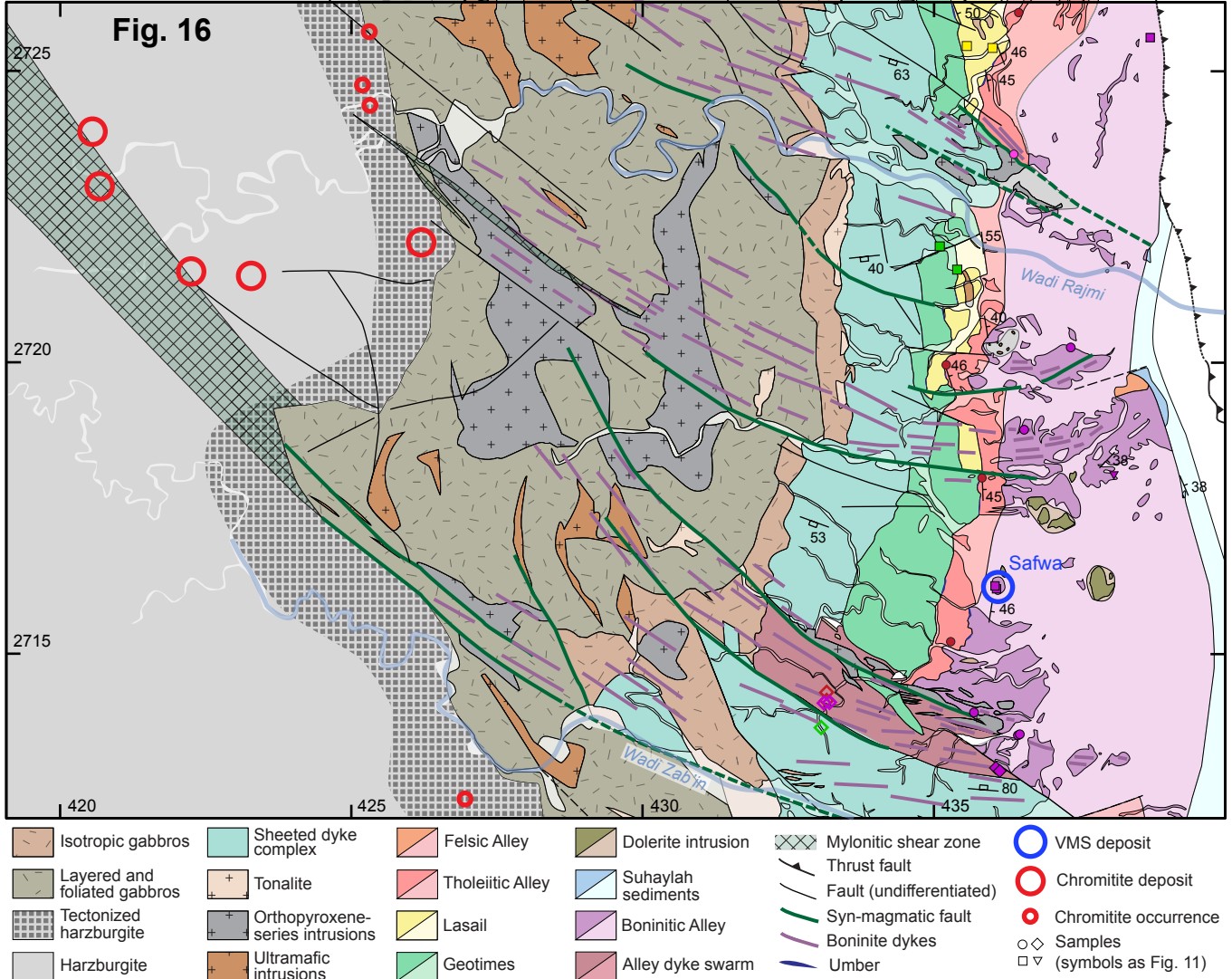

**Fig. 16**

Legend:

- Isotropic gabbros
- Layered and foliated gabbros
- Tectonized harzburgite
- Harzburgite
- Sheeted dyke complex
- Tonalite
- Orthopyroxene-series intrusions
- Ultramafic intrusions
- Felsic Alley
- Tholeiitic Alley
- Lasail
- Geotimes
- Dolerite intrusion
- Suhaylah sediments
- Boninitic Alley
- Alley dyke swarm
- Mylonitic shear zone
- Thrust fault
- Fault (undifferentiated)
- Syn-magmatic fault
- Boninite dykes
- Umber
- VMS deposit
- Chromitite deposit
- Chromitite occurrence
- Samples (symbols as Fig. 11)

Labels on map: 2725, 2720, 2715, 420, 425, 430, 435, Wadi Rajmi, Wadi Zabin, Safwa, 50, 46, 45, 63, 55, 40, 41, 46, 45, 53, 38, 38, 80

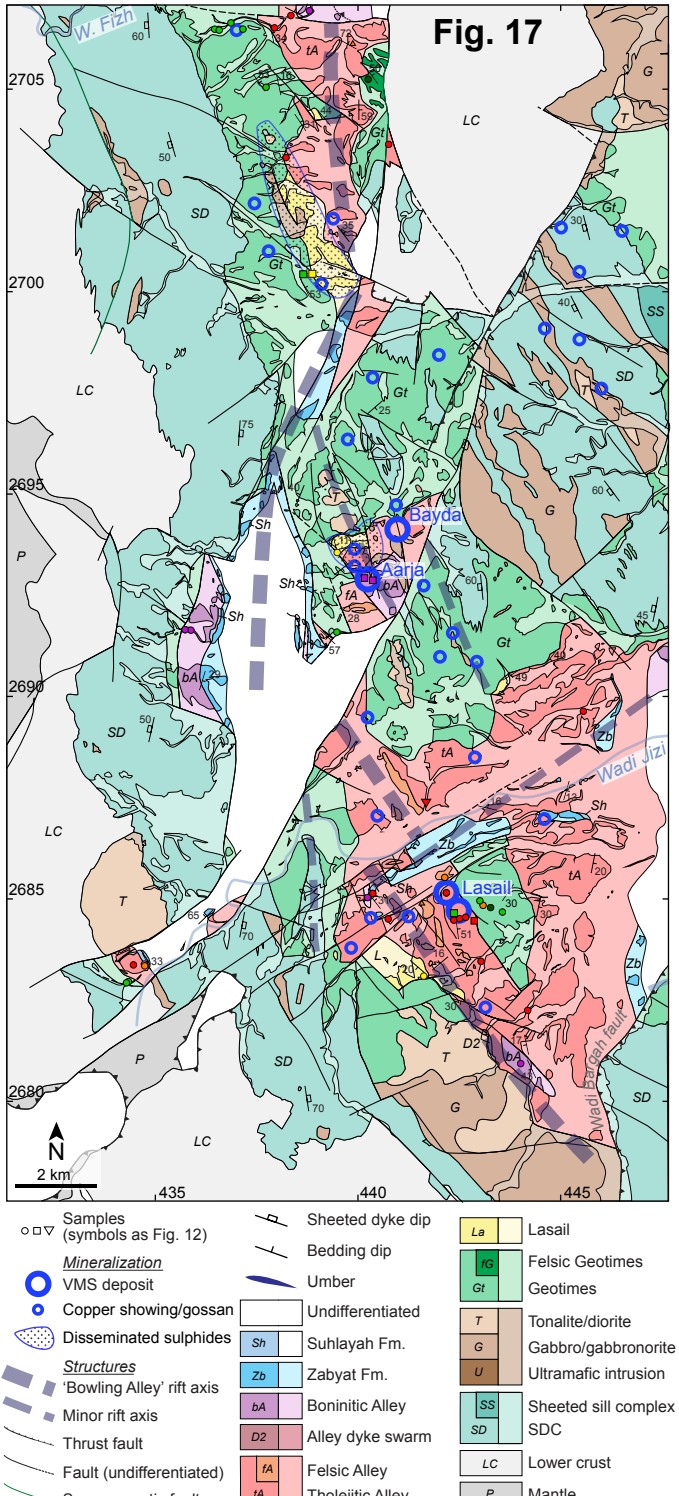

**Fig. 17**

Samples (symbols as Fig. 12)

*Mineralization*
- VMS deposit
- Copper showing/gossan
- Disseminated sulphides

*Structures*
- 'Bowling Alley' rift axis
- Minor rift axis
- Thrust fault
- Fault (undifferentiated)
- Syn-magmatic fault

- Sheeted dyke dip
- Bedding dip
- Umber
- Undifferentiated
- *Sh* Suhlayah Fm.
- *Zb* Zabyat Fm.
- *bA* Boninitic Alley
- *D2* Alley dyke swarm
- *fA* Felsic Alley
- *tA* Tholeiitic Alley

- *La* Lasail
- *fG* Felsic Geotimes
- *Gt* Geotimes
- *T* Tonalite/diorite
- *G* Gabbro/gabbronorite
- *U* Ultramafic intrusion
- *SS* Sheeted sill complex
- *SD* SDC
- *LC* Lower crust
- *P* Mantle

Bayda
Aarja
Lasail

W. Fizh
Wadi Jizi
Wadi Bardgen fault

2 km
N

**Fig. 18**

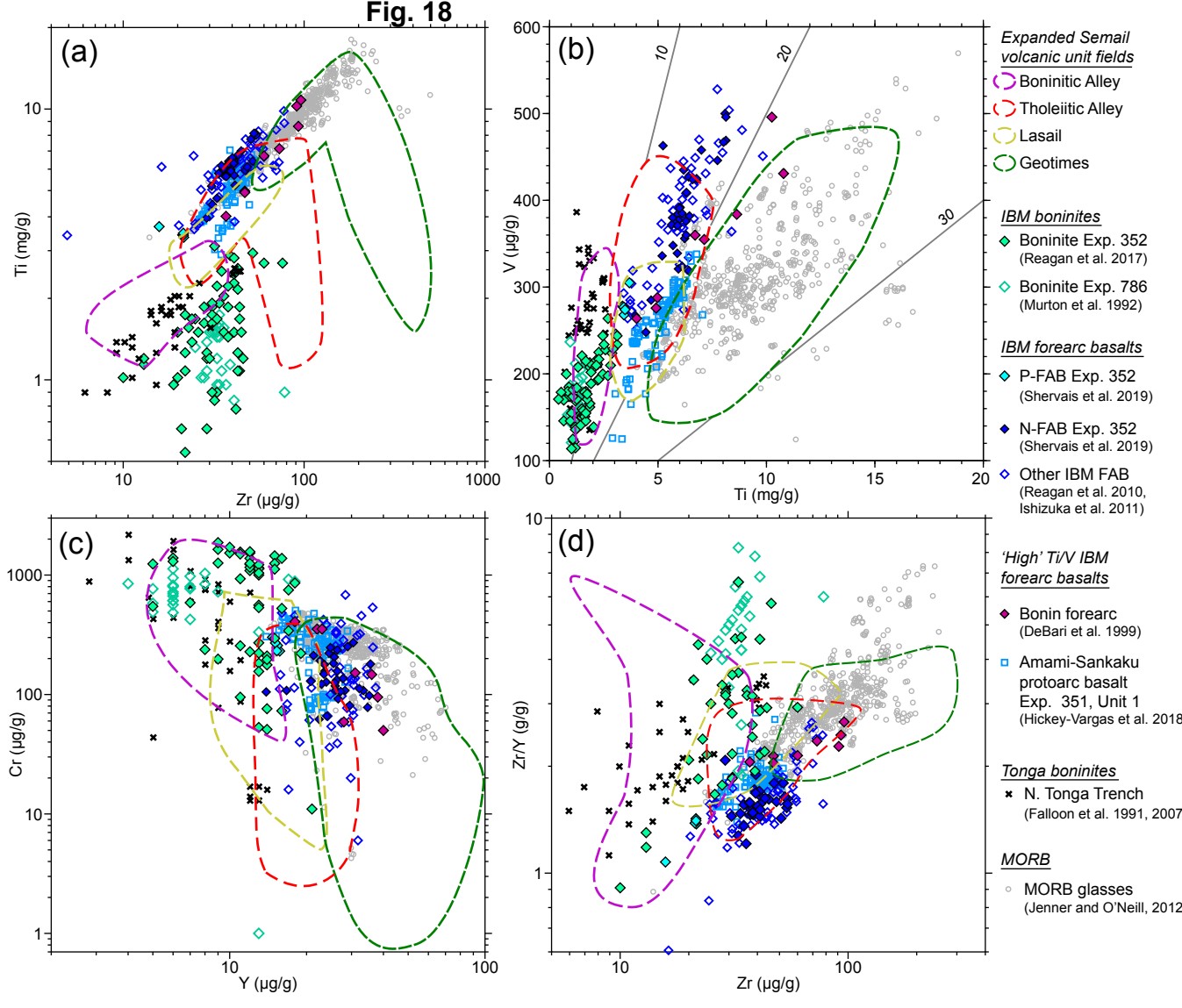