# Peer review of "A revised map of volcanic units in the Oman ophiolite: insights into the architecture of an oceanic proto-arc volcanic sequence"

_Solid Earth, 2019_

## Referee Comment (RC1) · Kathryn Goodenough (Referee) · 15 May 2019

I really enjoyed reading this paper and I think it's an excellent, well-written, thorough piece of work following a rigorous methodology. It's great to see such attention to detail applied to mapping of part of the Oman-UAE ophiolite, and as such I only really have a few very minor comments to make.

In the introduction, I note that you mention both names for the ophiolite (Semail and Oman-UAE) but then decide to go with the term Semail – this can be confusing, of course, because there is a Semail block within the ophiolite. Later on (e.g. section 1.4.1) you use the term 'Oman ophiolite' instead – it's probably good to be consistent!

[Figure]

I also note that you say your map is of the 'northern Semail volcanics' – why did you choose not to study the southeastern blocks? I have a suspicion that the Semail and Wadi Tayin blocks are somewhat different to the rest of the ophiolite, but consistent studies along the whole length are needed to confirm this. Also, I note that you didn't continue your map into the UAE and would be interested to know why this was - were our UAE maps difficult to obtain, or did you look at them and decide that it was not useful to extend into the UAE? The fact that there are volcanics to the north and southeast of your mapping area should at least be mentioned in the paper.

In section 1.1, you mention the previously published geological maps but don't cite any references to them. I understand you only want to include this long list of citations once, but it would be worth putting in a link to the section where they occur. As a minor point, it would be useful if some place names mentioned in the text (e.g. the Mandoos VMS deposit) were shown on the maps in the paper.

The field observations generally seem excellent, and very thorough – I absolutely agree that careful and extensive field observation is important in developing maps of the Oman-UAE ophiolite. At the start of section 4.1, please state how many samples were collected. The geochemistry section is very good and I find Figure 5 really useful as a data compilation. As you make clear, there is some overlap between the fields in some areas and this is only to be expected, as these groups represent a progressive evolution in both mantle source and degree of melting.

I can't really comment on the aeromagnetic survey section, as this is not my area of expertise. I do wonder whether some magnetic anomalies could be related to rocks beneath the volcanics – for example wehrlite intrusions at shallow levels might affect the mapped magnetic anomalies?

The map is great, and a nicely detailed piece of work. I found your discussion in section 7.3 interesting, because we mostly supported the idea of a 'gradient of subduction-zone influence' due to the absence of published evidence about the extent of later
magmatism in Oman. Your work, and the recent work of Haase, de Graaff and others, all confirms that the 'Phase 2' magmatism is extensive as far southeast as the Semail Gap. I still think there's some uncertainty about how extensive it is in the Semail and Wadi Tayin blocks on the southeast side of the Semail Gap, which is a question that deserves more study.

Do your Lasail 'seamounts' (section 7.5) generally correspond to areas with Phase 2 intrusive complexes, as originally mapped and described in Lippard et al. (1986) and more recently discussed by de Graaff et al. (2019)? Wadi Rajmi is an excellent example of an area where the whole magmatic plumbing system can be linked together, as you describe in section 7.6.1. Your reference to syn-magmatic faulting is consistent with our work in the UAE which also showed ductile faults controlling emplacement of many Phase 2 intrusions, and I know from my own observations that similar features occur in Wadi Rajmi. I think there is more work to be done to understand the complete magmatic architecture of 'Phase 2' magmatism in the Oman-UAE ophiolite (I am using the term Phase 2 because so far we have not managed to subdivide the intrusive rocks as effectively as has been done for the volcanics). Overall though, I'm very impressed with the work in this paper and would like to extend my congratulations to the whole team!

K M Goodenough

---

## Author Comment (AC1) · 21 May 2019

We thank Dr Goodenough for her effort in reviewing our manuscript and are glad that she enjoyed the read and found the work sound.

A full list of revisions will be posted together with the revised manuscript after the discussion period has closed. In this reply we address the main points of discussion raised by the Reviewer.

Regarding the name we use for the ophiolite, we agree that this could be more consistent in our text. We had intended the 'Oman ophiolite' to denote the part of the

ophiolite within Oman, but this is evidently not entirely clear to the reader. On the other hand, using the name 'Oman–UAE ophiolite', although unambiguous, could imply that our field area extends into the UAE, which it does not. As there are other ophiolites in Oman, we thus favour the traditional 'Semail ophiolite' designation, and we will update the manuscript accordingly.

Concerning the northern and southern limits of our mapping area, outcrops of volcanic rocks outside of our area are indicated by the existing regional maps in the southeastern ophiolite blocks, along the western side of the ophiolite, and in the UAE. The stratigraphy and geochemistry of the volcanic rocks in these areas has long been overlooked in favour of the more extensive outcrops along the Batinah coast, which lie within our mapping area. Our mapping area was chosen so as to cover the majority of outcropping volcanic rocks and prospective tenements for VMS exploration. The extent of the 1992 Batinah Aeromagnetic survey, which does not reach into UAE, also defined the area over which we could infer the presence of volcanics under cover.

In addition to these reasons, our decision to exclude the southeastern blocks was largely due to the practical constraints of mapping with a small team, with limited time, from a base in Sohar. It may be relatively straightforward to take the methodology outlined in this study and apply it to the sparsely outcropping volcanics south of the Semail and Tayin blocks. A comparable geophysical survey exists for that area, and the Washihi VMS deposit, currently in the mine development phase, is located in the Tayin block.

Based on the evidence for subduction zone influence in the southeastern blocks as described by de Graaff et al. (2019), Haase et al. (2016), MacLeod et al. (2013), Rollinson and Adetunji (2013, 2015), we suspect that the volcanic rocks in these blocks could be rather similar to those in our mapping area. It is feasible that the idea of a weaker 'Phase 2' overprint in the southeastern blocks stems from differences in the outcrop quality of the volcanic sequence.

As with the southeastern ophiolite blocks, it should be relatively straightforward to continue mapping the different volcanic units into the UAE, with the excellent British Geological Survey maps as a base and with our study as a guide. As suggested, we will add a few sentences describing the extent of volcanic outcrops outside of the mapping area to Section 1.4.3. Scope of new map.

With reference to the question of whether some of the magnetic anomalies could be related to shallow intrusions beneath the volcanics, this is a valid and interesting point. As tested with field magnets, the intrusive rocks are generally less magnetic than their volcanic equivalents. However, in well-exposed areas, negative anomalies over the intrusives are only well-resolved when the bodies are of considerable size (greater than ∼200–500 m across) and are emplaced in strongly-magnetized volcanics. For example, in Fig. 14, the reduced-to-pole magnetism of a large gabbro–tonalite intrusive complex can be seen as weak, and comparable to that of the sheeted dykes. This indeed implies that another possible source of patches of low magnetism could be hidden, shallow intrusions, and we will add this point the list in Section 5.4. However, as these intrusions only make up ∼8 vol% of the upper crust, and even less of the volcanic sequence, we expect this effect to be minor overall.

With regards to the spatial relations between the Lasail seamounts and the late intrusive complexes, our map rather supports a comagmatic connection between the Alley lava units and the intrusive bodies. Late intrusive complexes often do underlie or appear in the vicinity of the Lasail seamounts (e.g. Fig. 14), but there are also many cases where Lasail accumulations are not underlain by late intrusives. There are other arguments in favour of the Phase 2 intrusive complexes being related to the Alley lavas. Firstly, the greater proportion of Alley volcanics relative to Lasail suggests that Alley should have a more significant proportion of intrusive equivalents. Secondly, the intrusive complexes characteristically span a range of compositions, from gabbroic to tonalitic, often within single complexes (Lippard et al., 1986). This compositional series is characteristic of the Alley lava suite, but not of Lasail. For instance, in Fig. 14 the entire Lasail accumulation is made up of pale, primitive basalts, whereas the underlying intrusive complex consists of roughly equal portions of gabbro and tonalite. Similarly, the well-documented plutonic complex in the Lasail mine area, depicted in the southeastern corner of Fig. 16, is made up of gabbros and quartz diorite (Tsuchiya et al., 2013). Though this complex has been linked to the Lasail lava unit on the basis of its location, the intrusive complex and emanating dyke swarms cut through Alley lavas on its eastern side, showing it is rather related to late Alley phase magmatism. This major intrusive complex is shallowly emplaced under the lavas surrounding it (Tsuchiya et al., 2013), and thus underlies, and probably feeds, the significant accumulations of Felsic Alley lavas found just to the north of the exposed intrusives. While considering the intrusive–extrusive connection, we did notice a loose spatial association between upper-crustal ultramafic intrusions (wehrlites where checked) and the Lasail unit. Wehrlite bodies are not exceedingly common in the upper crust, but where they do occur, they usually underlie, or are intruded into, Lasail lavas (e.g. just south of Wadi Hatta; the small ultramafic body in Fig. 14; around Wadis Hilti–Ahin; Wadi Mahmum; Wadi Hawasina, beneath the Ghuzayn deposit). The particularly primitive, wet melts associated with both wehrlite and Lasail lava petrogenesis may further support this connection (Belgrano and Diamond, 2019; Koepke et al., 2009). We can add some sentences to the manuscript on these observations, and welcome any other thoughts on the subject.

References cited:

Belgrano, T. M. and Diamond, L. W.: Subduction-zone contributions to axial volcanism in the Oman-U.A.E. ophiolite, Lithosphere, doi:10.1130/L1045.1, 2019.

de Graaff, S. J., Goodenough, K. M., Klaver, M., Lissenberg, C. J., Jansen, M. N., Millar, I. and Davies, G. R.: Evidence for a Moist to Wet Source Transition Throughout the Oman-UAE Ophiolite, and Implications for the Geodynamic History, Geochemistry, Geophysics, Geosystems, 0(0), doi:10.1029/2018GC007923, 2019.

Haase, K. M., Freund, S., Beier, C., Koepke, J., Erdmann, M. and Hauff, F.: Constraints on the magmatic evolution of the oceanic crust from plagiogranite intrusions in the Oman ophiolite, Contributions to Mineralogy and Petrology, 171(5), 1–16, doi:10.1007/s00410-016-1261-9, 2016.

Koepke, J., Schoenborn, S., Oelze, M., Wittmann, H., Feig, S. T., Hellebrand, E., Boudier, F. and Schoenberg, R.: Petrogenesis of crustal wehrlites in the Oman ophiolite: Experiments and natural rocks, Geochemistry, Geophysics, Geosystems, 10(10), doi:10.1029/2009GC002488, 2009.

Lippard, S. J., Shelton, A. W. and Gass, I. G.: The Ophiolite of Northern Oman, Blackwell Scientific Publications Ltd. [online] Available from: https://books.google.ch/books?id=PCkcAQAAIAAJ, 1986.

MacLeod, C. J., Johan Lissenberg, C. and Bibby, L. E.: "Moist MORB" axial magmatism in the Oman ophiolite: The evidence against a mid-ocean ridge origin, Geology, 41(4), 459–462, doi:10.1130/G33904.1, 2013.

Rollinson, H. and Adetunji, J.: Mantle podiform chromitites do not form beneath mid-ocean ridges: A case study from the Moho transition zone of the Oman ophiolite, Lithos, 177, 314–327, doi:10.1016/j.lithos.2013.07.004, 2013.

Rollinson, H. and Adetunji, J.: The geochemistry and oxidation state of podiform chromitites from the mantle section of the Oman ophiolite: A review, Gondwana Research, 27(2), 543–554, doi:https://doi.org/10.1016/j.gr.2013.07.013, 2015.

Tsuchiya, N., Shibata, T., Yoshikawa, M., Adachi, Y., Miyashita, S., Adachi, T., Nakano, N. and Osanai, Y.: Petrology of Lasail plutonic complex, northern Oman ophiolite, Oman: An example of arc-like magmatism associated with ophiolite detachment, Lithos, 156–159, 120–138, doi:10.1016/j.lithos.2012.10.013, 2013.

---

## Referee Comment (RC2) · Yuki Kusano (Referee) · 22 May 2019

This is well described map integrated geology, geochemistry and magnetism data. I really enjoyed reading because your careful treatment of original field evidence in detailed area especially on Alley/Phase 2 magmatism. Volcanic units lost vast of outcrops and block continuity in the Oman ophiolite, and it makes hard to interpret the volcanism and magmatism, but RTP data well cover the weak point as underlying bedrock. This will be helpful information for economic geology of volcanic units but also field trip.

On the other hand, I think that interpretation of the ophiolite genesis still has some problems as indicated below.

[Figure]

1. Comparison with IBM arc Authors indicates similarity between the Oman ophiolite and IBM arc, but I think some are rough discussion. In Table 1, IBM protoarc equivalents are compared with volcanostratigraphy in Oman ophiolite. Please present the reference. If this is your interpretation, please write the reason (geochemistry, chronology or something?). In figure 5 and 6, geochemical data of the IBM arc looks scattered. Discussion 7.6 concludes that "boninitic Alley is compositionally closer ...Izu-Bonin forearc boninite than north Tonga...", but it is not visible.

2. Genesis of Oman ophiolite You point out that existence of Alley/Phase 2 volcanism comparable to Geotimes/Phase 1 volcanism reinforce subduction-influenced character in the Oman ophiolite. Recently broadly accepted that the Oman ophiolite records both axial and subduction phase magmatism. Mantle diapir is tectonic feature (structural geology) not geochemical/magmatic feature, so your data does not support that point. This manuscript cannot deny the existence of mantle diapir. I think it might be existing even the Phase 1 was subduction-related (e.g. backarc spreading or forearc spreading).

Minor comments

There is some typo. Please double check all company name and area name (BGRM –> BRGM, Harami –> Sarami?).

I know well that even geological map set (BME, 1987 and BRGM, 1987) not fit each other. How did you sew the discontinuity? Based on your original field evidence? It should be shown in Table 1 or manuscript.

Please unify your figure number in figure and manuscript: Figure 3a or Figure 3A?

Figure 4: geochemical reference of V2 type I and II are not shown.

Table S1: What your mean of negative sign (-) in "field character" column? Do you mean that only reliable way to discrimination is geochemical character?

Is your description of "transitional Alley" and "transitional Lasail" geochemical feature

or mappable feature?

Figure 9: (e) –> (c); Bar chart is difficult to understand without "All unit". Do you double or triple count magnetites? I think it is not necessary to show because total sample only 6-14 in each unit.

Figure 10: Please show reference on each zone. Whose Batinah complex, or satellite? Is "major fault zone" corresponds to RTP data same as Figure 1? I think both topographic ophiolite and inferred volcanic bedrock should be shown to support your integrated map. Moreover, put color with "weakly magnetic zones (P26L3)" will be helpful because general reader does not know detailed volcanostratigraphy.

Figure 16: Type locality of boninite lava along Wadi Jizi (Ishikawa et al., 2002; OM16-46C in Kusano et al., 2017) has been changed to tholeiitic Alley? It would be different from Table S2 to geological map. Around Suhaylah village, you mentioned occurrence of ~300 m thick tholeiitic Alley. Representative field photo in Figure 3c resembles to Geotimes lava in other area. Detailed discussion with enlarged RTP map like Figure14 will support your interpretation.
* * *

---

## Author Comment (AC2) · 6 Jun 2019

We are glad Dr Kusano enjoyed the read and thank her for her careful review. The comments are insightful and will greatly help to improve the manuscript. In the following, we discuss each point raised by the Reviewer and our proposed changes. We will then submit a detailed list of revisions arising from all the discussion comments with the revised manuscript.

The two main concerns raised by the Reviewer concern the genetic and tectonic context in which we have presented the volcanic map. Establishing this context was not the main aim of this study, and we hope that the map will be helpful regardless of which

consensus is eventually reached on the tectonic setting of each magmatic phase. Accordingly, we rely principally on previous studies for contextualizing the mapping work. These studies, especially those utilizing magmatic geochemistry and geochronology, have found many similarities between the IBM proto-arc sequence and the Semail sequence. We reviewed and added to this evidence more thoroughly in a recent paper Belgrano and Diamond (2019).

Regarding the issue of equivalence between the Semail and IBM units raised by the Reviewer, we agree that supporting evidence for this equivalence should be more clearly shown. We believe noting this equivalence will help workers unfamiliar with the Semail ophiolite to understand and use our map for comparison with other settings. To support the claims of equivalence, we will add the relevant IBM references to Table 1, and add a dedicated section to the Discussion, including a Figure based on Fig. 5a–d showing the Semail unit fields in comparison to the IBM lavas. We will also clarify that these are not exact equivalents, but rather the closest-available equivalents in a potentially analogous proto-arc sequence.

The Reviewer also points out that, in our Figures 5 and 6 the IBM boninite compositions are scattered and that the similarity between the Semail and IBM boninites, mentioned in Section 7.6, is not apparent. Our intention was to highlight the rather similar MgO contents of the Semail and IBM boninites (< 17 wt% MgO), and to contrast them with the remarkably magnesian Tonga boninites (up to 22 wt% MgO). We agree that in our other Figures, compositional distinctions are not apparent. Also, both the Semail and W. Pacific boninite compositions are rather scattered, in part due to the low concentrations. We have moved this discussion to a new dedicated discussion section (Section 8) with its own figure, and will clarify this text to point out the scatter and specify that the compositional similarity is limited to MgO content.

In the second main point, the Reviewer notes that a history of both axial and subduction phase magmatism is broadly accepted for the ophiolite. We agree on this point, with the important qualifier that we believe the available evidence, reviewed in Belgrano

and Diamond (2019), indicates that the axial phase also occurred above a subduction zone. This subduction influence then became more pronounced during Phase 2 Alley magmatism (Alabaster et al., 1982; Kusano et al., 2014, 2017; Umino et al., 1990).

The Reviewer also points out that the mantle diapirs are tectonic features, and that our data cannot deny the existence of such features. We agree completely on these points and had not intended to argue otherwise. We would, however, argue that mantle diapirs are linked to the magmatic system through their roles in melt generation and as sites of melt migration (Nicolas, 1986; Rabinowicz et al., 1987). We did not intend to call into question the existence of these carefully-mapped diapiric structures (e.g., Nicolas et al., 2000; Rabinowicz et al., 1987). Our point is that the significant Phase 2 volcanism documented in this study implies that melt-generating structures in the mantle (diapirs) may conceivably be coeval with and related to Phase 2 magmatism. Therefore, unless independent evidence suggests otherwise, it should not necessarily be assumed that all of the mantle diapirs belong to the axial phase. Up to 14 such diapirs have been structurally mapped by Nicolas et al. (2000). Although an abundance of troctolites above the Maqsad diapir suggests that it at least partly formed during the MORB-like axial phase, the same is not clear for many of the other diapirs (Python and Ceuleneer, 2003). We will add a few sentences clarifying this to Section 7.2: Proportions of the upper crustal units, and cite the compatible findings of Python and Ceuleneer (2003), who found that only a minority ($\sim$25%) of mantle dykes and cumulates in Oman can be related to the axial, MORB-like phase.

The Reviewer also made several minor comments, which we respond to below.

We are grateful that the Reviewer spotted some typographical errors, and inconsistent Figure naming (e.g. 3a vs 3A). We will correct these errors and carefully proof-read the revised manuscript.

The Reviewer pointed out that some of the existing regional geological maps are inconsistent with one another and questioned how we dealt with this. Fortunately, these

differences were not found to be significant for the volcanic rocks. The only significant difference we noticed was between the BRGM Fizh and BME Wadi Bani Umar map sheets at the gabbro–SDC contact. As mapping this contact was not the aim of our study, we simply drew a compromise between the two maps onto our map. However, in light of the Reviewer's comment, we reconsidered this area, and have used the more recent field map of Adachi and Miyashita (2003) to draw the contact in this area.

The Reviewer requested the addition of V2 type I and II (Godard et al., 2003) compositions to Fig 4, as was erroneously indicated in the caption, but had been cut in the Figure.. To support our discussion and grouping of these units, we will add a third panel of MORB-normalized trace element plots with the Godard et al. (2003) units to Fig. 4.

The Reviewer questioned the meaning of the (-) symbol for Field Character in Table S1. Indeed, this symbol indicates that the field character was ambiguous or not clear. We will add a footnote to the table to clarify this.

The Reviewer asked whether the transitional features are mappable or geochemical features. So far, we mainly recognized the transitional nature of these rocks in terms of geochemistry and stratigraphic position. The field character of the units is indeed often intermediate between the units in question, though this is typically easier to recognize in hindsight, with the knowledge of their geochemistry. In any case, the differences are subtle and difficult to map, especially without continuous outcrop. We will add this explanation to Section 4.5: Interpretation of transitional compositions, to clarify this.

The Reviewer found the bar chart in Fig. 9 difficult to understand, questioning whether we needed to show each unit and whether we counted minerals multiple times. Showing the magnetic carriers on both an 'all unit' and unit basis takes up little extra space and shows the interesting progression from dominantly secondary to primary magnetic carriers upwards through the stratigraphy. As shown in Figs. 9e & d, several magnetic carrier minerals can exist within a sample. In the bar chart, the minerals are counted

more than once if more than one is present in a single sample. We will add a note to the Fig. caption explaining this.

In Fig. 10, the Reviewer asks for references for each zone. The extents of these zones are all our own interpretations based on the aeromagnetic map, and we have updated the inset caption with (this study) to reflect this. The interpretation of the repeated ophiolite blocks and overlying Batinah Complex is supported by the gravity modelling of (Shelton, 1990), the structural observations of Woodcock and Robertson (1982b, 1982a) and outcrop mapping on the Yanqul and As Suwayq map sheets (BRGM, 1986b, 1986a). This is mentioned in the text, but we will also add these supporting references to the caption.

The Reviewer asked for a more concrete definition of what is meant by weak and strongly magnetic, in terms of colours on the RTP map. Outlining all these zones in Fig. 10 would render the underlying magnetic map unreadable. Instead, in the Section 5.4 (Observed reduced-to-pole anomalies) text we will define the typical values of weakly and strongly magnetic zones in RTP $\mu$T, which should allow the reader to more easily follow our references to Fig. 10.

The Reviewer also suggests that topography and volcanic bedrock units be added to Fig. 10. We agree that these layers would better help show how these datasets were integrated. Regarding the topography, the freely-available digital elevation models (NASA space shuttle and ASTER) are unfortunately noisy over the low relief of much of the volcanic terrain, which results in somewhat messy contouring and hill-shading. Any shading would also interfere with the interpretation of the magnetic colour scale. We hope that topographic data will be included into new editions of the regional map set if our map is incorporated. Until then, in Section 1.4.3 we have suggested that our map be used in tandem with the existing regional maps for this reason. Regarding the volcanic bedrock units, such details are difficult to show at the scale of the map shown in Fig. 10. We opted to show the outline of all the volcanic bedrock for this reason, as it can be shown with a single line. Adding shaded or hatched units would

hinder reading the underlying magnetic colour scale. To maintain the readability of Fig. 10, we show close-up examples of differently magnetized zones in Figs. 13 and 14. To address the Reviewer's concerns and aid the interested reader in understanding the integration of these datasets, we will add the aeromagnetic map as a layer to the supplementary Geospatial PDF, and mention this in Section 6: Map Construction and Presentation and in the Fig. 10 caption. This way, Fig. 10 remains readable, and the different datasets and maps can be easily viewed as layers at different transparencies, and at different scales, by interested readers.

With reference to Fig. 16, the Reviewer asks whether the type locality for boninites (Ishikawa et al., 2002) has been reclassified as Tholeiitic Alley. We have not reclassified the type locality for boninites. The type locality is not visible in Fig. 16; it lies just outside the Figure to the east, in an area that we have mapped as Boninitic Alley, in agreement with Ishikawa et al. (2002). We used the boninite sample OM16-46C and other samples from Kusano et al.,(2017), as well as our own field observations, to define the extent of boninites in this area.

The Reviewer also mentions that the representative field photo of a Tholeiitic Alley pillow in Fig. 3c resembles Geotimes lavas in the area. We agree that the overall shape and brownish-red colour of the pillow do resemble Geotimes lavas, and that is probably why this outcrop was originally mapped as Geotimes. However, the high vesicularity and black spherules of the pillows at this locality and intercalated andesite massive flows (not pictured) are in our experience diagnostic of Tholeiitic Alley. To confirm this, we collected five samples from along the Suhalylah volcanic section and mapped these outcrops in detail, so as be certain that Geotimes is not in fact the uppermost unit, as previously reported. The stratigraphically lower two samples were proven geochemically to be Geotimes, and the upper three were found to be Tholeiitic and Transitional Alley. For us, this unequivocally confirms that Alley lavas occupy the top of the Suhaylah volcanic section. To further support this point, we will add another representative photo of Tholeiitic Alley to Figure 3, and make a small close-up map figure of the

Suhaylah section in the discussion Section 7.4, as suggested by the Reviewer.

References cited:

Adachi, Y. and Miyashita, S.: Geology and petrology of the plutonic complexes in the Wadi Fizh area: Multiple magmatic events and segment structure in the northern Oman ophiolite, Geochemistry, Geophysics, Geosystems, 4(9), doi:10.1029/2001GC000272, 2003. Alabaster, T., Pearce, J. A. and Malpas, J.: The volcanic stratigraphy and petrogenesis of the Oman ophiolite complex, Contributions to Mineralogy and Petrology, 81(3), 168–183, doi:10.1007/BF00371294, 1982.

Belgrano, T. M. and Diamond, L. W.: Subduction-zone contributions to axial volcanism in the Oman-U.A.E. ophiolite, Lithosphere, doi:10.1130/L1045.1, 2019.

BRGM: Geological Map of As Suwayq (Scale 1:100,000: Sheet NF40-3A)., 1986a.

BRGM: Geological Map of Yanqul (Scale 1:100,000: Sheet NF40-2C)., 1986b.

Godard, M., Dautria, J.-M. and Perrin, M.: Geochemical variability of the Oman ophiolite lavas: Relationship with spatial distribution and paleomagnetic directions, Geochemistry, Geophysics, Geosystems, 4(6), n/a-n/a, doi:10.1029/2002GC000452, 2003.

Ishikawa, T., Nagaishi, K. and Umino, S.: Boninitic volcanism in the Oman ophiolite: Implications for thermal condition during transition from spreading ridge to arc, Geology , 30(10), 899–902, doi:10.1130/0091-7613(2002)030<0899:BVITOO>2.0.CO;2, 2002.

Kusano, Y., Hayashi, M., Adachi, Y., Umino, S. and Miyashita, S.: Evolution of volcanism and magmatism during initial arc stage: constraints on the tectonic setting of the Oman Ophiolite, Geological Society, London, Special Publications , 392(1), 177–193, doi:10.1144/SP392.9, 2014.

Kusano, Y., Umino, S., Shinjo, R., Ikei, A., Adachi, Y., Miyashita, S. and Arai, S.: Contribution of slab-derived fluid and sedimentary melt in the incipient arc magmas with development of the paleo-arc in the Oman Ophiolite, Chemical Geology, 449, 206–

225, doi:10.1016/j.chemgeo.2016.12.012, 2017.

Nicolas, A.: A Melt Extraction Model Based on Structural Studies in Mantle Peridotites, Journal of Petrology, 27(4), 999–1022, doi:10.1093/petrology/27.4.999, 1986. Nicolas, A., Boudier, F., Ildefonse, B. and Ball, E.: Accretion of Oman and United Arab Emirates ophiolite – Discussion of a new structural map, Marine Geophysical Researches, 21(3), 147–180, doi:10.1023/A:1026769727917, 2000.

Python, M. and Ceuleneer, G.: Nature and distribution of dykes and related melt migration structures in the mantle section of the Oman ophiolite, Geochemistry, Geophysics, Geosystems, 4(7), doi:10.1029/2002GC000354, 2003.

Rabinowicz, M., Ceuleneer, G. and Nicolas, A.: Melt segregation and flow in mantle diapirs below spreading centers: Evidence from the Oman Ophiolite, Journal of Geophysical Research: Solid Earth, 92(B5), 3475–3486, doi:10.1029/JB092iB05p03475, 1987.

Shelton, A. W.: The interpretation of gravity data in Oman: constraints on the ophiolite emplacement mechanism, Geological Society, London, Special Publications, 49(1), 459 LP – 471, doi:10.1144/GSL.SP.1992.049.01.29, 1990.

Umino, S., Yanai, S., Jaman, A. R., Nakamura, Y. and Iiyama, J. T.: The transition from spreading to subduction: Evidence from the Semail Ophiolite, northern Oman Mountains, in Oceanic Crustal Analogues: Proceedings of the Symposium "Troodos 1987," pp. 375–384, Cyprus Geological Survey Department, Nicosia. [online] Available from: http://ci.nii.ac.jp/naid/10023911794/en/, 1990.

Woodcock, N. H. and Robertson, A. H. F.: Stratigraphy of the Mesozoic rocks above the Semail Ophiolite, Oman, Geological Magazine, 119(1), 67–76, doi:DOI: 10.1017/S0016756800025668, 1982a.

Woodcock, N. H. and Robertson, A. H. F.: The upper Batinah Complex, Oman: allochthonous sediment sheets above the Semail ophiolite, Canadian Journal of Earth

Sciences, 19(8), 1635–1656, doi:10.1139/e82-140, 1982b.

---

## Author Response (AR1)

**List of revisions to "A revised map of volcanic units in the Oman ophiolite: insights into the architecture of an oceanic proto-arc volcanic sequence"**

(Belgrano et al.)
Solid Earth Discussion Paper SE-2019-69

We gave detailed responses to the reviewer's comments in the interactive discussion replies, and below, the reviewer's comments are paraphrased in red with the corresponding revisions detailed in black. We are grateful for the insightful comments of the reviewers and believe we have adequately addressed their mostly minor concerns with the following changes. Below this list is the working version of the manuscript marked up in Word with 'Track Changes'.

Revisions arising from Kathryn Goodenough's review

Reviewer comment: The ophiolite's name is inconsistent through the manuscript
Revision: We have harmonized the manuscript to use the term Semail ophiolite.

Reviewer comment: Refer to section where previously published maps are listed in Section 1.1.
Revision: We have added a reference to Section 6 to this sentence.

Reviewer comment: Occurrences of volcanic rocks outside of the mapping area are not described.
Revision: Added a list of volcanic occurrences in the ophiolite which lie outside of the mapping area to Section 1.4.3 "Scope of new map".

Reviewer comment: Shallow, buried intrusions may be an additional cause of patchy RTP magnetism.
Revision: Added a sentence explaining this possibility to Section 5.4 "Observed reduced-to-pole anomalies"

Reviewer comment: Is there a connection between the Lasail seamounts and late intrusive complexes?
Revision: The connection is more evident for the late intrusives and the Alley units. To show this, we have added published data of the Semail intrusives to Fig. 5a, which shows that the intrusive compositional arrays are continuous with the volcanic arrays. We have also added a sub-section (7.7) to the discussion explaining this continuity, and the evidence for intrusive–extrusive equivalence which rose out of the manuscript discussion.

Revisions arising from Yuki Kusano's review

Reviewer comment: There are typos through the text
Corrected typos throughout text for BGRM to BRGM, Harami to Sarami

Reviewer comment: References for equivalent rocks types were not clear in Table 1
Revision: Added relevant references to Table 1 footnote.

Review Comment: Mantle diapirs are tectonic features, and it is not clear how our new data has any bearing on their significance.

Revision: We have added two sentences to Section 7.2 clarifying our point that many major tectonic or magmatic features in the ophiolite, including mantle diapirs, could conceivably be related to Phase 2 magmatism, notwithstanding other evidence to the contrary.

Reviewer Comment: Reasoning behind equivalence of IBM rock types now clear

Revision: We moved the references to IBM equivalents to a new discussion section (Section 8) and expanded on the equivalence of the two settings and different units, clarifying the similarities and differences with new whole-rock comparison figure (Fig. 19). We also moved the West Pacific boninite data to Fig. 18 to declutter Fig. 5 and consolidate the IBM comparison into one section.

Reviewer comment: There are mismatches between the regional map set sheets, how did we deal with this?

Revision: We have added a brief comment regarding this point to Section 1.4.1 "Previous geological maps". We found the mismatches to be generally insignificant, with the exception of one example at the gabbro–SDC contact between the Fizh and Wadi Bani Umar map sheets. We have updated this boundary to fit more recent detailed mapping from Adachi and Miyashita (2003), and added this outline (#22) and reference to Fig. 11.

Reviewer comment: Show V2 Type 1&2 (Godard et al. 2003) on Fig. 4 to support interpretation of these units in Table 1.

Revision: We have added a panel to Fig. 4 showing these units, and clarified our interpretation in section 4.4 "Geochemical discrimination of Lasail from Alley".

Reviewer comment: Are the transitional units mappable?

Revision:  We have added a sentence to the end of Section 4.5 explaining that the transitional character of these units' field characteristics is subtle and thus difficult to map in the field.

Reviewer comment: In Fig. 9, the (c) label is missing, and it is not clear whether minerals are counted multiple times.

Revision: We have corrected the lettering error and added a note to the caption explaining that minerals may be counted multiple times.

Reviewer comment: The references behind each zone and what is meant by weakly and strongly magnetic is not clear in Fig. 10.

Revision: We have clarified in the Fig. caption that these references are original interpretations, and to maintain legibility of the Figure, have referred to typical values for weakly and strongly magnetic zones in $\mu$T the text of Section 5.4. We have also added the RTP map to the Geospatial PDF so that interested users can check our interpretations and added a note the Fig. 10 caption explaining this.

Reviewer comment: The Field photo of Tholeiitic Alley in Fig. 3 resembles some Geotimes lavas,

Revision: This photo resembles Geotimes in some ways (colour, shape), but is also one of our best photos showing the key features which distinguish such similar lavas from Geotimes (vesicles, dark spherules). To address the reviewer's comment, we added another photo of Alley lavas to Fig. 3 to show the variability in the unit's appearance.

Reviewer comment: The claims of Tholeiitic Alley's presence at the top of the Suhaylah section would benefit from a dedicated close up map Fig.
Revision: We have added close-up Fig. 14 of the Suhaylah area, showing the sample locations and numbers which prove the presence of Tholeiitic Alley here. The aeromagnetic map does not extend over the thin faulted offshoot where the Suhaylah section is located. However, as both Geotimes and Alley are similarly fresh and magnetic along this section, the aeromagnetic coverage might not be particularly helpful.

Other Author changes

We added the Oman–UAE border to Fig. 1 and updated the map area to exclude the UAE.

We added three lava and two dyke samples collected and analyzed by XRF in 2019 to the map and the supplementary data set and made resulting minor adjustments to the map in the Yanqul (more Geotimes lavas), East Fizh (changed a strip of bA to uA), and Wadi Zab'in ( wider Phase 2 dyke complex) areas. The unit areas in Table 2 have been accordingly updated.

We updated the boninite classification schemes in Fig. 6 to that of the newly published Pearce & Reagan (2019) fields. This allowed us to simplify and update the text describing the boninite major element compositions as Low-Si Boninite in Section 4.7. We also updated the Tholeiitic Alley descriptions to include high-magnesium andesites, which is apparent from previously measured data (Kusano et al. 2017) plotted with the updated fields in Fig. 6.

We deleted a few points Alley and Geotimes points in Fig. 5c with Cr concentrations below the calculated XRF detection limit (and thus unreliable concentrations) and reduced the lower expansion of the Alley and Geotimes unit fields accordingly.

We have reemphasized the significance of the 'Depleted Lasail' compositions as marking the onset of boninitic volcanism in the ophiolite in Sect. 4.6, added a corresponding sentence to the abstract and conclusions. This potentially important finding was somewhat lost in the text previously. To support this, we have added the Depleted Lasail major element compositions to Fig. 6 to show their boninitic composition,

[revised manuscript text omitted]

Ballhaus, C., Fonseca, R. O. C., Münker, C., Kirchenbaur, M. and Zirner, A.: Spheroidal